# A Thorough Review of the Clinical Applications of Artificial Intelligence in Lung Cancer

**DOI:** 10.3390/cancers17050882

**Published:** 2025-03-04

**Authors:** Serafeim-Chrysovalantis Kotoulas, Dionysios Spyratos, Konstantinos Porpodis, Kalliopi Domvri, Afroditi Boutou, Evangelos Kaimakamis, Christina Mouratidou, Ioannis Alevroudis, Vasiliki Dourliou, Kalliopi Tsakiri, Agni Sakkou, Alexandra Marneri, Elena Angeloudi, Ioanna Papagiouvanni, Anastasia Michailidou, Konstantinos Malandris, Constantinos Mourelatos, Alexandros Tsantos, Athanasia Pataka

**Affiliations:** 1Adult ICU, General Hospital of Thessaloniki “Ippokrateio”, Konstantinoupoleos 49, 54642 Thessaloniki, Greece; chris1mourat@gmail.com (C.M.); giannis.alevroudis@gmail.com (I.A.); vicky_dourliou@hotmail.com (V.D.); kalliopi.tsakiri@gmail.com (K.T.); agnisak@hotmail.com (A.S.); alexia.mrn@gmail.com (A.M.); elena-angeloudi@hotmail.com (E.A.); 2Pulmonary Department, Unit of thoracic Malignancies Research, General Hospital of Thessaloniki “G. Papanikolaou”, Aristotle’s University of Thessaloniki, Leoforos Papanikolaou Municipality of Chortiatis, 57010 Thessaloniki, Greece; diospyrato@yahoo.gr (D.S.); kporpodis@yahoo.gr (K.P.); kellybio4@hotmail.com (K.D.); 3Pulmonary Department General, Hospital of Thessaloniki “Ippokrateio”, Konstantinoupoleos 49, 54642 Thessaloniki, Greece; afboutou@yahoo.com (A.B.); alextsantos@yahoo.com (A.T.); 41st ICU, Medical Informatics Laboratory, General Hospital of Thessaloniki “G. Papanikolaou”, Aristotle’s University of Thessaloniki, Leoforos Papanikolaou Municipality of Chortiatis, 57010 Thessaloniki, Greece; vakaimak@yahoo.gr; 54th Internal Medicine Department, General Hospital of Thessaloniki “Ippokrateio”, Aristotle’s University of Thessaloniki, Konstantinoupoleos 49, 54642 Thessaloniki, Greece; ioanna.d.pap@gmail.com; 62nd Propaedeutic Internal Medicine Department, General Hospital of Thessaloniki “Ippokrateio”, Aristotle’s University of Thessaloniki, Konstantinoupoleos 49, 54642 Thessaloniki, Greece; manastasia96@gmail.com; 72nd Internal Medicine Department, General Hospital of Thessaloniki “Ippokrateio”, Aristotle’s University of Thessaloniki, Konstantinoupoleos 49, 54642 Thessaloniki, Greece; kostas_malandris@yahoo.gr; 8Biology and Genetics Laboratory, Aristotle’s University of Thessaloniki, 54624 Thessaloniki, Greece; mourelatos@yahoo.com; 9Respiratory Failure Clinic and Sleep Laboratory, General Hospital of Thessaloniki “G. Papanikolaou”, Aristotle’s University of Thessaloniki, Leoforos Papanikolaou Municipality of Chortiatis, 57010 Thessaloniki, Greece; patakath@yahoo.gr

**Keywords:** artificial intelligence, artificial neural network, convolutional neural network, lung cancer, thorough review

## Abstract

In this thorough review, the applications of artificial intelligence (AI) with respect to the epidemiology (prevention and screening), clinical features, diagnosis (X-rays; chest computed tomography, CT; positron emission tomography, PET), biomarkers (biopsy, staging), treatment (general information, surgical treatment, radiotherapy, chemotherapy, targeted therapy, immunotherapy), and prognosis of lung cancer are summarized. AI can help in the discrimination between benign and malignant lung nodules, in the detection of biomarkers related to lung cancer years before its development, in the recognition of specific histologic or genetic markers of lung tumors and in the planning of personalized treatment for lung cancer patients that improve their prognosis.

## 1. Introduction

According to the World Health Organization (WHO), the number of deaths due to trachea, bronchus, and lung cancers rose to 1.8 million worldwide in 2019, remaining the leading cause of death from malignancies and the sixth overall; furthermore, for upper-middle- and high-income countries, they are ranked fourth [1]. Yet, significant breakthroughs in their management take place every year, with the implementation of artificial intelligence (AI) being one of the most influential changes over the last decade [2]. AI models have been used for the prevention of lung cancer [3] and in the development of early screening models, using lung cancer’s epidemiological and clinical features [4] along with various imaging modalities, such as chest X-ray [5], computed tomography (CT) [6], or positron emission tomography (PET) [7]. Furthermore, AI has helped to improve various diagnostic methods, such as bronchoscopy [8], endobronchial ultrasound (EBUS) [9], fine needle aspiration biopsy (FNAB) [10], and pathology and/or cytology results [11], as well as staging [12]. Finally, through the use of AI, surgical, radial, and pharmacological treatment methods have been improved [13,14,15] and prognostic models have become more precise [16]. The rationale behind the writing of the present review is that, until ten years ago, the number of papers that had been published regarding the application of AI in the context of lung cancer were minimal. However, this figure has changed dramatically over the last decade—especially in the last five years—with the number of relevant publications having increased exponentially. Although numerous reviews have focused on specific aspects of the applications of AI for lung cancer, such as prevention, screening, clinical features, diagnosis, treatment, or prognosis, there are very few that have included all the aforementioned aspects and reviewed them thoroughly and comprehensively. Furthermore, virtually thousands of new papers on this subject are being published each year, making existing knowledge obsolete at an extremely rapid pace. For all of the abovementioned reasons, we proceeded to consolidate all existing knowledge regarding the application of AI for lung cancer in a thorough review conducted from the physician’s point of view, which seems particularly necessary at present.

## 2. Materials and Methods

For the purpose of the present review, we performed a PubMed search in “All Fields” using the search terms “artificial intelligence” OR “artificial neural network” OR “convolutional neural network” OR “deep learning” OR “machine learning” OR “omics” AND “lung cancer” up to 31 December 2023, with no start date. This search identified 6095 unique results, 11 of which were duplicates and were removed before screening. During screening, no studies were removed by the automation tool, 147 studies were removed as their reports could not be retrieved, 9 studies were removed because they were editorials, 28 studies were removed as they were comments on articles, 7 studies were removed because they were meeting abstracts, and 5420 studies were removed as they were irrelevant to the research theme. As a result, 5622 items were removed in total, and 473 studies were finally included in this review. Interestingly, more than 75% of all the papers included in this review were published within the last five years and more than 90% in the last decade. Articles were categorized based on the thematic sectors of this review. Some of the studies included were included in more than one thematic sector. For ‘epidemiology’, 78 studies were used (14 for the ‘prevention’ sector and 69 for the ‘screening’ sector). For the ‘clinical features’ sector, 23 studies were used. For ‘diagnosis’, 350 studies were used (25 for the ‘X-ray’ sector, 181 for the ‘chest CT’ sector, 44 for the ‘PET’ sector, 62 for the biomarkers sector, 80 for the ‘biopsy’ sector, and 62 for the ‘staging’ sector). For ‘treatment’, 183 studies were used (75 for the ‘general information’ sector, 21 for the ‘surgical treatment’ sector, 25 for the ‘radiotherapy’ sector, 13 for the ‘chemotherapy’ sector, 26 for the ‘targeted therapy’ sector and 42 for the ‘immunotherapy’ sector). Finally, 123 studies were used for the ‘prognosis’ sector. The PRISMA flow diagram of study selection is shown as Figure 1. For any studies written by one or more co-authors of the present review, an independent author, who did not participate in the study, reviewed it for suitability of inclusion.

## 3. Discussion

### 3.1. Epidemiology

#### 3.1.1. Prevention

Screening with low-dose computed tomography (LDCT) and cessation of smoking are the only reliable methods that have repeatedly been proven to be effective, in multiple randomized controlled trials, for lung cancer prevention [3]. Smoking history, along with other epidemiological factors such as gender, age, chronic obstructive pulmonary disease (COPD), cholesterol and albumin globulin ratio, have been related, through AI models, with the early detection of lung cancer, the existence of certain mutations in lung tumors, and the efficacy of certain treatments [17,18,19]. In a Chinese study, age and smoking history were used to improve an AI model of lung cancer prediction based on radiological features alone [20], while smoking-related molecular abnormalities of the cells in the buccal mucosa, known as “field carcinogenesis”, were used for the detection of lung cancer in early stages using an optical spectroscopic statistical nanosensing AI technique [21]. Moreover, an AI screening program for lung cancer with LDCT has been designed to evaluate the outcome of smoking cessation, the adherence to and psychological consequences of screening in female patients, the cost of screening, and the detection of coronary artery disease (CAD), COPD, and osteoporosis [22].

However, apart from those factors, other epidemiological factors that are not strictly health-related have also been implicated in the prevention of lung cancer. A consensus and engagement AI study from South Africa suggested that poor patient outcomes in low- and middle-income countries are related with healthcare system constraints at multiple levels, which act as barriers to prevention and disease recognition. More particularly, from the patient perspective, a lack of knowledge and awareness regarding lung cancer and its symptoms and signs and the apparent stigma associated with the disease and its risk factors contribute critically to delaying the search for healthcare assistance; meanwhile, from the healthcare system’s perspective, inadequate knowledge of nurses and doctors in primary healthcare delays the up-referral of patients with suspected lung cancer, whilst the insufficient availability of specialized diagnostic resources at the tertiary level delays definite diagnoses [23]. Even in high-income countries, there remain obstacles related to striking geographic, racial, and socio-economic disparities, especially in those populations at greatest risk of lung cancer—who are therefore most likely to benefit from screening—leading to significantly lower adherence to screening programs than that reported in clinical trials, potentially reducing the realized benefit. Therefore, improved eligibility criteria that more closely match with the full spectrum of persons at risk, taking into account a socio-ecological framework of healthcare, regardless of smoking history, are required [24]. Another study has shown that such epidemiological factors can significantly improve the efficiency of AI models in lung cancer screening [25], while another study has shown that automated data abstraction through an AI model is highly accurate and faster than manual abstraction, considering data related to lung cancer [26]. Consequently, such socio-economic criteria are now being taken into consideration when policies for lung cancer prevention are developed with the assistance of AI [27,28,29]. Table 1 lists the studies that relate to the sector of prevention.

#### 3.1.2. Screening

A revolution in lung cancer screening has taken place over the past decade, after the publication of two fundamental studies showing excellent results in terms of survival through the implementation of LDCT on screening programs in certain population groups at increased risk for the development of lung cancer [30,31]. Earlier prospective studies have continually been adapted to novel technological advances [32], further promoting lung cancer screening [4].

In the present decade, the pioneering technological advance in lung cancer screening is the implementation of AI programs in LDCT and other screening modalities. Numerous studies have been published over the last five years, with the majority of them demonstrating excellent results in terms of the detection of lung cancer or distinguishing between benign and malignant lung nodules [33,34,35,36,37,38,39,40,41,42,43,44,45,46,47,48,49]. More particularly, these studies—which have included tens of thousands of patients with lung cancer and more than a million controls—have shown an accuracy of near or over 90% when using AI models, approaching 100% in some cases, when used as a second reader, which was greater than that of the radiologists, diminishing both the number of lost cases (due to false negative results) and the number of unnecessary follow-ups (due to false positive ones) [33,34,35,36,37,38,39,40,41,42,43,44,45,46,47,48,49]. Apart from solely LDCT, this method was combined with other modalities in some studies, such as circulating molecular biomarkers (liquid biopsy) and autofluorescence bronchoscopy, also presenting excellent results [20,50]. Nevertheless, in another study, despite the fact that an AI-based vessel suppression algorithm also showed positive results, the AI-based automatic detection algorithm presented an accuracy lower than that of radiologists [51]. In addition, the percentage of low attenuation areas led to failure to predict lung cancer incidence in one study [52], while the size and morphology—rather than the internal texture and background parenchyma—played a key role in AI prediction in another study [53]. Moreover, in five studies, AI failed to exhibit accuracy comparable to that of experienced radiologists [54,55,56], at least at lower radiation levels [57] or as far as the clinical outcome [58], raising questions about the universal supremacy of AI-based algorithms.

A controversial aspect of AI-based LDCT screening programs is that they question the usefulness of the radiologists; however, many studies have suggested that AI should play an auxiliary, rather than a leading role in lung cancer screening [22,56,59]. Nevertheless, an aspect of lung cancer screening that could possibly be improved by AI is the dose of radiation received by the examinees. Many studies have suggested that AI-based algorithms are able to successfully detect malignant lung tumors even with ultra-low doses of radiation, either in CTs [47,60,61,62,63] or in PET CTs [64]; however, in one study, the lower dose of radiation affected the AI model’s accuracy negatively [57]. Other AI-related studies have focused on the interpretation of radiologists via AI programs and vice versa [65,66,67,68], the statistical and further considerations of performing AI-based studies [69,70], and even patient data protection [71]. Finally, another aspect of AI implementation in LDCT screening programs is cost-effectiveness, which has been shown to be optimized [29,39,72] either through reducing the number of false negatives (thus decreasing the costs for additional future treatments) or false positives (thus decreasing the costs for unnecessary follow-ups).

Speaking of cost, AI-based LDCT screening programs might seem expensive; particularly for low-income countries. Thus, other more economic programs have been proposed. AI-based algorithms can enhance the ability to detect pulmonary nodules which might represent lung tumors in a curable stage [73,74,75]—an ability that has been accelerated during the COVID-19 pandemic and can be extremely useful especially in low- and middle-income countries [5]. Epidemiological data, such as age, gender, smoking status, family history of cancer, emphysema metrics, and biomarkers, might be used to enhance the accuracy of AI-based screening programs [25,48,49,52,76]. Such biomarkers include single-cell RNA, DNA methylation analysis of circulating cell-free tumor DNA, abnormal lipids, other serum tumor markers, and even abnormal scleral images from the eye [48,77,78,79,80]. Last, but not least, AI-based screening programs for lung cancer might influence smoking cessation; promote the opportunistic diagnosis of osteopenia and osteoporosis, COPD, and CAD; and provide measurements of left atrium volume index (LAVi) and body composition, thus allowing for the prediction of cardiovascular morbidity and mortality and all-cause mortality in patients with lung cancer, while also decreasing the costs for screening for these diseases [22,62,81,82,83,84,85,86,87].

Taking into consideration these advantages and the pooled sensitivity and specificity for AI-based imaging for lung cancer screening—which were determined as 94.6% and 93.6%, respectively, in a recent meta-analysis [88]—various suggestions for the implementation of AI-based algorithms in lung cancer screening programs have already been made [27,89,90,91]. Table 2 lists the studies that relate to the sector of screening.

### 3.2. Clinical Features

Cough, hemoptysis, chest pain, wheezing, hoarseness, shortness of breath, and weight loss are the most common symptoms and signs of lung cancer and, along with other clinical features such as age, gender, and smoking history, might aid in the diagnosis, categorization, management, and prognosis of the disease. However, in contrast with other cancer types, the signs and symptoms of lung cancer might be initially missed or disregarded [23,92], or may only appear in an advanced stage, when the disease has already spread and the treatment has low success rates [93,94,95,96]. In order to address this issue, AI programs could aid clinicians to manage a huge amount of information related to clinical features of the disease that originate from the patients’ electronic medical records, allowing for earlier identification of those who are at greater risk [97]. To that end, clinical features such as age, gender, smoking status and history, drinking status, history of lung infection, chest tightness or pain, expectoration, bloody sputum, cough, hemoptysis, fever, sweating, and family history of tumor in general or lung cancer specifically have been used in the creation of AI-based programs for early lung cancer detection [20,48].

Apart from early detection, AI-based programs that use information related to the clinical features of lung cancer have been used successfully for the prediction of certain, important, disease-specific characteristics. Age, gender, smoking history, pathology, and staging have been related with programmed death-ligand 1 (PD-L1), CD8, and CD45RO^+^ expression in the tumor immune microenvironment and tumor-infiltrating lymphocytes (TILs) [98,99,100,101], while dyspnea, cough, and hypoxia have been used to distinguish between checkpoint inhibitor pneumonitis (CIP) and lung cancer progression [102]. Such clinical features, along with cholesterol levels and the albumin to globulin ratio, have also been used for the prediction of epidermal growth factor receptor (EGFR) mutation [18,101,103], while age, burr, and pleural effusion were indicative of anaplastic lymphoma kinase (ALK) rearrangement status in another AI-based prediction model [104].

Finally, various clinical features have also been used in AI models to improve the management and prognosis of lung cancer. An AI-based program has been developed to educate patients with lung cancer on how they can manage their symptoms [105], while another is under development for the psychiatric assessment of such patients [106]. Furthermore, numerous clinical features have been used in AI models for prediction of the response to chemotherapy [107], radiotherapy [14], and various prognostic indices, such as overall survival (OS), disease-free survival (DFS), and progression-free survival (PFS), for both non-small cell lung cancer (NSCLC) [100,108] and small cell lung cancer (SCLC) [109]. Table 3 lists the studies that relate to the sector of clinical features.

### 3.3. Diagnosis

#### 3.3.1. X-Rays

Some of the first applications of AI in the management of lung cancer concerned the early detection of lung nodules in chest X-rays [110,111,112,113,114]. Although AI-based chest X-ray programs are not as efficient as chest CT ones [6], they have been shown to detect regions suspicious for malignancy in chest X-ray in cases that the human eye could not [92]. Due to the increased performance of chest X-rays during the coronavirus disease 2019 (COVID-19) pandemic, AI-based programs for early diagnosis of lung cancer through incidental nodule detection in chest X-rays in low- and middle-income countries have been accelerated [5]. Incidental identification of lung nodules (which represent resectable lung cancer lesions) in chest X-rays through an AI program might be as high as 17.3% for tumors with total and invasive sizes of the part-solid nodules between 2.3–3.3 cm and 0.75–2.20 cm, respectively [115].

X-ray velocimetry, an AI-based method which integrates 3-dimentional (3D) X-ray (fluoroscopic) imaging with unique image processing to produce quantitative 4-dimensional (4D) measurements of lung tissue motion, as well as accurate calculations of lung ventilation [116] and bone suppression of up to 89.2% through AI [75,117], have been used in the context of chest X-ray images, thus enhancing the early detection of lung cancer through chest X-rays. Furthermore, various chest X-ray features have also been used in deep learning programs for the early detection of lung cancer. More specifically, atelectasis, pleural effusion, mass, nodule, and “no finding” (as control) have been characterized as features that are useful in AI-based detection models, while cardiomegaly, consolidation, edema, emphysema, fibrosis, hernia, infiltration, pleural thickening, pneumonia, and pneumothorax have been characterized as rarely related with lung cancer [112,118,119]. With sensitivity, specificity, positive and negative predictive values (PPV and NPV, respectively), and area under the curve (AUC) varying between 60–90%, 60–97%, 80–95%, 89–91%, and 0.740–0.884, respectively, AI-based lung cancer detection algorithms utilizing chest X-rays have proven their value [38,112,119,120,121,122]. However, in other large-scale real-world settings, such algorithms underperformed compared to radiologists, did not enhance their accuracy, or even drove radiologists to make incorrect follow-up decisions while they were correct without AI, suggesting that AI-based programs are feasible only in sorting and removing normal chest X-rays with a tailored cut-off, in order to increase efficiency and reduce the workload of radiologists [74,123,124,125].

Finally, it has been shown that AI-based algorithms for the detection of lung cancer through X-rays might also have other useful applications, such as in pediatric pulmonary oncology [126], identifying suitable patients with lung cancer for “CyberKnife” robotic radiosurgery [127], or identifying a patient’s race for medical purposes [128]. Table 4 lists the studies that relate to the sector of X-rays.

#### 3.3.2. Chest Computed Tomography (Chest CT)

Chest CT remains the cornerstone in lung cancer diagnostics [4,6,30,31,32]. AI has been implemented in the context of chest CT, utilizing various nodule morphologic characteristics [129,130,131,132] through different novel techniques [6,110,116,117,130,133,134,135,136,137], mainly in terms of automated nodule detection [4,5,89,110,112,116,130,135,138,139,140,141,142,143,144,145,146,147], as well as discrimination between benign and malignant nodules [89,90,91,110,129,130,138,139,140,141,142,143,144,145,146,147,148,149,150], histological and molecular categorization of malignant tumors [110,129,135,151,152], staging [135,153,154] and treatment planning [110,131,135,140], response prediction [135,140,153], adverse events [155], and prognosis [110,116,135,153], although various additional uses have also been described [4,6,29,90,126,130,141,156,157,158]. The need for implementation of LDCT in large-scale screening programs, in order to decrease lung cancer-related morbidity and mortality, has been made much easier through the use of AI-based algorithms [4,6,24,27,29,44,89,90,118,156,158,159,160,161,162]. Moreover, the vast number of applications of chest CT in lung cancer diagnostics has led to the necessity for creating much more sophisticated screening criteria, also with the use of AI [4,6,24,27,28,29,89,90,130,141,158,161,163].

The majority of research on the implementation of AI-based programs to chest CT in the context of lung cancer has focused on automated nodule detection, the malignancy prediction risk, or the combination of these. AI-based programs for automated nodule detection presented positive screening results in about 19.5% of cases, with solid nodule and multiple nodule cases being about 53.6% and 26.8% of all the positive results, respectively, while the detection was made in an early stage (0–1) in about 91.4% of the cases [50]. Such algorithms for automated nodule detection present sensitivity, specificity, false positive rate per scan, PPV, NPV, and AUC between 0.592–1.000, 0.708–0.991, 0.067–25.000, 0.556, 0.882, and 0.880–0.942 respectively [33,36,41,42,44,56,57,87,164,165,166,167], depending on factors such as nodule size, ground glass composition, and recency of the program [36,41,56,57,165,166,168]. Such AI-based programs might aid in the detection of lung nodules when radiologists are not able to do so [169]. For the discrimination between benign and malignant lung nodules in chest CT, the respective AI models present sensitivity, specificity, false positive and false negative rates, PPV, NPV, and AUC between 0.408–0.990, 0.393–0.973, 0.026–0.245, 0.004–0.248, 0.789–0.976, 0.582–0.973, and 0.640–0.970, respectively [17,39,43,45,46,48,49,53,54,55,59,78,170,171,172,173,174,175,176,177,178,179,180,181,182,183,184,185,186]; figures better than those of the human readers in most cases [17,39,43,46,53,54,55,59,182,184,186,187,188], and more timely as well [46,172]. However, in some cases, humans performed better than AI [54,172,176,189]. The obtained figures were also largely dependent on nodule size, ground glass composition, and nodule volume change [49,54,178,184].

In addition, certain AI-based models are able to perform both automated nodule detection and malignancy discrimination in a single chest CT with high accuracy rates [34,40,95,137,160,190,191,192,193,194]. In three recently published systematic reviews with meta-analyses, the AUC for lung nodule detection with AI assistance in chest CT was 0.937 [112], the pooled sensitivity and specificity for malignancy classification of lung nodules were 0.90 and 0.89, respectively, with a pooled diagnostic odds ratio (DOR) at 70.33 [150]; while, for lung cancer diagnosis, the sensitivity and specificity were 0.87, missed diagnosis rate and misdiagnosis rate were 13%, positive likelihood ratio was 6.5, negative likelihood ratio was 0.15, diagnostic ratio was 43, and the sum of area under the combined subject operating characteristic curve was 0.93 [144].

Another important application of AI in lung cancer diagnostics is the prediction of cancer histologic types and sub-types, staging, and molecular and immunological classification. In the discrimination between SCLC, adenocarcinoma, and squamous cell lung carcinoma, AI programs have achieved various results, with an accuracy of 57.7% [195], sensitivity and specificity of 0.90 and 0.44, respectively [194], and AUC between 0.87 and 0.97 [196,197]. Moreover, in the differentiation between the sub-types of lung adenocarcinoma (i.e., atypical adenomatous hyperplasia/adenocarcinoma in situ, minimally invasive adenocarcinoma, and invasive adenocarcinoma), an AI model achieved AUC values between 0.779 and 0.946 [198,199,200]. In a case report, intrapulmonary metastatic carcinomatosis was under-detected [201]; however, in two AI studies, thoracic lymph node invasion was successfully identified in chest CTs, with accuracy and AUC of over 0.850 [202,203] while, in another one, it dropped to 0.761 for an optimal cut-off value of −103 Hounsfield units (HU) [204]. As far as distant metastases, two studies have reported results for visceral pleura and bones [205,206]. For molecular classification, the main emphasis has been given in the prediction of EGFR mutation, with reasonable AUC values [103,207,208,209,210], while attention has also been paid to the Kirsten rat sarcoma (KRAS) viral oncogene [209,210] and ALK rearrangement [104]. As for immunological status, CD8 expression prediction was achieved with an AUC of 0.861 [98] while, for PD-L1 expression, the corresponding AUC values were over 0.930 regardless of the percentage of expression [100]. Finally, another AI model predicted the mutation status for 8 genes in addition to the expression of PD-L1, with AUC values of 0.862 and 0.856, respectively [211].

The importance of prediction of the molecular and immunological status of lung carcinomas lies in the favorable response to novel treatments that have been developed depending on this status. Response to immune checkpoint inhibitors (ICIs) could be predicted successfully in four occasions with acceptable accuracy [212,213,214,215], and both response to tyrosine kinase inhibitors (TKIs) and ICIs was predicted in another study [216], while resistance to TKIs was predicted in another study, which was associated with multiple genotypes and gene pathways [207]. In addition to ICIs and TKIs, treatment response could also be predicted for certain chemotherapy regimens using an AI-based algorithm, based on chest CT features [15] while, for demanding surgical interventions, treatment planning could also be assisted [127,217]. Furthermore, radiotherapy, planning [218], treatment response [219], and prediction of adverse events [220] could also be enhanced. All of this information is particularly useful for the prognosis of lung tumors. The response to radiotherapy predicted via AI-enhanced chest CT interpretation could predict local recurrence-free survival (RFS), DFS, and OS [219]. The response to ICIs and TKIs was also associated with OS, either alone [100,213,216,221] or in combination with other chest CT features (e.g., TILs) [222] or biomarkers (e.g., gut microbial community signatures) [212]. Regardless of the treatment response in lung cancer, chest CT-based AI-valuated radiomic features could predict prognosis, either alone [223,224] or in combination with gene expression [196], histopathological features [205], clinical features [108], or clinical stage [225,226,227,228]. Finally, through chest CT, AI software can predict all-cause mortality in lung cancer based on coronary artery calcification [52,85,86], quantification of emphysema [52], skeletal muscle mass [229], and body composition measurements [84].

In order to perform all the aforementioned applications, AI-based software can use certain morphological characteristics from lung tumors in chest CTs. Radiomics is the part of radiology which utilizes AI-based, lung cancer image analysis that, beyond the classic morphological characteristics such as lesion margin, size, volume, and densitometric features, quantifies the intralesional heterogeneity in order to construe the image phenotype and extract genomic and proteomic information [129,130]. The two main features that affect automated nodule detection using AI are size and density. AI-based programs can accurately determine the maximal diameter of the lesions [230], and tend to detect easier and more consistent lesions of smaller diameter and ground glass opacity [36,41]—features that, along with central localization, cause missed diagnoses by humans more often [164]; although ground glass opacity lesion detection via AI might be affected by radiation dosage [57]. Other features, such as higher volume, pure ground glass opacity, sub-solid (rather than solid) status, lower average CT number in HU in general, upper lobe localization, malignant sign at the nodule edge, and nodule count, are also suggestive of malignancy [20,56]. Furthermore, other non-nodule-related findings, such as lymphadenopathies and mediastinal and vessel invasion, cavitations, and pleural effusions, should also raise the suspicion of malignancy instead of infection, as other features such as the number of lesions, density, and even crazy paving pattern might be misleading [55,131]. The number of the lesions and their density might also guide the discrimination between benign and malignant nodules [217]. However, this discrimination might become extremely difficult; thus, numerous radiomic features, which are more easily determined by AI-based programs, have been used. Three-dimensional texture features of heterogeneity in gray level configuration, such as inverse difference moment, correlation, sum entropy, and sum variance, along with shape features such as size, area, perimeter, eccentricity, extend, compactness, radial distance, roughness, elongation convexity, equivalent diameter, and sphericity, are examples of such features [181].

Nevertheless, imaging features akin to malignant growths might also be found in benign ones. Such features include small size, average CT value in HU, solid ratio, burr sign, lobulated sign, pleural indentation, smooth edges, and cavity [178,189], although kurtosis, skewness, or the ratio of length to short diameter are not [178]. In any case, the natural growth history of a nodule in serial chest CTs remains a valuable feature to determine its possible malignancy, both for radiologists and AI-based programs, especially in cases of sub-solid nodules [132]. Apart from nodule-related features, parenchymatic ones might also provide important information for determination of the nature of a nodule. The quantification of emphysema metric [49] and parenchymatic measurements—such as the lung tissue including HU histogram mean, standard deviation, and skew; total, tissue, and air volumes, across the whole lung or isolated from the lobe containing the pulmonary nodule; and measurements from the segmental bronchi, across the whole lung or limited only to the bronchi in the lung lobe containing the pulmonary nodule (e.g., wall thickness, diameters, perimeters, areas, and wall area fraction)—have also been used [17]. It seems that the inclusion of parenchymatic features along with nodular ones improved the performance of a model that discriminates benign from malignant nodules, while ablating the nodule from the parenchyma texture worsened its performance [53,179].

As far as histological type, CT-derived kurtosis seems to differ significantly between adenocarcinoma and squamous cell carcinoma [196], while the diameters of each lesion and the presence or absence of the chest wall seems to help in differentiating between the two aforementioned types and SCLC [195]. Among the different types of early-stage adenocarcinoma, 2D mean diameter, 3D mean diameter, mean CT value, maximum CT value, volume of ground glass nodule, and invasiveness seem to increase gradually from atypical adenomatous hyperplasia/adenocarcinoma in situ, to minimally invasive adenocarcinoma through to invasive adenocarcinoma groups [198,199], while the density, volume, shape, vacuolar signs, air bronchogram, lobulation, spiculation, pleural indentation/retraction sign, vascular convergence signs, and malignant probability also differed significantly [198,199]. Furthermore, clinical stage IA3, solid-part size, and average solid CT value were independently associated with pathological lymph node metastasis in a multivariate analysis of 39 imaging features [204], while such characteristics also predicted visceral pleural invasion and lymphovascular invasion in another study [205]. In terms of EGFR mutation, vascular convergence sign was significantly more prominent in tumors with this mutation, while pleural effusion, pericardial effusion, pleural thickening, and other signs in tumor imaging were not [103]. As far as treatment-related adverse events, consolidations or ground-glass opacities next to radiotherapy fields or pneumonitis arising during or after chemo-immuno-therapy should always raise the suspicion of an adverse event related to treatment [131]. Prediction of response to ICIs seems to be related more to morphological nodal changes in the mediastinum, supraclavicular, and hilar regions; lung and bone metastases; and pleural effusions, atelectasis, and consolidations after visual analysis of AI-generated prognostic heatmaps, rather than to other radiological features [221]. Finally, in terms of prognosis, solid-part size, CT value standard deviation, solid-part volume ratio, and bronchus translucency were associated with early recurrence [227]. CT-based composite score predicted OS in early-stage lung adenocarcinomas [205], while tumor size and heterogeneity were among the strongest predictors of survival in another study [223]. Moreover, solid nodules seem to carry a worse prognosis [225] while, for tumors with non-solid parts, solid-part volume rather than size predicted survival more accurately [226].

In order to detect all of the the aforementioned radiomic features, AI-based programs implement a particularly sophisticated image analysis using various approaches, such as distributed learning, statistical learning, computer-aided diagnosis and detection systems, fully automated image analysis tools, and natural language processing, such that they can perform detection, contouring and segmentation, prediction of histology and tumor stage, prediction of mutational status and targets for molecular therapies, prediction of treatment response, and outcome [135]. Segmentation describes the procedure of drawing the margins of an anatomical structure or a lesion, and is used widely in AI-based image analyses of chest CT in lung tumors. Numerous 3D algorithms have been used for the segmentation of both lungs [55,190,231], of each lung separately [231], of lung lobes [232], and of anatomical structures, such as lung segments [233], segmental pulmonary arteries and lobular pulmonary veins [234], and airways [235]. After segmentation, various techniques, such as bone or vessel suppression, are used in order to optimize the region of interest (ROI) in lung parenchyma [51,117]. Apart from segmentation of the lungs, the same procedure could also be followed for tumors. In such cases, after segmentation, there are two courses of action: (a) assessment of the lesion size, volume, and densitometric features; and (b) radiomic features extraction to characterize the whole abnormalities providing the so-called “virtual biopsy” [130,134]. Automated tumor segmentation in mice via AI models using micro-CTs has been achieved, along with soft tissue and tumor volumetry, as well as blood vessel identification and discrimination from the tumor [236,237]. Nodule segmentation with consequent localization and size measurement of the imaging features inside the tumor has also been achieved in humans [110,160,238].

In cases of poor image quality, AI programs can improve the images using various quality enhancement methods. This can be achieved by focusing on the ROI and subsequently augmenting the lower-dimension features or obtaining the nodule’s cross-sections from multiple view angles and encoding its volumetric information into a compact representation through aggregating information from its different cross-sections via a view pooling layer, thus achieving an improved ability for small object detection [44,67,183]. Other methods for image quality improvement that have been used include data augmentation using fictitious images [239], reducing the noise in images using a weighted filter [240], and reducing the radiation dosage and subsequently reconstructing the image using a reconstruction algorithm [57]. After tumor segmentation and image quality enhancement—not necessarily in this order—the next step is volumetric quantification of the tumor and the isolation of its separate components [6,47,226]. In addition, a brand-new method—namely, X-ray velocimetry—integrates X-ray imaging with a unique image processing approach to produce quantitative 4D measurements of lung tissue motion, allowing for accurate calculations of lung ventilation over time [116,133].

Apart from the aforementioned applications, AI-based programs which use chest CTs can provide additional utilities. One of these would be the significant reduction of radiation dosage for screening programs. A drastic dosage reduction of 15% to 76% did not affect the diagnostic accuracy for solid or sub-solid nodules [47,60,62,241]; however, a 10- to 20-fold dose reduction affected the diagnostic accuracy for ground glass nodules [57]. Nonetheless, in other studies, a reduction in radiation in values below 1 mSv—and, more particularly, between 0.07–0.14 mSv—did not affect the diagnostic accuracy, neither for pulmonary lesions nor mediastinal lymph nodes [61,63]. Another useful application is the screening or evaluation of other conditions or diseases. More particularly, AI-interpreted chest CT for lung cancer screening might be useful in pediatric oncology [126,242], in the identification of a patient’s race for medical purposes [128], in chronic kidney disease [192], osteoporosis [81], aortic aneurysms [60], left atrium volume evaluation [83], CAD [62,85,86,87], body composition determination [84], or in multiple diseases, including smoking cessation, COPD, and emphysema [22,52,82]. The majority of all these diseases (with the exception of osteoporosis) might also affect significantly survival in patients with lung cancer [22,52,62,82,83,84,85,86,87]. Last, but not least, as prevention is more efficient than treatment, and as screening for multiple diseases with one examination is like “hitting two birds with one stone,” it is easy to understand why the implementation of AI in chest CT is also cost-efficient [39,72]. Table 5 lists the studies that relate to the sector of chest CT.

#### 3.3.3. Positron Emission Tomography (PET)

PET is a relatively recent modality, compared to chest CT; however, its potential has led many researchers to implement AI-based algorithms in the context of PET. As a consequence, almost every application of chest CT in lung cancer diagnostics has also been able to be executed with PET—either alone or (more commonly) in combination with chest CT, leading to better results in most cases. Furthermore, PET has also enabled novel pathways that were not easily accessible with chest CT. Using similar techniques such as segmentation [129,133,135,243,244,245,246,247,248,249,250,251] or “virtual biopsy” [129,243,247], PET is able to assist in lung cancer diagnostics through automated nodule detection [7,114,133,135,140,153,244,246,249], discrimination between benign and malignant nodules [114,133,140,153,244,246,247], histology prediction [7,114,129,133,135,247,249], genotyping [129,133,135,152,153,243,249,250,251], staging [7,114,133,135,153,244,246,247,249], treatment planning [133,135,140,152,153,243,248], response [7,114,133,135,140,153,243,245,246,247,248,249,250], adverse events [133], and outcome [7,129,135,153,243,245,246,247,249,250] analyses. In addition, PET can also be used for minimally invasive endoscopic biopsy [153].

As with chest CT, the cornerstone of PET image processing and subsequent interpretation by AI programs is segmentation. This procedure is usually performed automatically (i.e., by the AI-based program) and then compared with segmentation performed manually by one or more trained nuclear radiologists [224,252,253,254,255,256,257]. This comparison is evaluated according to objective metrics, such as the Dice score—a figure of merit that measures the spatial overlap between the estimated segmentation and a reference standard (e.g., manual segmentation) [257]. Although such scores present high reliability as measures of consistency in static examinations such as chest CT, this does not seem to apply to the same degree for PET [257]. The reason for this probably lies to the fact that, in contrast with chest CT—in which mainly static indices related to anatomic structures, such as size, volume and density are evaluated—in PET, apart from these static indices, dynamic ones (e.g., related to glucose metabolism) are also taken into consideration. Such indices include total lesion glycolysis (TLG), metabolic tumor volume (MTV), and standardized uptake value (SUV) [107,252,254,255,256,257,258,259,260]. Of course, static parameters such as gross tumor volume, diameter, and skewness are also evaluated in PET [259,260,261].

As PET and chest CT can be combined into a multi-modal examination—namely, PET/CT—multiple AI algorithms, such as the maximum relevance minimum redundancy (mRMR), which selects the features with high correlation to the pathological results while retaining the features with minimum correlation between them, and the least absolute shrinkage and selection operator (LASSO), a statistical formula whose main purpose is the feature selection and regularization of data models through shrinking the regression coefficients [104], can be implemented. Furthermore, the validation of AI implementations in PET/CT can be performed using different methods, such as inter-observer and intra-observer benchmarking, primary validation, functional validation, end-user testing [238], and standard benchmark image utilization [262], and such models can be combined in multiple layers, such as early fusion, late fusion, dense fusion, or hyper-dense fusion [263]. Moreover, the fact that PET sweeps the whole body permits the segmentation of multiple anatomic regions, such as the mediastinum, in order to detect regional lymph node invasion in separate nodal groups [258,264,265,266], or literally any other part of the body, which allows for the identification of distant metastases [265,266]. Finally, other applications of AI in PET have allowed for a reduction in radiation dosage of up to 30 times, with values reaching ~0.11 mSv [64], as well as providing an automated examination report and further guidance [267].

Under the aforementioned conditions, PET/CT achieved an accuracy of 97.0%, a sensitivity between 90.0% and 95.9%, a specificity between 94.2% and 98.1%, PPV and NPV at 88% and 100% respectively, and an AUC between 0.970 and 0.989 in the context of lung nodule detection, even at ultra-low radiation dosages [64,255,262]. As far as malignancy discrimination, PET’s sensitivity has been shown to vary between 0.556 and 0.830, and its specificity is between 0.700 and 0.930—depending by the condition of the measurements (per lesion–per patient and low prevalence—high prevalence)—while the AUC varied between 0.743 and 0.900 [173,184,254]. PET radiomics have also been used for the histological prediction of lung cancer, along with chest CT radiomics and gene expression, yielding modest results [196].

As far as immune status and gene expression, a PET/CT radiomics–clinical combined model showed the best AUC (0.932) in predicting CD8 expression [98], while PET alone showed an overall accuracy of greater than 82% in predicting the high proliferation group for MIB-1 (Ki 67) [260]. A combined PET/CT–clinical model achieved an AUC of 0.880 in predicting ALK rearrangement [104] while, for KRAS, the respective AUC reached 0.930 for a solely PET/CT model [209]. Finally, for EGFR mutation prediction, a PET/CT model achieved an AUC of 0.940 [209] while, in a meta-analysis, the use of PET/CT was included among the factors that improved diagnostic performance [251].

Due to its nature, PET is definitely superior in staging compared to other single examinations, such as chest CT. For T lesions, PET/CT achieved a sensitivity of 86.2% with a false positive ratio of 2 per case [266]; for N lesions in one model, the sensitivity, specificity, and accuracy were 0.74 ± 0.32, 0.84 ± 0.16, and 0.80 ± 0.17, respectively [265] and, in two other models, the AUC varied between 0.88 and 0.94 [258,264]; while, for M lesions, the sensitivity, specificity, and accuracy reached 0.45 ± 0.08, 0.79 ± 0.06, and 0.63 ± 0.05, respectively [265]. Furthermore, for complete TNM, the evaluation accuracy varied between 94.3% and 99.1% in three other studies [262,266,268]. PET/CT was also found to be useful in planning the radiation treatment, taking into consideration standard free-breathing [259]. For predicting the response to ICIs or overall progression, the AUC values were 0.69 for “PET-Skewness” and 0.75 for “PET-Median”, respectively [261], while late PET/CT stage and metabolic tumor volume 50% (MTV50) > 4.04 were independent factors of poor survival in patients with NSCLC who received chemotherapy [107]. Finally, there are several PET/CT biomarkers for predicting survival [107,254,261]. PET/CT models seem to perform better compared to solely PET ones in this field [224]. The best radiogenomic rule for predicting the outcome, using gene expression and PET/CT, achieved an AUC of 0.87 [196]. Table 6 shows the studies that relate to the sector of PET.

#### 3.3.4. Biomarkers

Even though LDCT screening programs have improved morbidity and mortality due to lung cancer significantly, it is now evident that additional actions are also necessary. It is well-established that there is a gap in the knowledge regarding the people at risk who should be referred for LDCT screening, both in identifying them as persons at risk as well as the correct timing for their referral. Many researchers have suggested that this gap might close with the use of appropriate biomarkers [4,23,24,27,32,269]. The implementation of widespread biomarker-based screening is only possible with the use of vast datasets such as electronic health records, the interpretation of which is efficient only when utilizing AI [26,270,271,272]. To that end, numerous biomarkers and methods have been considered [93,148,163,273,274], while novel studies are yet to be published [275]. Biosensors, nanotechnology-based diagnostics, wearable devices, mobile health, aptamers, and molecularly imprinted polymers are among the methods that have been used [276], while molecules such as mRNA, microRNAs, long non-coding RNA (lncRNA), airway or nasal RNA signatures, circulating cell-free tumor DNA, methylated DNA, or chromatin structural changes in the buccal mucosa are among the most promising biomarkers, achieving AUC values greater than 0.90 for the early detection of lung cancer [21,79,277,278,279,280]. Apart from genetic material, other molecules, such as multiple serum proteins [281], known tumor biomarkers—such as Pro-gastrin-releasing peptide (ProGRP), vascular endothelial growth factor (VEGF), carcinoembryonic antigen (CEA), CYFRA21-1 (a cytokeratin-19 fragment), and Neuroson-specific enolase (NSE) [48]—urinary concentrations of 4-methoxyphenylacetic acid (4MPLA) [278], autoantibodies, complement fragments, blood protein profiling, exhaled breath biomarkers, metabolomics [280], or even routine blood indices, such as those derived from whole blood counts, creatine phosphokinase (CPK), creatine kinase (CK)-isoenzymes, alkaline phosphatase (ALP), lactate dehydrogenase (LDH), albumin, albumin/globulin ratio, and so on [282], have also been used; although with relatively lower diagnostic yield.

Biomarkers, with the help of AI, might also be used to distinguish between benign and malignant nodules [140,283]; discriminate between lung cancer and hematologic malignancy [284], COPD [285], or tuberculosis [282]; or differentiate between benign and malignant pleural effusion [286], with AUC values greater than 0.80. Furthermore, certain genomic signatures in AI models are correlated with the histologic types or sub-types of lung cancer [196,287,288], which is also a target in upcoming research [275]. In addition, AI models can predict the expression of critical biomarkers for management of the disease such as PD-L1 [289], while biomarkers such as EGFR, ALK, KRAS, PD-L1, and TILs can be considered as a target for detection by radiomic AI models [222,290,291]. Last, but not least, as far as the diagnostic work-up, increased CEA levels have been related with lymphovascular invasion in a pathology review in otherwise clinical T1 stage NSCLC—a fact that worsens the stage and, consequently, the prognosis [292].

Apart from the diagnostic part, biomarkers could also be helpful in the treatment and prognosis of patients with lung cancer. The so-called omics approaches, which involve huge amount of data, obtained (with the help of AI) through the analysis of various molecules (e.g., those related to genes or proteins) via procedures such as next-generation sequencing genomic or mass spectrometry-based proteomic analysis, have been used to identify several biomarkers that could potentially predict treatment response [2,163,270,272,273,288,293,294]. This has also been set as a target for upcoming research [275,295]. The most prominent pharmacological treatments for this purpose are immunotherapy and targeted therapy [296,297,298,299]. The inflamed immune phenotype [300], high PD-L1 expression via AI evaluation [301], and higher number of TILs predicted by CT radiomics [222] have been associated with better response to immunotherapy, while other serum tumor markers, such as CYFRA and CEA, predicted poor response with very high specificity [302]. Biomarkers recognized through radiomic analysis have achieved satisfactory AUC values in predicting treatment response [214,215], although genomic analysis also seems to provide promising results [279,303,304]. Phosphoproteins could also predict the response to several targeted anticancer drugs and their hundreds of combinations [305]; while another, rather unexpected, biomarker that could predict the response to immunotherapy is the intestinal microbiome [212,306]. Furthermore, biomarkers derived from liquid biopsy [307] and nanoparticles functionalized with immune agonists [308] could also help to identify therapeutic targets in the tumor microenvironment.

As for prognosis, it usually comes in parallel with the treatment response, either as PFS or as OS [163,270,272,288,293,294]. This is particularly valid for immunotherapy, in which different biomarkers—such as PD-L1, anti-cytotoxic T-lymphocyte-associated protein 4 (CTLA-4) antibodies, or TILs—have been used to predict the outcome, either through immunohistochemistry or radiomics [213,214,222,296,297,298,300,301]. Patients with overexpression of PD-L1 have worse survival; as such, PD-L1 or programmed cell death 1 (PD-1) inhibitors may substantially improve OS [299]. Apart from immunotherapy-related biomarkers, other biomarkers with predictive value (as far as outcome) include genomic ones [196,304] and gut micro-organisms [212]. More particularly, Bacteroides dorei and Parabacteroides distasonis have been associated with longer OS, while Clostridium perfringens, Enterococcus faecium, Cortinarius davemallochii, Helotiales, Chaetosphaeriales, and Tremellomycetes were associated with shorter OS [212]. Furthermore, Hymenoscyphus immutabilis and Clavulinopsis fusiformis were related with higher PD-L1 expression, whereas Thelephoraceae and Lachnospiraceae bacterium were related with toxicity to immunotherapy [212]. Other widely used biomarkers for prognosis are those found on lung cancer tissue, such as ALK, mesenchymal–epithelial transition (MET), C-ros oncogene 1 (ROS1), or tumor protein p53 (TP53), although with ambiguous yield [290,299]. Nevertheless, other similar biomarkers, such as protein arginine methyltransferase 6 (PRMT6), and KRAS, seem to be correlated with worse outcomes [290,299,309], while EGFR-enriched tumors might be related with a more favorable outcome in patients receiving targeted therapy [290,299]. As far as blood biomarkers, perioperative peripheral venous circulating tumor cells (CTCs), circulating tumor DNA (ctDNA) and CEA, and pulmonary venous CTCs predicted unfavorable outcomes and distant metastases [299]. Finally, various metabolic biomarkers, such as guanine, choline, creatine, leucine, tryptophan, N6,N6,N6-Trimethyl-L-lysine, L-pyrogluatmic acid, benzoic acid, cystine, methionine sulfoxide, histamine, itaconic acid, pyruvate, and malonic acid have also been associated with different outcomes [310].

In addition to the above, there is also the intention for biomarker-based AI algorithms to be used in other aspects of lung cancer management, such as personalized predictive models, feedback data loops for preventive healthcare strategies, individualized treatments, toxicity prevention (or, at least, minimization), quality of life improvement, efficient resource allocation, and cost-effectiveness [275,295]. Table 7 summarizes the biomarkers that have been used in the implementation of AI approaches for lung cancer diagnosis, and Table 8 lists the studies that relate to the sector of biomarkers.

#### 3.3.5. Biopsy

The identification of tumor cells through cytological or pathological exams in tissues plays a significant role in the management of lung cancer. AI has been implemented in the context of bronchoscopy, biopsy acquisition, and sputum cytology [93,311]. Nonetheless, the main field of research includes histopathological images, and digital pathology results from the implementation of AI in tissue slides [293,312,313,314]. AI not only can improve the diagnostic efficiency when using cytological or histopathological images [2,315,316,317,318,319,320,321], but can also enhance the efficacy of analysis of the tumor histological sub-type, microenvironment, and genetic signature [2,320,321,322]. In a meta-analysis, the pooled AUC for diagnosing lung cancer through AI in histopathological specimens was calculated as 0.90, with a pooled sensitivity and specificity of 0.93 and 0.68, respectively [323]. Furthermore, the implementation of AI in histopathology—sometimes combined with other modalities, such as radiomics—could predict the response to precision treatments, such as immunotherapy and targeted therapy [296,314,324,325,326,327]. In addition, it can also distinguish CIP from disease progression [102] and, combined with liquid biopsy, could play a key role in the early diagnosis of lung cancer in the near-future [132,276,283,328,329,330].

In the struggle to obtain tissue for the diagnosis of lung cancer, AI has been implemented in fiberoptic bronchoscopy, autofluorescence bronchoscopy, and a lung airway navigation systems using a fiber-Bragg shape-sensing catheter [50,331,332]. Sonographic images from EBUS have also been used along with AI for differentiating between benign and malignant tumors and predicting nodal metastasis, with an AUC of 0.88 and an accuracy as high as 82% [9,333,334].

The implementation of AI in cytological specimens containing lung cancer cells is a novel technique [335,336], which has yielded an accuracy and precision of 0.945 and 0.991, respectively, in the diagnosis of lung cancer [337]; while its accuracy in differentiating between lung cancer sub-types varies between 66.8% and 76.8% [338]. In histopathology, the goal of AI implementation is the identification, classification, and characterization of tumors [275,339]. AI makes pathological image classification more accurate, transparent, and convincing [340]. The sensitivity, specificity, accuracy, precision, and AUC of AI in lung cancer histopathological diagnosis, in the majority of studies, exceeded 94%, reaching figures close to or at 100% in most cases [341,342,343,344,345,346,347,348,349,350]. AI can also help in the differentiation between multiple primary lung cancer and intrapulmonary metastasis [351], as well as between benign and malignant regions in paraffin-embedded tissue blocks of pulmonary adenocarcinoma using micro-CT [352]. Furthermore, AI can detect the progressive increase in atypical epithelial cells and the progressive decrease in lymphocytic cells along with the progressively increasing cellular intratumor heterogeneity—from normal to atypical adenomatous hyperplasia, to adenocarcinoma in situ, to minimally invasive adenocarcinoma, and eventually to invasive adenocarcinoma—consistent with the results obtained through tissue-consuming and expensive molecular/immune profiling [353]. AI can also discriminate between lepidic, papillary, acinar, solid, micropapillary, and cribriform arrangements of invasive non-mucinous lung adenocarcinoma, with an AUC greater than 0.8 [354].

Apart from histopathological types and sub-types, AI can also determine the PD-L1 status of the tumor with an accuracy and specificity of over 95% and an AUC of 0.8 [289,301,355], while the concordance between pathologists when scoring the PD-L1 status of a tumor with the assistance of AI was increased by between 4% and 9%, when compared to scoring without AI assistance [291,356,357]; although these results mainly concern tumors with cutoff points in PD-L1 expression of up to 49% and worsen when PD-L1 expression is 50% or higher [358]. Other immunohistochemical tumor markers, such as CD45RO⁺ and PRMT6, can also be detected by AI [99,309]. Furthermore, AI natural language processors can be used for the interpretation of histopathological reports [359]. AI can also be used in the genetic context of lung cancer. Genetic sequence abnormalities of circulating blood cells detected by AI can be used for discrimination between benign and malignant lung nodules, histology determination, cancer phase differentiation, structural metabolic behavior pattern detection, and determining signaling property, obtaining AUCs greater than 0.8, especially when combined with chest CT radiomics [78,360]. Besides abnormal circulating leukocytes, the expression levels of TP63, erythropoietin-producing hepatocellular carcinoma receptor A10 (EPHA10), fibrillin-2 (FBN2), and interleukin 1 receptor accessory protein (IL1RAP) genes in lung tumor cells are predictive of the histological type [196]. Furthermore, AI can predict certain tumor genetic characteristics from histopathological or cytological images that are important for both treatment and prognosis [361]. AI can predict the EGFR mutation status of a lung tumor (AUC between 0.883 and 0.897 in histopathological images and 0.950 in cytological images), distinguish between common and uncommon EGFR mutations that guide TKI selection (accuracy between 0.830 and 0.930 in histopathological images), predict KRAS mutation status (AUC at 0.950 in cytological images), and predict ALK rearrangement status (AUC between 0.921 and 0.995 in histopathological images and 0.750 in cytological images) [337,362].

In recent years, both cytology and histopathology have been boosted by novel techniques that provide a clearer image of the tumor microenvironment and improve lung cancer diagnostics through the implementation of AI. Imaging mass cytometry has enabled the spatial analysis of immune lineages and activation states, through AI with distinct clinical correlates, using a single 1 mm^2^ tumor core [363]. Digital pathology—namely, the digitization of whole-slide images of tissues—enabled 3D pathology, while the application of image analysis algorithms (including random single-pixel noise in the correct frequency) in combination with multiplex staining, multiplex bioimaging technology, and multimodal data analysis catalyzed the automated analysis of tissue slides via AI, leading to a comprehensive and in-depth quantitative and spatial analysis and characterization of the broader tumor microenvironment, including the distribution, function, and interactions of distinct immune cells [314,326,327,364,365]. Through this technique, various characteristics of the tumor microenvironment, such as TILs and PD-L1 expression, have been accurately determined [366,367]. On the other hand, the dynamic tumor–stroma interactions that continuously alter the tumor microenvironment, as well as the invasive nature of tissue-based biomarkers, along with their lack of holistic representation of the evolving disease in a real-time manner, can result in sub-optimal treatment decisions. Liquid biopsy-based biomarkers could monitor intercellular communication within the tumor microenvironment, which occurs through the exchange of extracellular vesicles. These vesicles transfer a plethora of biomolecules that mediate intercellular cross-talk, shape the tumor microenvironment, and modify drug responses. They could be obtained in a minimally invasive manner, over the course of the disease, through the use of innovative approaches such as microfluidics, magnetic beads, and aptamers [307]. Technological advances in mass spectrometry-based proteomics enable the pinpointing of abnormal protein expressions in these vesicles—a procedure that enhances the understanding of lung cancer’s intricate biological milieu, identifies novel treatment targets, allows for real-time treatment monitoring, and detects early resistance [272]. Diagnostic models that include liquid biopsy and AI analysis have been shown to produce the optimal diagnostic yield [20]. MicroRNAs, which can be obtained through liquid biopsy and analyzed through AI, might be indicative of lung tumors decades before their clinical appearance [368], while multimodal detection of PD-L1 (e.g., through liquid biopsy specimens of PD-L1 protein, PD-L1 mRNA, and circulating PD-L1) might provide a more comprehensive tumor profile and could help to determine those patients who are more suitable for immunotherapy [298]; although the specimen processing technique could limit the accuracy of the method [335]. Table 9 lists the studies that relate to the sector of biopsy.

#### 3.3.6. Staging

Lung cancer stage at diagnosis is one of the strongest determinants of prognosis, as stage 1A lung cancer has a >75% 5-year survival rate, compared to <5% at stage 4 [6]. The high mortality rates of lung cancer are due to its usual diagnosis at a late stage [113]; although this has begun to change due to LDCT screening programs [149]. AI has been implemented in various imaging modalities, such as chest CT, PET, and magnetic resonance imaging (MRI), to assist in staging [7,114,133,244,246,247,249,263], although hybrid imaging seems to be the best match for precision medicine [135]; as is the case for the brain, where both parenchymal and leptomeningeal metastases are possible [369]. Apart from imaging, AI has also been implemented in histopathological images, in search of lymph node or distant metastases [312,313,321]. In a recent meta-analysis, the AUC of determining lymph node metastases with AI applications was 0.74 [12]. In addition, stage—and, more particularly, N stage—is one of the most-chosen features in AI algorithms for the prediction of lung cancer survival [370].

Field carcinogenesis can be detected through optical nanosensing, discriminating stage I lung cancer from disease-free individuals in this way [21]. The clinical feature that is the closest to the lung, and which can change tumor stage, is malignant pleural effusion. AI models can discriminate between benign and malignant pleural effusion with PPV at 93.3%, NPV at 86.67%, sensitivity at 87.5%, specificity at 92.86% and AUC between 0.886 and 0.918 [180,286]. The next-closest anatomical structures that can be invaded by tumor cells are mediastinal lymph nodes, which determine the N stage of the tumor. EBUS image analysis via AI can predict malignant lymph nodes with an accuracy between 72.9% and 82%, sensitivity at 89%, specificity between 72% and 90.8%, PPV at 83%, NPV between 75.9% and 81%, and AUC at 0.782 [9,334]. Chest CT, through image segmentation and classification performed by AI, can also be used to detect malignant mediastinal lymph nodes [339], either as ultra-low-dose CT [63] or regular CT, with accuracy varying between 86.4% and 98.6%, sensitivity between 81% and 98.4%, specificity between 75.3% and 100%, and AUC between 0.859 and 0.864, with best accuracies achieved for sub-carinal level 7 (98.31%) and axillary region (98.74%) and a mean classification accuracy of up to 96.14% for automated tumor N-staging [202,203]. Furthermore, AI programs can predict the mediastinal lymph node invasion in 0–IA stage tumors, with sensitivity, specificity, NPV, and AUC of 69%, 65%, 94%, and 0.761, respectively [204]. Another image modality that is used to evaluate malignant mediastinal lymph nodes is PET, either alone or in combination with CT [256]. AI algorithms using PET scans, through image segmentation and augmentation techniques, determined the malignancy of mediastinal lymph nodes with an accuracy at 80%, sensitivity varying between 74% and 87%, specificity at 84%, and AUC between 0.91 and 0.94 [258,264,265].

Beyond thorax, distant metastases determine the M stage of lung cancer. AI using PET can also determine this stage of lung cancer, either alone [252] or in combination with other exams such as CT or MRI [153]. In three PET studies, the whole TNM stage was categorized based on the anatomical region, with an accuracy varying between 94.3% and 99.1% [262,266,268]; although, in another study, this figure dropped to 55.6% per lesion and 80.2% per patient [254]. Moreover, for distant metastasis exclusively, PET achieved sensitivity, specificity, and accuracy of 45%, 79%, and 63%, respectively [265]. Chest CT has been used to predict visceral pleural and lymphovascular invasion using AI [205,292]. However, the discrimination between multiple primary lung cancer and intrapulmonary metastasis usually requires the assistance of histopathological or molecular techniques [351] while, in a case of miliary intrapulmonary metastatic pattern with over 200 uniformly sized pulmonary nodules, AI was able to identify only 18 [201]. Furthermore, AI presented an accuracy of only 50.9% in identifying metastases in whole-body CT [176] although, for bone metastasis, an AI algorithm adjusted to CT images achieved a sensitivity of 0.894 in identifying them, while it also raised the corresponding accuracy and sensitivity of the radiologists from 0.617 to 0.879 and from 0.680 to 0.902, respectively [206]. Bone scintigraphy is much more sensitive for the detection of bone metastases. In two AI studies using this image modality, the accuracy varied between 76.71% and 81.23%, the sensitivity and specificity were 81.30% and 81.14%, respectively, and the AUC varied between 0.838 and 0.957 [371,372].

Apart from bone scintigraphy, AI can also detect bone metastases in MRI, mainly in the T2W-FS sequence, with encouraging results [154]. However, MRI is mainly used for the study of brain metastases. In one meta-analysis, the true progression of brain metastases after radiotherapy was predicted by AI, with pooled sensitivity and specificity of 77% and 74%, respectively [373]. The tumor volume in brain metastasis was a significant predictor for the PD-L1 status of a tumor in an AI model, with AUC at 0.77 [374]; while, in another study, brain metastasis was a significant predictor of survival, depending on the gene status of EGFR, ALK, and KRAS mutation-positive groups (AUC at 0.977, 0.905, and 0.947, respectively) [290]. In addition, AI models predicted the process of metastasis through the use of a Bayesian network applied to the Epithelial–Mesenchymal Transition data set [375], while another AI model predicted the development of brain metastasis with an AUC at 0.825, using various information from patients’ electronic health records [376].

As can be seen from the information above, AI can manage a vast quantity of information regarding patients’ clinical features—including staging [26,376]—reproduce them through natural language processing [359], and use them in various clinical and research tasks, such as prediction [65,267,359]. Consequently, many AI prognostic models use information related to intrapulmonary metastases and pleural effusions [221]; morphological nodal changes in the mediastinum, supraclavicular, and hilar regions [221,256]; bone [221,377] and brain metastases [290,376]; or TNM stage in general [86,378,379], rendering AI an appropriate tool for decision making in lung cancer staging [68]. Table 10 lists the studies that relate to the sector of staging.

### 3.4. Treatment

#### 3.4.1. General Information

AI has improved the management of patients with lung cancer [293,380]. AI is now able to determine a precise and optimized treatment strategy among all the available ones [273,329,381,382,383,384], while simultaneously predicting, monitoring, and assessing the patient’s response to treatment [134,135,329,330,385,386,387]; usually through the monitoring of various non-invasive biomarkers [387]. In a recent systematic review, the maximum values of the assessment indicators accuracy, sensitivity, specificity, precision, recall, and AUC were found to be 0.99, 1.00, 0.96, 0.98, 0.99, and 0.9929, respectively [388]. To that end, factors such as patients’ socio-economic factors, demographics, clinical history, and genetic profiles have also been utilized [140,320], although radiogenomics have provided the most promising biomarkers—either through chest CT (mainly in mice [237]) or, more often, through PET [153,245,247,248,253]; however, without avoiding pitfalls such as pseudo-progression [153].

More particularly, the aforementioned treatment strategies refer to combinations within the entire spectrum of treatment choices, including the planning of or response to surgery [2,113,328,351,389,390], radiotherapy [113,133,253], chemotherapy [2,113,328], targeted therapy [2,113,133,271,294,296,299,351,361,391,392], or immunotherapy [2,113,133,248,271,296,299,328,351,364,389,393]; even in—or, better, especially in—complex cases such as multiple primary lung cancer [351] or leptomeningeal metastatic spread in brain metastasis [361,369]. Furthermore, AI could identify adverse events such as pneumonitis induced by radiation [133,153,155] or immunotherapy [102,133,155], or those resulting from other clinical entities, such as COVID-19 [155], and discriminate them from disease recurrence [113,133], oligoprogression, and early widespread progression [393]. As far as recurrence in particular, AI has demonstrated a high predictive ability, varying between 87.5% and 89.9% [394].

Other general uses of AI in lung cancer treatment that have been described in the literature or are foreseen to be researched are calculation of cost-effectiveness of different anticancer drugs [395], the suggestion of different nutritional interventions to prevent sarcopenia [106,396], the management of depression and sleep dysfunction [108], the implementation of traditional Chinese medicine for lung cancer management [397], the extraction of precise medical data from vast databases (especially for patients in different lines of treatment) [270,398], the recruitment of eligible patients for participation in research on novel treatments [399,400,401], or providing answers to lung cancer patients, regarding common questions about their disease [402]. Finally, another promising aspect is the use of AI in lung cancer treatment for novel multi-target drug discovery [403,404], like the de novo design of anticancer peptides [405], such as those for Notch signaling dysregulation [406].

Considering the above, AI is thought as an appropriate tool to support clinicians in decision making regarding their patients’ treatment [163,275,288,407,408,409,410,411,412]. The most notable clinical decision support system is the “Watson for Oncology”, which has been tested in different studies and has shown a concordance with multidisciplinary teams’ decisions of between 73.3% and 92.4% [413,414,415,416,417], with better concordance rates for clinical stages I and IV of NSCLC (92.4–100%), and extensive disease of SCLC (100%), while lower rates were observed for clinical stages II and III of NSCLC (77.8–83.3%), and limited disease for SCLC (84.6%) [414,415,416]; furthermore, squamous cell carcinoma cases also demonstrated better concordance than adenocarcinoma cases [417]. The majority of discordant cases concerned surgery (12.3%), radiotherapy (16.7%), and chemoradiotherapy (11.6%), but not patients with metastatic disease (0%) [414], while it remained unclear whether EGFR status or immunotherapy affected concordance [415,416,417]. Two recent meta-analyses have shown that the concordance rates between clinical decision support systems and multidisciplinary teams varied between 81.52% and 85% [418,419], with better concordance for clinical stages I and II (85%), rather than III and IV (83%) [419]. Clinicians in general reported favorable opinions of AI-based cancer technologies, but also cited barriers such as lack of trust, validation studies, and time [420,421]. Table 11 lists the studies that relate to the sector of treatment (general information).

#### 3.4.2. Surgical Treatment

Before reaching the surgical table, clinicians should first and foremost determine which patients present high risk of life-threatening complications. AI may assist in the recognition of frail patients and reduce the incidence of unnecessary surgery [173], especially in those who are in great danger for severe cardiac complications; for example, elderly males with advanced clinical stage of cancer [422]. Furthermore, post-surgical complications in thoracic surgery are continuously diminishing with the implementation of AI. New applications can recognize anatomic structures which play a critical role in thoracic surgery, such as pulmonary vessels (with a total accuracy of 82.8%), especially segmental arteries and lobular veins (with accuracies of 79.7% and 96.3% respectively, which is higher than that of junior thoracic surgeons) [234], and can also determine the pulmonary segments, where most errors frequently occur [234,423], thus creating an atlas for lobectomy and segmentectomy at the sub-segmental or more distal level [423]. This allows for minimally invasive procedures, resulting in the reservation of as much post-operative lung function as possible [424], further minimizing post-operative risks. This has also allowed for the performance of sub-lobectomies in patients with synchronous or metachronous multiple pulmonary nodules (primary binodules, trinodules, or even quadronodules) in both lungs, with concurrent bilateral resection of the lesions [217,425]. It is also worth mentioning that, through various techniques—such as CT- or bronchoscopy-guided placement of markers, intraoperative ultrasonography, 3D printing technology, intraoperative molecular imaging, and AI—these multiple lesions can be recognized intraoperatively [311].

There is no doubt that the major breakthrough regarding the implementation of AI in thoracic surgery has taken place via robotic surgery [13], either through CyberKnife—a robotic radiosurgery platform that allows for the delivery of stereotactic body radiotherapy treatments to the lungs using fiducial-free soft-tissue tracking with 100% accuracy in tumor tracking after training, with no false classifications [127]—or via robotic-assisted thoracic surgery (RATS), which promotes sub-lobar resections to treat early-stage lung cancer, small intrapulmonary metastases, or concurrent localized benign pathology; increases the quality of lymphadenectomies; shortens the learning curve compared to video-assisted thoracic surgery (VATS), although with equivalent oncological results; and has been used in complex resections such as bronchial sleeve [233,426]. Furthermore, AI robotic surgery platforms, used as supplementary imaging tools, can recognize variations in anatomical structures, leading to adjustments in the surgical technique [233].

However, a minimally invasive procedure is not always the correct answer. The kind of surgical treatment has been related, by AI, to the subsequent appearance of bone metastasis and survival [377]. Several AI techniques have been shown to be sufficiently accurate in predicting disease progression after surgical resection, even when using a single 1 mm^2^ tumor core [363]. AI methods can distinguish between minimally invasive adenocarcinoma, invasive adenocarcinoma, and normal tissue with an overall accuracy of 84.9% (sensitivity and specificity of 89% and 82.7% for minimally invasive adenocarcinoma and 94% and 80.6% for invasive adenocarcinoma, respectively) and an AUC of 0.862, which is higher than that of frozen section analysis, senior thoracic surgeons, radiologists, and junior thoracic surgeons [200,427]. Finally, AI algorithms have shown that higher total nodule number and SUV_max_ values in CT and PET scans and abnormal CEA levels are unfavorable prognostic factors, which may suggest that lobectomy should be preferred to segmentectomy in such cases [228,428]. To that end, new research about the effectiveness of surgical treatment and neo-adjuvant chemotherapy with the help of AI, focusing on clinical stages I–IIIA, has been programmed [19,429]. In such cases, AI can assist with data extraction related to therapy from large databases with an accuracy and precision of 100% [359]. Table 12 lists the studies that relate to the sector of surgical treatment.

#### 3.4.3. Radiotherapy

Stereotactic body radiotherapy is another field of lung cancer treatment in which AI has been integrated in various domains, such as image guidance, adaptive radiotherapy, heavy-particle therapy, radiation dose recommendation, and ‘flash’ ultra-high dose-rate radiotherapy [430,431]. Auto-segmentation of organs at risk using AI has provided an average Dice similarity coefficient (DSC) for the lung higher than 0.75 in one study [232], and that for the lung, heart, and liver higher than 0.9 in another study [432]; however, for the esophagus, the respective value varied between 0.71 and 0.87 [432], leading (despite equivalent radiation dose coverage to that of experts) to a 65% reduction in segmentation time and a 32% reduction in inter-observer variability [238]. Another promising AI-based segmentation approach is that using PET [253,259]. Perhaps, through PET segmentation, deeply accelerated adaptive radiation therapy could be implemented, leading a reduction in time to treatment of up to 50% [218]; however, preliminary qualitative results regarding this method seem to be modest [433]. Another aspect of AI implementation in radiotherapy is the identification of types and magnitude of treatment errors, in which the dose comparison method seems to have a larger impact compared to image pre-processing [434,435]. Another AI-based method—namely, the O-ring gantry—seems to effectively deliver stereotactic body radiation therapy, bypassing the mobile target barrier with no local recurrences or acute grade ≥ 3 toxicities after three months [436].

The prediction of the response to treatment and local progression is another use of the AI in radiotherapy, although with AUC between 0.66 and 0.80 [14,219,437]; meanwhile, for true progression of brain metastasis after radiotherapy, the pooled sensitivity and specificity were 77% and 74%, respectively [373]. Apart from response to treatment, adverse events is another field that requires more efficient prediction. The prediction of radiation pneumonitis using AI through cumulative and differential dose–volume histograms yielded an AUC of 0.73 [438] while, in another study using a different methodology, the AUC varied between 0.55 and 0.83 [220]. For hippocampal avoidance during prophylactic cranial irradiation, the implementation of AI planning resulted in a 55% reduction compared to manual plan [439] while, in another study, the thresholds for pulmonary and esophageal toxicity were identified at a mean dose of >18 Gy and lung volume receiving ≥20 Gy (V20) > 37% and a mean dose > 34 Gy and V20 > 37%, respectively, with no significantly identifiable thresholds for cardiac toxicity [440], where the pooled effect estimate of the mean heart dose and the history of cardiovascular diseases were shown to significantly increase the risk of developing late cardiac toxicity [422]. Radiotherapy also seems to be a significant prognostic factor in AI models of bone metastasis and survival prediction [109,377]. Furthermore, AI may assist in recruiting suitable patients for clinical trials based on novel radiotherapy methods [400], as well as extracting radiotherapy-related data from large databases [359]. Table 13 lists the studies that relate to the sector of radiotherapy.

#### 3.4.4. Chemotherapy

AI-based graph regularized matrix factorization is a model that can be used for the de novo prediction of cell drug sensitivities, including chemotherapy [441]. Another AI program with diagnostic accuracy of 97.4% has demonstrated that methotrexate combined with platinum chemotherapy can improve the chemotherapy effect in elderly patients with lung cancer, with low degree of adverse reactions and good overall tolerance, and can be used as a first-line treatment in elderly patients with lung cancer [15]. Furthermore, AI is expected to help in prediction of the efficiency of neo-adjuvant chemotherapy [19], and it is not uncommon for chemotherapeutic agents to be used in combination with other drugs—especially immunotherapeutics—as a first-line treatment in therapy-naïve patients, with measurable results, as has been shown using AI algorithms [261]. Moreover, PET parameters such as SUVmax, MTV 50%, and TLG 50% have been used by AI programs to predict the outcome in patients receiving chemotherapy [107], while chemotherapy has also been shown to be a significant prognostic factor for bone metastasis and survival [109,377].

Other aspects related to chemotherapy are quality of life and emotional status, which seem to be negatively affected by chemotherapy [442]. Adverse events such as vomiting could play a part in this regard, as their successful management has been shown to significantly improve the psychological state, quality of life, serum tumor marker levels, and nursing satisfaction of patients receiving chemotherapy [443]. The nutritional status of chemotherapy-treated patients seems to be another important prognostic factor [444], as patients with advanced squamous cell carcinoma lose a significant amount of skeletal muscle, with the rate of reduction being faster at the end of life [229], making nutritional interventions to prevent sarcopenia important factors in their overall prognosis [396]. Meanwhile, the history of cardiovascular disease seems to be related with cardiac toxicity [422]. Table 14 lists the studies that relate to the sector of chemotherapy.

#### 3.4.5. Targeted Therapy

Apart from surgery, chemotherapy, and radiotherapy—which were the cornerstones of lung cancer treatment for many years, novel therapeutic choices such as targeted therapy and immunotherapy—have emerged in the last two decades. Some AI studies have attempted to predict the EGFR status of a tumor based on certain radiomic features from chest CT, obtaining AUC values around 0.765 [103,151,208]. In another study, two AI modules managed to identify EGFR status in a chest CT tumor with AUC of 0.842 and 0.928, respectively, while one of them achieved an AUC of 0.805 in mutated-EGFR sub-type discrimination (19Del, L858R, other mutations) [101]. Meanwhile, in another study, an AI system achieved an AUC of 0.862 in the discrimination of a panel of 8 mutated genes, including EGFR, ALK, ERBB2, BRAF, MET, ROS1, RET, and KRAS, and an AUC of 0.868 for classifying the EGFR sub-type [211].

Apart from chest CT, PET has also been used for this purpose, with an AI system predicting ALK mutation status with an AUC of 0.87–0.88, which was significantly better than the solely clinical prediction [104], while another AI system predicted the MIB-1 (Ki 67) proliferation group from PET/CT images with an overall accuracy of more than 82% (78% and 86% for the low- and high-proliferation groups, respectively) [260]. In a recent systematic review, 35 imaging-based models were built to predict the EGFR status, with performances ranging from weak (n = 5) to acceptable (n = 11), to excellent (n = 18) and outstanding (n = 1), with positive outcomes reported for the prediction of ALK rearrangement and ALK/ROS1/RET fusions [152]. Apart from radiological modalities, AI has also been utilized in cytological images to predict the status of certain biomarkers which are critical for targeted therapy, with an accuracy of ~0.95 for the EGFR and KRAS groups, ~0.75 for the ALK group, and ~0.80 for the “None” group [337].

Apart from radiomics and cytological images, proteomic and genomic signatures are of the same—or perhaps greater—usefulness for targeted therapy [272]; in this context, AI has been shown to be significantly superior, when compared to manual analysis [445]. In one study, an AI model evaluated KRAS^G12C^ inhibitors and obtained an accuracy score of validation = 0.85 and Q^2^_Ext_ = 0.76, predicting that, of 67 Food and Drug Administration (FDA)-covalent drugs, afatinib, dacomitinib, acalabrutinib, neratinib, zanubrutinib, dutasteride, and finasteride would be active inhibitors [446]. In another study, AI identified octenidine as an inhibitor of B-cell lymphoma-extra large (Bcl-xL)—an anti-apoptotic protein which is an important modulator of the mitochondrial apoptosis pathway [447]—while, in another study, AI analysis revealed that optimizing component formula (OCF)—a molecule extracted from Salvia miltiorrhiza Bunge and Panax ginseng C. A. Meyer—might elicit an anti-lung cancer effect by blocking denticleless E3 ubiquitin protein ligase homolog (DTL)-mediated programmed cell death 4 (PDCD4) ubiquitination and suppressing the mitogen-activated protein kinases (MAPK)/Jun N-terminal kinase (JNK) pathway [448]. In another study, AI methods were used to develop prospective EGFR TKIs that would overcome the limitations of existing third-generation drugs. These methods included the simplified molecular-input line-entry system datasets of the generated ligands, from which drug-like ligands were obtained, through parameter filtering, cyclic skeleton analysis, and virtual screening analysis [449]. Finally, a study hypothesized that acute protein perturbation in signal transduction via targeted anticancer drugs can predict the drug sensitivity of these agent—either as single agents or as rational combinations—predicted, through the use of AI, the most and least probable combinations of drug sensitivity. AUC values of 0.79 and 0.78 were obtained, respectively, which was significantly better than those for predictions based on mutations in genes commonly known to predict responses to drugs (i.e., EGFR, PIK3CA, and KRAS), with an AUC equal to 0.5 [305].

Regarding genomic analysis, the EGFR mutation status prediction AUCs have been compared between different AI programs, and were found to range between 0.740 and 0.771 [18]; meanwhile, in another study, the AUC values for predicting EGFR mutation status and ALK rearrangement status varied from 0.883 to 0.897 and from 0.921 to 0.995, respectively, while the accuracy for distinguishing common and uncommon EGFR mutations varied between 0.83 and 0.93 [362]. In a study which evaluated the three immune phenotypes of lung cancer—namely, inflamed, immune-excluded, and immune-desert—it was found that KRAS mutation, BRAF mutation, and MET splicing variant were mostly observed in the inflamed sub-type, while EGFR and PIK3CA mutations were the most prominent mutations found in the immune-excluded sub-type [450]. In another study, the most frequently detected single nucleotide variants were found in TP53 (n = 107), BRCA2 (n = 24), and NOTCH2 (n = 23), while MYC (n = 10) was the most frequently detected gene amplification, followed by ERBB2 (n = 9) and CCND1 (n = 6) [451]. Another AI study revealed that lung cancer cell lines which are considered insensitive to the cytotoxic action of the p53 activator Nutlin-3, due to the mTOR and PI3K pathway, were actually highly sensitive to the dual PI3K/mTOR inhibitor NVP-BEZ235, while they did not respond to either the PI3K-specific LY294002 or Bcl-XL-specific 2,3-DCPE [452]. Finally, in another AI study, it was found that solasonine treatment rescued the Bcl-2/Bax ratio and cleaved caspase-3 expression, suggesting that solasonine induces apoptosis while, simultaneously, solasonine exhibited anti-inflammatory effects through inhibiting P38 MAPK [453].

Regarding response to treatment, in one study, AI achieved AUCs ranging from 0.748 to 0.813 for EGFR-TKIs treatment, while also determining 29 prognostic deep learning features that were able to identify patients with an EGFR mutation at high risk of TKI resistance [207]. In another study, AI achieved positive predictive values between 75.40% and 80.40% for additional EGFR-TKI survival benefit prediction, when using the probability score of 0.2 as threshold, with higher scores (>0.2) indicating a better prognosis; this study depicts the potential of AI to identify sub-groups of benefiting patients through decoding the commonalities in pre-therapy CT images from stage IV EGFR-mutant NSCLC patients receiving additional survival benefit from EGFR-TKIs [216]. It is expected that, in the near future, AI will be able to incorporate mutated genes (among other aspects), in order to construct prognostic models that predict patient survival accurately and guide individualized treatment [299]; however, at present, AI found that up to 32.1% of lung cancer patients with highly actionable gene variants such as ALK, BRAF, EGFR, ERBB2, MET, RET, and ROS1 might not receive targeted agents as they do not carry a diagnosis of metastatic disease, or as their treatment provider did not discuss the AI results or felt that the patient could not tolerate the therapy [454]. Table 15 lists the studies that relate to the sector of targeted therapy.

#### 3.4.6. Immunotherapy

Since it was introduced as a lung cancer treatment option, immunotherapy has radically changed the landscape of lung cancer therapeutics. An AI-powered analysis managed to define the three immune phenotypes of lung cancer—(1) inflamed (44.0%), (2) immune-excluded (37.1%), and (3) immune-desert (18.9%)—based on TILs in the tumor microenvironment, determine the incidence of the inflamed type for different levels of PD-L1 expression—31.7%, 42.5%, and 56.8% for <1%, 1–49%, and ≥50%, respectively—and, conversely, predict the outcome: median PFS and OS of 4.1 months and 24.8 months for the inflamed type, 2.2 months and 14.0 months for the immune-excluded type, and 2.4 months and 10.6 months for the immune-desert type, respectively [300]. It has been shown that in the inflamed sub-type, which presents higher cytolytic score, the enriched pathways are generally associated with immune response and immune-related cell types, which are highly expressed; whereas, in the immune-excluded sub-type, enriched glycolysis, fatty acid, and cholesterol metabolism pathways are observed [450]. Furthermore, PD-L1 overexpression seems to indicate worse survival; as such, PD-L1 or PD-1 inhibitors can substantially improve patient survival [299]. Thus, it seems highly important to be able to assess PD-L1 scoring and TILs in the tumor immune microenvironment with the assistance of AI, integrating PD-L1 and new biomarkers [327]. Such biomarkers could be established through AI-powered analysis of extracellular vesicles, which provide real-time snapshots of cellular communication, promising to spur the recognition of strategies to overcome resistance to immunotherapy [307]. Apart from currently used ICIs such as CTLA-4 and PD-1/PD-L1, along with combination therapies, AI can assist in the development of new immunological therapies, such as cytokine modulating therapies, cellular therapies such as chimeric antigen receptor (CAR)-T and CAR-natural killer (NK) cells, personalized cancer vaccines, and genetic engineering tools such as CRISPR/Cas9-mediated gene editing of T cells to enhance their effector function [94,297]. AI systems are becoming essential for the development of powerful decision-making tools that are able to deal with this high complexity and provide individualized predictions to better match treatments to individual patients, thus improving patient outcomes and reducing the economic burden on healthcare systems [295].

As with targeted therapy, the recognition of non-invasive, immunotherapy-related biomarkers through radiomics can also be utilized [152,455]. In one study, the combination of a clinical and radiomic model achieved an AUC between 0.63 and 0.67 in predicting PD-L1 expression status [213]; however, in two other studies, the respective AUC varied between 0.830 and 0.856 [211,214], while it reached at 0.868 for the determination of PD-L1 sub-types [211]. In another study, AI demonstrated an AUC of 0.799 in determining the PD-L1 expression status and 0.837 for positive-PD-L1 sub-types (PD-L1 tumor proportion score, 1–49% and ≥50%), and also achieved an AUC of 0.905 in discriminating PD-L1 expression status [101]. Furthermore, in another study including patients with brain metastasis, the respective AUC was calculated between 0.77 and 0.84, depending on the parameters included for prediction [374]. Meanwhile, in a PET study, the AUC for predicting response was 0.69 and that for predicting overall progression was 0.75 [261].

However, compared to predicting PD-L1 status, confirming it is much more important, as it allows the initiation of an appropriate therapy [314,325]. In one study, the AUC of an AI program varied between 0.67 and 0.81 for the identification of PD-L1 status of a tumor [355] while, in another study, a similar AI-powered method achieved an accuracy between 83% and 88% [291]. In a third study, AI was able to identify the different categories of PD-L1 expression (<1%, 1–49%, or ≥50%) which, in turn, can potentially influence treatment decisions [367]. It should be noted that a single biomarker might not be efficient to precisely find suitable patients; thus, combined biomarkers could better instruct the consideration of therapeutic regimens. In this context, multi-modal detection of PD-L1, such as PD-L1 protein, PD-L1 mRNA, and circulating PD-L1, might provide a comprehensive tumor profile and could allow for the determination of patients who are more suitable for ICI therapy [298]. Furthermore, AI has shown that new biomarkers, such as CCDC26 and interferon gamma antisense RNA 1 (IFNG-AS1), may be related to lung cancer and Small Nucleolar RNA Host Gene 3 (SNHG3) may be associated with PD-L1 [279].

Apart from the prediction and identification of suitable patients for immunotherapy, AI can also assist in the prediction of outcomes [456], either by matching the immunotherapy to the host’s immune response, thus improving the activity of existing treatments, or through combining current immunotherapies with novel agents targeting complementary mechanisms [457]. Different AI models have achieved AUC values between 0.82 and 0.91 and accuracy between 0.756 and 0.839 in predicting the response to immunotherapy [215,304], while also increasing the concordance rates between the pathologists to 90.2%, as far as PD-L1 status, thus improving the final outcomes [356]. Furthermore, AI has also managed to accurately recognize the different values of PD-L1 expression (<1%, 1–49%, and ≥50%) with AUCs between 0.934 and 0.950 [100], and generally provided better quantification of PD-L1 expression than manual scoring for different cutoff points [301]. Furthermore, several other AI models have led to the conclusion that the most significant factors for the outcome of immunotherapy are neutrophil to lymphocyte ratio (NLR), eastern cooperative oncology group-performance status (ECOG-PS), PD-L1 expression, line of treatment of immunotherapy, and the combination of chemo-immunotherapy, with accuracy ranging between 0.73 and 0.83 [458].

AI has also enabled the identification, among ∼26,000 genes, of the six most-important genes (CCR7, SELL, GZMB, WARS, GZMH, and LGALS1) for accurate prediction of the response to immunotherapy [303]. An apparent interest in the relationship between the gut microbiome and the response to immunotherapy seems to have emerged recently [306], where it has been shown that Hymenoscyphus immutabilis and Clavulinopsis fusiformis were more abundant in patients with high (≥50%) PD-L1-expressing tumors, whereas Thelephoraceae and Lachnospiraceae bacterium were enriched in patients with ICI-related toxicities and the relative abundances of Bacteroides dorei and Parabacteroides distasonis were associated with long OS (>6 months), whereas the bacteria Clostridium perfringens and Enterococcus faecium and the fungal taxa Cortinarius davemallochii, Helotiales, Chaetosphaeriales, and Tremellomycetes were associated with shorter OS (≤6 months) [212]. Finally, another factor that has been shown to improve the efficiency of immunotherapy is its delivery using nanomaterial-based strategies, which identify and target various immunological markers in the tumor microenvironment and improve treatment outcomes through triggering an immune response in the tumor microenvironment [308].

Finally, another aspect of AI utilization in immunotherapy, which is related with response, is the final outcome [216]. In patients with clinical stage IB–IIIA, neo-adjuvant chemotherapy combined with immunotherapy might improve prognosis, which is the matter of study in forthcoming AI-based research [19]. In one study, several AI models predicting various outcomes, such as disease control rate (DCR), objective response rate (ORR), PFS, and OS, achieved AUCs varying between 0.704 and 0.9526 [459]. The number of TILs has been shown to be a significant predictor of PFS, regardless of PD-L1 status; although patients who experience a response or stable disease to ICI seem to have a higher number of TILs [222]. In another study, an AI-derived survival score achieved AUCs between 0.67 and 0.75 for survival, in a manner independent of clinical, radiological, PD-L1, and histopathological factors, whereas it was affected by morphological nodal changes in the mediastinum, supraclavicular, and hilar regions; lung and bone metastases; as well as pleural effusions, atelectasis, and consolidations [221]. Meanwhile, in another study, various AI systems achieved AUCs between 0.80 and 0.90, and the significant factors that affected survival seemed to include galectin-9, OX40, OX40L, KIR2D, KIR3D, and the NLR combined with the PD-1/PD-L1 signature [460]. Last, but not least, AI models are able to predict, with an accuracy and AUC of over 90%, the emergence of cardiovascular disease in cancer patients who receive immunotherapy [461]. Table 16 lists the studies that relate to the sector of immunotherapy.

### 3.5. Prognosis

Many studies have demonstrated that AI can help in the determination or even improvement of the prognosis in lung cancer patients [2,273,293,381,382,384,391,408,409,410]. The implementation of AI in imaging lung cancer screening programs can lead to a much earlier diagnosis, as AI models have demonstrated a sensitivity and specificity of up to 94.6% and 93.6%, respectively, in lung cancer detection [88] which, in turn, effectively improves OS [4,6,90,130,139,149,283,462]. Apart from imaging screening, AI might also play the same role as proteomic and genomic biomarker-based screening [163,274,288,294]. Additionally, AI can predict patient outcomes based on their clinical data [110,370], such as their nutritional status [396]. Similarly, histopathological or cytological features of the tumor might be utilized by AI in the prediction of lung cancer [312,315,317,321], although the majority of the research in this field concerns radiomic features, either in CT or PET scans [116,131,246,247,249,385,387,463,464,465,466,467,468]. Finally, AI has been used successfully in lung cancer patients for the prediction of response to a treatment [134,298,364,411], or the course of the disease after the implementation of a certain treatment [113,140,297,299].

Screening with LDCT is particularly important for the improvement of lung cancer prognosis [22,23], and AI has been shown to increase the sensitivity of identifying lung tumors by almost 20% [36]. In one study, lung cancer was detected in 1.6% of more than 4300 high-risk participants and, in 91.4% of the cases, it was in an early stage (0–1) with excellent prognosis [50]. Furthermore, it seems that other factors in LDCT, such as coronary artery calcium and low attenuation areas, also exhibit high prognostic value, with AUC sometimes exceeding 0.9 in predicting all-cause mortality, lung cancer mortality, and other major cardiopulmonary events [52,85,86,87]. Clinical data such as sex, age at diagnosis, tumor size and stage, smoking and drinking status, skeletal muscle mass, and body mass index (BMI) have also been used by researchers in AI models to predict the outcomes of lung cancer patients [229,378,402,469,470]. Such models have achieved C-indices above 0.7 for both SCLC [109] and NSCLC [471,472].

The utilization of radiomic features in chest CT scans of patients with lung cancer in order to acquire information about the prognosis is now a common practice [223]. A radiomic model combined with clinical data, similar to those mentioned above, achieved an AUC of 0.949 in disease prognosis [108]. Body composition measurements through radiomic features allowed an AUC of over 0.7 to be achieved in predicting OS [84,444]. Various radiomic features have been used to predict the invasiveness and, consequently, the prognosis of adenocarcinoma, with an AUC of 0.871 [199]. Another AI model predicted lymphovascular invasion from chest CT radiomic features and correlated it with RFS and OS, obtaining a C-index of 0.915 [292]. In another study, the solid-part of a lung tumor in chest CT seemed to be significantly correlated with RFS and OS [226], while a meta-analysis demonstrated that this particular radiomic feature increases the hazard ratios (HRs) for RFS and OS by 2.50 and 2.13, respectively [225]. In another study, radiomic features such as solid-part size, CT value standard deviation, solid-part volume ratio, and bronchus translucency were associated with RFS in patients with resected lung cancer of stage IA [227] while, in another study, the total nodule number was reported as an independent prognostic factor in patients with resected lung cancer of stage III [228]. All these radiomic features are subject to inter-observer variation; thus, semi-automated contouring can improve feature stability [473].

In another study, AI predicted the EGFR status of a lung tumor and its response to TKIs based on radiomic features, with an AUC ranging from 0.748 to 0.813 [207]; while, in four more studies, this was achieved for immunotherapy and consequent survival, with AUCs ranging between 0.60 and 0.95 [100,212,213,214]. In two further chest CT studies using radiomic features, AI predicted the outcomes in patients already under treatment with ICIs [222] or under radiotherapy [219] while, in a meta-analysis of studies with patients receiving radiotherapy, OS was predicted with a C-index of 0.57 [437]. Apart from CT radiomic characteristics, PET features have also been utilized for prognostic purposes via AI in lung cancer [224,261]. SUVmax, MTV 50%, and TLG 50% are some of the features that have been correlated with RFS and OS, with varying results [107,254,256,428]. Radiomic features of another less-used imaging modality in lung cancer—that of MRI for brain metastases—have also been used to predict specific genetic mutations, such as EGFR, ALK, and KRAS, with AUC of over 0.9, as well as OS [290]. Finally, radiomic features have also been used in combination with other features, such as tumor biomarkers and histopathological characteristics, to predict outcomes in lung cancer patients [177,205].

Different biomarkers have also been used alone in lung cancer prognostics [270,375]. There is an increased interest in molecular biomarkers such as microRNA, lncRNA, and differentially methylated regions [277]. More particularly, circulating microRNA has been found even decades before the diagnosis of lung cancer [368], meaning that it might be used in screening for early diagnosis with an AUC as high as 0.91 and, consequently, improved prognosis [278]. As far as the cases of already existing pulmonary nodules, RNA sequencing of peripheral blood leukocytes has identified an immune index that can discriminate between benign and malignant tumors with an AUC exceeding 0.82 [360]. As for histopathology, various biomarkers from the heterogenous tumor microenvironment have been correlated with PFS and OS [310,474]. In fact, there is a particular interest in certain immunohistochemical indices in the tumor immune microenvironment, as they can predict the response to immunotherapy [99,300,309,363].

The type of treatment (surgery, radiotherapy, chemotherapy) is a feature that has been used to predict survival in lung cancer patients with bone metastases [377]. The response to neoadjuvant therapy before surgery is a very important topic [19]. An AI model was able to predict major pathologic remission to pre-operative PD-1 blockade with an AUC of 0.91 [215]. Another significant issue is the prognosis of patients after a curable surgery [429]. In one AI study, OS and RFS were predicted with an AUC between 0.80 and 0.90 and between 0.83 and 0.94, respectively [460]; meanwhile, in another study, the AUCs for DFS, OS, and cancer-specific survival (CSS) at 5 years were 0.890, 0.926, and 0.960, respectively [379]. As far as radiotherapy, in one AI study, the AUC for OS in NSCLC patients who had gone through radiotherapy was 0.65 [400] while, in a meta-analysis of 18 papers including 4719 patients, the respective AUCs for OS and local control were 0.75 and 0.80 [14]. Regarding targeted therapy, an AI study showed better prognosis for PFS in patients under treatment with TKIs [216]. However, the main focus concerning lung cancer treatment and prognosis is directed to immunotherapy [295,301,458]. In a study with patients who were receiving immunotherapy, AI predicted one-year OS with an AUC between 0.69 and 0.75 and six-month PFS with an AUC of 0.67 [221] while, in two more studies, the AUCs for OS reached 0.80 and 0.87 [304,459]. Table 17 lists the studies that relate to the sector of prognosis.

## 4. Conclusions

Progress in the field of lung cancer research is mainly associated with the growing diversity of methods for screening, diagnosis, treatment, and prognosis. However, the introduction of AI technology is part of this process and is expected to boost it further, and it seems that the use of AI in the management of lung cancer is here to stay. Although there are not many AI applications for prevention of the appearance of lung cancer, there has been a tremendous breakthrough in screening programs, in which AI can effectively recognize suspicious lung nodules in chest CT, PET scans, or even X-rays, and can also discriminate benign from malignant tumors. In many cases, the success rate of AI is comparable to or even better than that of experienced radiologists. Furthermore, AI seems to be able to recognize biomarkers that appear in patients who may develop lung cancer in the future, even years before this event. In the field of biopsy, it seems that there are no great developments regarding the process of obtaining the tissue; however, AI has proved particularly capable in assisting pathologists or cytologists in recognizing the types and sub-types of lung tumors, as well as identifying tumor markers, either through immunohistochemistry or genetic analysis, which is of great importance for the implementation of an appropriate treatment. Finally, in the field of treatment, AI can guide personalized treatment options for lung cancer patients and, as such, possibly improve their prognosis. Table 18 summarizes the main applications of AI for lung cancer diagnosis and management, along with their reported performance.

There are several limitations in the present review. In many cases, the boundaries between the diagnostic and therapeutic utilities of many applications were indiscernible; therefore, we chose to present our findings in a way that would allow other researchers or clinicians to use each section of our manuscript autonomously, accessing all the necessary information needed in each case for their research or clinical use. Unfortunately, this led to some overlap between the different sections of our manuscript; however, we believe that, in this way, our manuscript might be more useful for other researchers or clinicians. Furthermore, we chose not to name the different platforms as we did not want our paper to be considered as more favorable to one platform over another. We felt that, by citing the different results and the respective references, it would be convenient enough to whom it might concern to dig a little further and conclude based on their own needs, rather than being biased by our promotion of certain platforms. We think that this approach is the fairest one, as physicians from different countries might value different things in an AI platform based on the conditions of their country (i.e., the financial status), and the same also applies for physicians of different specialties, as surgeons, radiologists, pathologists, cytologists, pulmonologists, and oncologists might also need different things from an AI platform. Nevertheless, it should be kept in mind that the results presented in this thorough review have been reported by the creators of each AI platform respectively; a fact that, perhaps, indicates considerable bias. The fact that we have relied solely on a single database—namely, PubMed—for this review could be viewed as another limitation. It is true that, in previous decades, it might have been unorthodox to rely on a single database for a thorough review. However, at present, the majority of the medical journals and their respective articles are indexed in multiple databases, including PubMed, which is considered a main database. Furthermore, PubMed comprises more than 37 million citations for biomedical literature from MEDLINE, life science journals, and online books, and is widely considered as the most thorough and qualitative medical database available. Although there are numerous databases, such as IEEE Xplore and ACM Digital Library, which are rich sources of computer science and machine learning literature, and thus seem relevant to the present review, this is not the case, as these are non-medical databases, which present scientific literature more suitable for computer engineers, rather than physicians, who are the main addressees of the present review. Moreover, while reporting the results, we focused on indices such as accuracy, sensitivity, specificity, positive and negative predictive value, and AUC, and did not report results concerning computational complexity, training time, interpretability, ease of use, robustness, generalizability, infrastructural requirements for hospitals or clinics to adopt these technologies, and so on. Those indices were regarded as being beyond the scope of the present review and, additionally, as the papers that were included in the present review were mainly medical, they generally did not report such indices. For the same reason, the discussion regarding ethical concerns or the implementation of AI in low-resource settings was minimal and only relevant to the clinical applications. The abovementioned considerations could serve as the basis for future reviews.

## Figures and Tables

**Figure 1 cancers-17-00882-f001:**
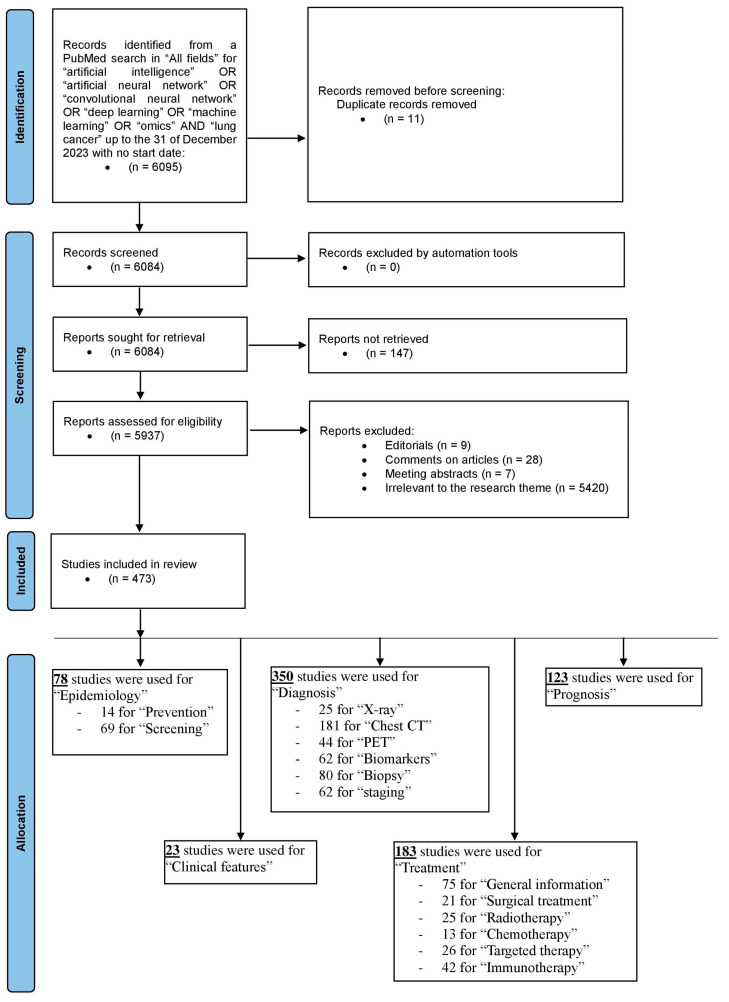
PRISMA flow diagram of study selection.

**Table 1 cancers-17-00882-t001:** Studies in the sector of prevention.

Reference	Type of Study	Participants	AI Status	Main Conclusions
Uthoff JM, et al., 2022 [17]	Benchmarking study	278 individuals with a pulmonary nodule	In vitro diagnostics	Automatically extracted quantitative CT imaging features of the lung can be informative of the differentiation between malignant and benign pulmonary nodules, without requiring nodule segmentation and analysis.
Yang R, et al., 2022 [18]	Benchmarking study	7413 patients with lung adenocarcinoma	In vitro diagnostics	AI-based prediction model may become a practical tool to guide EGFR mutation prediction in lung adenocarcinoma.
Zhou H, et al., 2021 [19]	Prospective, single-arm, multi-center, phase II trial	40 patients with lung cancer	Research	The efficacy of ICIs is influenced by many factors, including patient’s driver genes and smoking status.
Ye M, et al., 2022 [20]	Benchmarking study	728 participants	Research	Age and smoking history were used to improve an AI model of lung cancer prediction based on radiological features alone.
Daneshkhah A, et al., 2023 [21]	Laboratory study	96 patients with lung cancer and 83 with negative LDCT	In vitro diagnostics	Smoking-related molecular abnormalities of the cells in the buccal mucosa, known as “field carcinogenesis,” were used for the detection of lung cancer in early stages using an optical spectroscopic statistical nanosensing AI technique.
Revel MP, et al., 2022 [22]	Prospective cohort study protocol	2400 asymptomatic women	Research	Assistance with smoking cessation will be offered to current smokers.
Mapanga W, et al., 2021 [23]	Consensus study	27 health stakeholders	Research	Considerable barriers to the early identification and treatment of lung cancer exist.
Liu D, et al., 2021 [25]	Benchmarking study	4644 participants	Research	Epidemiological characteristics should be considered in lung cancer screening, which can significantly improve the efficiency of the AI model alone.
Gauthier MP, et al., 2022 [26]	Retrospective observational study	333 patients with lung cancer	In vitro diagnostics	Automated data abstraction from electronic health records is highly accurate and faster than manual abstraction.

**Table 2 cancers-17-00882-t002:** Studies in the sector of screening.

Reference	Type of Study	Participants	AI Status	Main Conclusions
Ye M, et al., 2022 [20]	Benchmarking study	728 participants	Research	A model integrating clinical characteristics, radiological characteristics of pulmonary nodules, AI analysis of LDCT data, and liquid biopsy achieved the best diagnostic performance.
Revel MP, et al., 2022 [22]	Prospective cohort study protocol	2400 asymptomatic women	Research	An initial low-dose CT scan will be performed, with subsequent follow-ups at 1 and 2 years.
Liu D, et al., 2021 [25]	Benchmarking study	4644 participants	Research	Epidemiological characteristics should be considered in lung cancer screening, which can significantly improve the efficiency of the AI model alone.
Aberle DR, et al., 2011 [30]	Randomized Controlled Trial	53,454 persons at high risk for lung cancer	Research	Screening with the use of low-dose CT reduces mortality from lung cancer.
de Koning HJ, et al., 2020 [31]	Randomized Controlled Trial	15,789 participants	Research	Lung-cancer mortality was significantly lower among those who underwent volume CT screening than among those who underwent no screening.
Zhang Y, et al., 2022 [33]	Benchmarking study	860 participants	Research	AI-assisted reading greatly improves the accuracy and sensitivity of nodule detection in chest CT and benefits lung nodule detection, especially for non-solid nodules.
Guo J, et al., 2020 [34]	Benchmarking study	1590 lung nodules from 306 patients	In vitro diagnostics	An AI-based system with the potential to improve the performance and work efficiency of radiologists in lung cancer screening was presented.
Yeh MC, et al., 2021 [35]	Benchmarking study	11,617 lung cancer patients and 1,423,154 controls	Research	AI model achieved excellent performance in predicting lung cancer within 1 year and has potential to be deployed for digital patient screening using electronic medical records.
Chao HS, et al., 2023 [36]	Retrospective observational study	200 patients with lung cancer	In vitro diagnostics	AI algorithm-embedded CT screening significantly ameliorates tedious LDCT practices for doctors.
Alexander M, et al., 2020 [37]	Benchmarking study	102 lung cancer patients and 10 lung cancer clinical trials	Research	The AI-based clinical trial matching system allows for efficient and reliable screening of cancer patients in clinical trials, with 95.7% accuracy for exclusion and 91.6% accuracy for overall eligibility assessment.
Chauvie S, et al., 2020 [38]	Clinical trial	Patients of the SOS clinical trial	Research	AI could help radiologists to reduce the number of false positive in chest digital tomosynthesis.
Adams SJ, et al., 2021 [39]	Benchmarking study	192 LDCT from a representative cohort of 3197 patients	Research	Using an AI risk score combined with Lung-RADS at baseline lung cancer screening may result in fewer follow-up investigations and substantial cost savings.
Venkatesh C, et al., 2022 [40]	Benchmarking study	CT images from a private hospital	In vitro diagnostics	This study reveals that AI accuracy is improved, and the results are compiled using particle swarm optimization and genetic algorithms.
Cui X, et al., 2022 [41]	Benchmarking study	180 patients with lung nodules and 180 controls	In vitro diagnostics	The AI system can accurately detect pulmonary nodules on LDCT, with an acceptable false positive rate of 1 nodule per scan and higher detection performance than double reading.
Katase S, et al., 2022 [42]	Benchmarking study	1997 chest CT scans	Research	An AI-based computer-aided detection system that is robust to imaging conditions was developed, and using it as a second reader increased detection performance.
Adams SJ, et al., 2023 [43]	Benchmarking study	1331 nodules in 963 subjects	In vitro diagnostics	A computer-assisted diagnosis software improved risk classification from chest CTs of screening and incidentally detected lung nodules.
Xu J, et al., 2023 [44]	Benchmarking study	LUNA16 dataset	Research	Compared with the mainstream object detection AI algorithms, the improved model was superior to the above models in every index.
Heuvelmans MA, et al., 2021 [45]	Multi-center clinical study	2106 lung nodules (205 malignant)	In vitro diagnostics	AI showed excellent performance on identification of benign lung nodules in a multi-center external dataset, ruling out malignancy with high accuracy in about one-fifth of the patients with 5–15 mm nodules.
Du W, et al., 2022 [46]	Benchmarking study	194 nodules in 152 patients	Research	AI pulmonary nodule detection system can improve the coincidence rate and accuracy of early diagnosis of lung cancer, shorten the average detection time, and provide more accurate information for clinical decision-making.
Lancaster HL, et al., 2022 [47]	Benchmarking study	283 participants who underwent an ultra-LDCT scan	In vitro diagnostics	Through the use of AI as an impartial reader in baseline lung cancer screening, negative misclassification results could exceed that of four out of five experienced radiologists, and radiologists’ workload could be drastically diminished by up to 86.7%.
Duan S, et al., 2020 [48]	Benchmarking study	372 lung cancer and 470 benign lung patients	In vitro diagnostics	A three-layer diagnosis system for lung cancer based on 14 epidemiological data and clinical symptoms, 5 tumor biomarkers, and 22 CT nodule-based radiomic features.
Lam S, et al., 2023 [49]	Benchmarking study	1394 CTs from 697 participants	In vitro diagnostics	Lung nodule volume change measurements had better sensitivity and specificity for classifying malignant from benign lung nodules when applied to solid lung nodules from high-quality CT scans.
Zhang Y, et al., 2023 [50]	Observational study	4395 participants	Research	LDCT screening detected a high proportion of early-stage lung cancer patients in a Chinese high-risk population.
Singh R, et al., 2021 [51]	Retrospective observational study	123 LDCT examinations	In vitro diagnostics	AI-vessel suppression improved the detection and classification of sub-solid nodules into ground-glass nodules and part-solid nodules on LDCT of the chest for two radiologist readers.
Balbi M, et al., 2023 [52]	Retrospective observational study	4098 volunteers	In vitro diagnostics	Automated coronary artery calcium and quantitative emphysema added prognostic information to age, sex, and pack-years for predicting mortality, but not lung cancer incidence, in a lung cancer screening setting.
Chetan MR, et al., 2022 [53]	Benchmarking study	10,485 lung nodules in 4660 participants	In vitro diagnostics	Nodule size and morphology play the largest role in AI prediction, with nodule internal texture and background parenchyma playing a limited role.
Wan YL, et al., 2020 [54]	Benchmarking study	50 patients with 75 lung nodules	Research	More part-solid nodules were proven to be malignant than benign (90.9% vs. 9.1%) and more solid nodules to be benign than malignant (86.7% vs. 13.3%).
Cheng X, et al., 2022 [55]	Benchmarking study	CT images from 357 patients	Research	Using the proposed AI system in separating peripheral lung cancer and focal pneumonia in chest CT data yielded an accuracy similar to that of a junior physician.
Diao K, et al., 2022 [56]	Prospective cohort study	251 patients with high risk nodules	Research	The AI algorithm had high negative predictive value but low positive predictive value in diagnosing high-risk lung lesions in a clinical setting.
Schwyzer M, et al., 2022 [57]	Laboratory study	A phantom with 15 pulmonary nodules	In vitro diagnostics	Dose reduction and iterative reconstruction settings have an impact on detectability of pulmonary nodules by AI software.
Hendrix W, et al., 2023 [59]	Benchmarking study	2189 patients with lung nodules	In vitro diagnostics	The AI system reliably detects benign and malignant pulmonary nodules in clinically indicated CT scans and can potentially assist radiologists in this setting.
Gorenstein L, et al., 2023 [60]	Prospective study	123 patients underwent ultra-LDCT	Research	A novel AI-based denoising method allows for a substantial decrease in radiation dose, without misinterpretation of actionable pulmonary nodules or life-threatening findings such as aortic aneurysms.
Morozov SP, et al., 2019 [61]	Benchmarking study	Patients underwent ultra-LDCT	Research	AI protocols allow for high-quality chest computed tomography for lung nodule detection with an effective dose of less than 1 mSv.
Andre F, et al., 2023 [62]	Benchmarking study	176 patients underwent heart and lung CT	Research	ECG-gated tin-filtered ultra-low dose chest CT protocol allows for reliable coronary artery disease and lung cancer screening with ultra-low radiation exposure.
Zhao K, et al., 2022 [63]	Prospective observational study	141 patients	In vitro diagnostics	The measured diameters of pulmonary lesions and lymph nodes in AI images of ultra-low dose CT were highly close to those of contrast-enhanced CT.
Schwyzer M, et al., 2018 [64]	Benchmarking study	50 lung cancer patients and 50 controls	Research	AI algorithms may aid fully automated lung cancer detection, even at very low effective radiation doses of 0.11 mSv.
Lyu Q, et al., 2023 [65]	Benchmarking study	62 LDCT and 76 brain MRI reports	Research	It is feasible to utilize large language models in clinical education, and further efforts are needed to address their limitations and maximize their potential.
Gandomkar Z, et al., 2022 [66]	Benchmarking study	36 radiologists, 10 malignant and 10 benign cases	In vitro diagnostics	Occlusion-based saliency maps were used to explain an AI decision-making process for radiologists who, in turn, provided insight into the level of agreement between the AI’s decision and radiological lexicon.
Sousa JV, et al., 2023 [67]	Benchmarking study	1079 cases of lung nodules in 1005 patients	In vitro diagnostics	The combination of different types in lung cancer screening may have the potential to produce more comprehensive analyses of the disease by the models.
Obuchowski NA, et al., 2019 [69]	Methodology study	Various AI models	Research	The statistical considerations for designing a study to test a new AI pre-screening algorithm for identifying normal lung cancer screening CTs are described.
Wang J, et al., 2023 [70]	Methodology study	Data from LIDC-IDRI, LUNA16, NLST and NELSON datasets	Research	This study proposes a standardized data pre-paration process while clearly demonstrating the principles and sequence of different steps and as a result, a data preparation pipeline can be quickly realized by following these proposed steps and implementing the suggested example codes and tools.
Dong C, et al., 2023 [71]	Laboratory study	CT screening data with 19,229,589 parameters	In vitro diagnostics	This work presents a privacy-preserved solution for medical imaging AI applications that minimizes the risk of patient health information exposure.
Ziegelmayer S, et al., 2022 [72]	Cost-effectiveness study	Model input parameters based on current literature	Research	The use of an AI-based system in the initial low-dose CT scan of lung cancer screening is a feasible diagnostic strategy from a cost-effectiveness perspective.
Yoo H, et al., 2020 [73]	Benchmarking study	5485 participants	In vitro diagnostics	The AI algorithm performed better than radiologists for the detection of pulmonary nodules on digital radiographs; thus, when used as a second reader, the AI algorithm may help to detect lung cancer.
Yoo H, et al., 2021 [74]	Benchmarking study	173 images from 98 cancer and 346 images from 196 non-cancer patients	Research	The AI algorithm can enhance the performance of readers for the detection of lung cancers on chest radiographs when used as second reader.
Matsubara N, et al., 2019 [75]	Benchmarking study	Chest X-ray images	In vitro diagnostics	The chances of missing abnormalities may be reduced using the proposed AI method for bone suppression in chest X-ray images.
Simon J, et al., 2023 [76]	Benchmarking study	10,573 LDCTs from 6127 participants	In vitro diagnostics	This AI model can accurately predict future lung cancer risk in females and males in a real-world setting and performs better in females than in males for predicting 6-year lung cancer risk.
Wang G, et al., 2022 [77]	Benchmarking study	1456 participants	In vitro diagnostics	This AI method may be useful for early detection of lung cancer or large-scale screening of high-risk populations for cancer prevention.
Yang H, et al., 2022 [78]	Benchmarking study	93 patients with pulmonary nodules	In vitro diagnostics	Circulating genetically abnormal cells had a higher diagnostic value than traditional tumor markers in early-stage lung cancer and a supportive value for AI in the diagnosis of cancer based on lung nodules.
Bahado-Singh R, et al., 2022 [79]	Laboratory study	10 lung cancer and 20 controls	Research	Using AI and DNA methylation analysis of ctDNA, high lung cancer detection rates were achieved.
Huang Q, et al., 2023 [80]	Benchmarking study	950 scleral images from 95 participants	Research	AI system for distinguishing lung neoplasms based on scleral data may hold promise for evaluating the risk of lung cancer in an asymptomatic population in areas with a shortage of medical resources.
Yang J, et al., 2022 [81]	Benchmarking study	1046 patients who underwent LDCT and DXA the same period	Research	Routine chest CT with AI is of great value in opportunistic screening for osteopenia or osteoporosis, which can quickly screen the population at high risk of osteoporosis without increasing the radiation dose, thus reducing the incidence of osteoporotic fracture.
Naghavi M, et al., 2023 [82]	Benchmarking study	169 cases with 29,567 lung scan images + 6941 controls	In vitro diagnostics	No extra radiation plus the high prevalence of asymptomatic osteoporosis makes AI an ideal screening tool for osteopenia and osteoporosis in CT scans performed for other reasons.
Aquino GJ, et al., 2022 [83]	Retrospective observational study	273 patients who underwent LDCT	Research	AI-Left Atrium Volume index is significantly associated with increased risk of new-onset atrial fibrillation, hospitalization due to heart failure, and major adverse cardiac and cerebrovascular events within 5 years.
Xu K, et al., 2023 [84]	Secondary analysis of a clinical trial	20,768 participants	In vitro diagnostics	The body composition measurements automatically derived from baseline low-dose CT examinations added predictive value for lung cancer death, CVD death, and all-cause death, but not for lung cancer incidence.
Sabia F, et al., 2023 [85]	Clinical trial	2239 volunteers	In vitro diagnostics	Fully automated coronary artery calcium scoring was effective in predicting all-cause mortality at 12 years in a lung cancer screening setting.
Ruggirello M, et al., 2023 [86]	Retrospective observational study	6495 heavy smokers	In vitro diagnostics	The lower coronary artery calcification burden observed in women at all ages might contribute to explaining their lower rates of all-cause mortality and better lung cancer survival.
Chamberlin J, et al., 2021 [87]	Retrospective observational study	117 patients who underwent LDCT	In vitro diagnostics	AI rapidly and accurately identifies significant risk factors for cardiopulmonary disease on standard screening low-dose chest CT.

**Table 3 cancers-17-00882-t003:** Studies in the sector of clinical features.

Reference	Type of Study	Participants	AI Status	Main Conclusions
Yang R, et al., 2022 [18]	Benchmarking study	7413 patients with lung adenocarcinoma	In vitro diagnostics	The five most influential features for EGFR mutation were smoking consumption, sex, cholesterol, age, and albumin to globulin ratio.
Ye M, et al., 2022 [20]	Benchmarking study	728 participants	Research	Age and smoking history were used to improve an AI model of lung cancer prediction based on radiological features alone.
Mapanga W, et al., 2021 [23]	Consensus study	27 health stakeholders	Research	Improving clinical integration across all levels of the healthcare system is essential for lung cancer management.
Duan S, et al., 2020 [48]	Benchmarking study	372 lung cancer and 470 benign lung patients	In vitro diagnostics	A three-layer diagnosis system for lung cancer based on 14 epidemiological data and clinical symptoms, 5 tumor biomarkers, and 22 CT nodule-based radiomic features.
Sicular S, et al., 2022 [92]	Case report	A 46 year-old male	In vitro diagnostics	Additional information provided by AI software holds promise to prevent missed detection of lung cancer on chest radiographs.
Ahmed I, et al., 2023 [95]	Benchmarking study	LIDC-IDRI data set	In vitro diagnostics	Using AI, the False Positive Rate is reduced and the accuracy of nodule detection and classification is enhanced.
Mezher MA, et al., 2022 [96]	Benchmarking study	309 patients with benign and malignant tumors	Research	While using Genetic Folding Strategy for classifying lung cancer, an accuracy of 96.2% was obtained.
Wang C, et al., 2022 [98]	Benchmarking study	1135 NSCLC patients with PD-L1 status	In vitro diagnostics	An AI model combined with clinical characteristics improved prediction capabilities, which could assist physicians in making rapid decisions regarding clinical treatment options.
Tan Q, et al., 2021 [99]	Benchmarking study	167 patients with NSCLC	Research	The density of CD45RO⁺ TIL and the combination of CD45RO⁺ TIL and PD-L1 in tumor areas is significantly correlated with clinicopathological features and prognosis of NSCLC, and can be used as a new marker.
Tong H, et al., 2022 [100]	Retrospective multi-cohort study	1405 NSCLC patients	In vitro diagnostics	^18^F-FDG PET/CT radiomics–clinical combined model could be a clinically practical method to non-invasively detect the tumor immune status in NSCLCs.
Wang C, et al., 2022 [101]	Benchmarking study	9136 patients with NSCLC	Research	Both clinical indicators and radiomics features showed a complementary role in prediction and provided accurate estimates to predict EGFR and PD-L1 status.
Zhou G, et al., 2023 [103]	Benchmarking study	164 patients with lung adenocarcinoma	Research	There were significant differences between the patient’s sex, smoking history, pulmonary nodule morphology, and the EGFR gene.
Chang C, et al., 2021 [104]	Benchmarking study	526 patients with lung adenocarcinoma	In vitro diagnostics	PET/CT radiomics-based AI model has potential to be used as a non-invasive diagnostic method to help in diagnosing ALK mutation status for lung adenocarcinoma patients in the clinic.
Kataoka Y, et al., 2021 [105]	Sequential mixed methods study	11 patients and 1 caregiver	Research	Medical staff providing care to patients with lung cancer can use AI to educate patients on how they can manage their symptoms.
Lemos R, et al., 2022 [106]	Prospective observational cohort study protocol	300 patients with breast or lung cancer	Research	If this AI application is effective, it will provide healthcare systems with a novel and innovative method to screen depressive symptoms in oncological settings.
Li X, et al., 2019 [107]	Benchmarking study	137 patients with NSCLC	Research	Several potential prognostic biomarkers of PET/CT imaging were extracted for the prediction of survival and selecting patients with NSCLC who are more likely to benefit from chemotherapy.
Tang FH, et al., 2023 [108]	Benchmarking study	422 NSCLC patients	Research	The proposed AI model has the potential to improve NSCLC prognosis and facilitate personalized treatment decisions.
Zhang D, et al., 2023 [109]	Benchmarking study	21,093 patients’ clinical data	In vitro diagnostics	The interpretable AI survival predictive tool for SCLC, using clinical data, had a reliable predictive value on their overall survival.

**Table 4 cancers-17-00882-t004:** Studies in the sector of X-rays.

Reference	Type of Study	Participants	AI Status	Main Conclusions
Chauvie S, et al., 2020 [38]	Clinical trial	Patients of the SOS clinical trial	Research	AI could help radiologists to reduce the number of false positives in chest digital tomosynthesis.
Yoo H, et al., 2021 [74]	Benchmarking study	173 images from 98 cancer and 346 images from 196 non-cancer patients	Research	The AI algorithm enhanced the performance of readers for the detection of lung cancers on chest radiographs when used as second reader.
Matsubara N, et al., 2019 [75]	Benchmarking study	Chest X-ray images	In vitro diagnostics	The chances of missing abnormalities may be reduced through using the proposed AI method for bone suppression in chest X-ray images.
Sicular S, et al., 2022 [92]	Case report	A 46-year-old male	In vitro diagnostics	Additional information provided by AI software holds promise to prevent missed detection of lung cancer on chest radiographs.
Kwak SH, et al., 2023 [115]	Retrospective observational study	75 patients with lung cancer	In vitro diagnostics	AI is beneficial for incidental detection of early lung cancer in chest radiographs.
Horry M, et al., 2021 [119]	Benchmarking study	ChestX-ray14 dataset LIDC-IDRI dataset and JSRT dataset	Research	Decision trees mined using AI may be considered as a starting point for refinement into clinically useful multi-variate lung cancer malignancy models for implementation as a workflow augmentation tool to improve the efficiency of human radiologists.
Chiu HY, et al., 2022 [120]	Benchmarking study	1196 chest X-rays	In vitro diagnostics	The median time from detection to diagnosis for radiologists assisted with AI was 46 (3–523) days, longer than that for radiologists (8 (0–263) days).
Ueda D, et al., 2021 [121]	Multi-center retrospective observational study	312 chest radiographs	In vitro diagnostics	The AI-based detection system was able to improve the ability of physicians to detect nodules of lung cancer in chest radiographs.
Higuchi M, et al., 2023 [122]	Benchmarking study	5800 chest X-rays	In vitro diagnostics	The proprietary AI algorithms had a similar accuracy for interpreting chest radiographs, when compared with previous studies and radiologists.
Maiter A, et al., 2023 [123]	Retrospective observational study	5722 chest X-rays from 5592 patients	In vitro diagnostics	The findings highlight the importance of training and testing software in representative datasets, with broader implications for the implementation of AI tools.
Bernstein MH, et al., 2023 [124]	Observational study	90 chest X-rays from 90 patients	Research	When AI is wrong, radiologists make more errors than they would have without AI; therefore, this manuscript provides evidence for two AI implementation strategies that reduce the deleterious effects of incorrect AI.
Yoo H, et al., 2022 [125]	Retrospective simulation study	CXRs of 5887 adults	In vitro diagnostics	The feasibility of sorting and removing normal CXRs using AI with a tailored cut-off to increase efficiency and reduce the workload of radiologists.
Lafrenière M, et al., 2023 [127]	Benchmarking study	271 images from 129 patients	In vitro diagnostics	An AI model can distinguish features of trackable and untrackable lesions in digital radiographs reconstructed from the simulation CT scan, and can predict successful candidates for fiducial-free lung tumor tracking.
Gichoya JW, et al., 2022 [128]	Observational study	4 private and 5 public datasets	Research	AI can accurately predict self-reported race, even from corrupted, cropped, and noised medical images.

**Table 5 cancers-17-00882-t005:** Studies in the sector of chest CT.

Reference	Type of Study	Participants	AI Status	Main Conclusions
Gu Q, et al., 2022 [15]	Benchmarking study	80 elderly patients with lung cancer	In vitro diagnostics	Evaluation of the clinical efficacy of pemetrexed combined with platinum-based chemotherapy in the treatment of elderly lung cancer using electronic CT images based on AI algorithms.
Uthoff JM, et al., 2022 [17]	Benchmarking study	278 individuals with a pulmonary nodule	In vitro diagnostics	Automatically extracted quantitative CT imaging features of the lung can be informative of the differentiation between malignant and benign pulmonary nodules, without requiring nodule segmentation and analysis.
Ye M, et al., 2022 [20]	Benchmarking study	728 participants	Research	A model integrating clinical characteristics, radiological characteristics of pulmonary nodules, AI analysis of LDCT data, and liquid biopsy achieved the best diagnostic performance.
Revel MP, et al., 2022 [22]	Prospective cohort study protocol	2400 asymptomatic women	Research	The primary objective is to compare CT scan readings by a single lung cancer screening-trained, AI-assisted radiologist to that of an expert double reading.
Aberle DR, et al., 2011 [30]	Randomized Controlled Trial	53,454 high risk for lung cancer persons	Research	Screening with the use of low-dose CT reduces mortality from lung cancer.
de Koning HJ, et al., 2020 [31]	Randomized Controlled Trial	15,789 participants	Research	Lung-cancer mortality was significantly lower among those who underwent volume CT screening than among those who underwent no screening.
Zhang Y, et al., 2022 [33]	Benchmarking study	860 participants	Research	AI-assisted reading greatly improves the accuracy and sensitivity of nodule detection in chest CT and benefits lung nodule detection, especially for non-solid nodules.
Guo J, et al., 2020 [34]	Benchmarking study	1590 lung nodules from 306 patients	In vitro diagnostics	An AI-based system with the potential to improve the performance and work efficiency of radiologists in lung cancer screening was presented.
Chao HS, et al., 2023 [36]	Retrospective observational study	200 patients with lung cancer	In vitro diagnostics	AI algorithm-embedded CT screening significantly ameliorates tedious LDCT practices for doctors.
Adams SJ, et al., 2021 [39]	Benchmarking study	192 LDCT from a representative cohort of 3197 patients	Research	Using an AI risk score combined with Lung-RADS at baseline lung cancer screening may result in fewer follow-up investigations and substantial cost savings.
Venkatesh C, et al., 2022 [40]	Benchmarking study	CT images from a private hospital	In vitro diagnostics	This study reveals that AI accuracy is improved, and the results are compiled using particle swarm optimization and genetic algorithms.
Cui X, et al., 2022 [41]	Benchmarking study	180 patients with lung nodules and 180 controls	In vitro diagnostics	The AI system can accurately detect pulmonary nodules on LDCT, with an acceptable false-positive rate of 1 nodule per scan and higher detection performance than double reading.
Katase S, et al., 2022 [42]	Benchmarking study	1997 chest CT scans	Research	An AI-based computer-aided detection system that is robust to imaging conditions was developed, and using it as a second reader increased detection performance.
Adams SJ, et al., 2023 [43]	Benchmarking study	1331 nodules in 963 subjects	In vitro diagnostics	A computer-assisted diagnosis software improved risk classification from chest CTs of screening and incidentally detected lung nodules.
Xu J, et al., 2023 [44]	Benchmarking study	LUNA16 dataset	Research	Compared with the mainstream object detection AI algorithms, the improved model was superior to these models in every index.
Heuvelmans MA, et al., 2021 [45]	Multi-center clinical study	2106 lung nodules (205 malignant)	In vitro diagnostics	AI showed excellent performance on identification of benign lung nodules in a multi-center external dataset, ruling out malignancy with high accuracy in about one-fifth of the patients with 5–15 mm nodules.
Du W, et al., 2022 [46]	Benchmarking study	194 nodules in 152 patients	Research	AI pulmonary nodule detection system can improve the coincidence rate and accuracy of early diagnosis of lung cancer, shorten the average detection time, and provide more accurate information for clinical decision making.
Lancaster HL, et al., 2022 [47]	Benchmarking study	283 participants who underwent an ultra-LDCT scan	In vitro diagnostics	Through the use of AI as an impartial reader in baseline lung cancer screening, negative-misclassification results could exceed that of four out of five experienced radiologists, and radiologists’ workloads could be drastically diminished by up to 86.7%.
Duan S, et al., 2020 [48]	Benchmarking study	372 lung cancer and 470 benign lung patients	In vitro diagnostics	A three-layer diagnosis system for lung cancer based on 14 epidemiological data and clinical symptoms, 5 tumor biomarkers, and 22 CT nodule-based radiomic features.
Lam S, et al., 2023 [49]	Benchmarking study	1394 CTs from 697 participants	In vitro diagnostics	Lung nodule volume change measurements had better sensitivity and specificity for classifying malignant from benign lung nodules when applied to solid lung nodules from high-quality CT scans.
Zhang Y, et al., 2023 [50]	Observational study	4395 participants	Research	LDCT screening detected a high proportion of early-stage lung cancer patients in a Chinese high-risk population.
Singh R, et al., 2021 [51]	Retrospective observational study	123 LDCT examinations	In vitro diagnostics	AI-based vessel suppression improved the detection and classification of sub-solid nodules into ground-glass nodules and part-solid nodules on LDCT of the chest for two radiologist readers.
Balbi M, et al., 2023 [52]	Retrospective observational study	4098 volunteers	In vitro diagnostics	Automated coronary artery calcium and quantitative emphysema added prognostic information to age, sex, and pack-years for predicting mortality, but not lung cancer incidence, in a lung cancer screening setting.
Chetan MR, et al., 2022 [53]	Benchmarking study	10,485 lung nodules in 4660 participants	In vitro diagnostics	Nodule size and morphology play the largest role in AI prediction, with nodule internal texture and background parenchyma playing a limited role.
Wan YL, et al., 2020 [54]	Benchmarking study	50 patients with 75 lung nodules	Research	More part-solid nodules were proven to be malignant than benign (90.9% vs. 9.1%) and more solid nodules to be benign than malignant (86.7% vs. 13.3%).
Cheng X, et al., 2022 [55]	Benchmarking study	CT images from 357 patients	Research	The proposed AI system for separating peripheral lung cancer and focal pneumonia in chest CT data achieved accuracy similar to that of a junior physician.
Diao K, et al., 2022 [56]	Prospective cohort study	251 patients with high risk nodules	Research	The AI algorithm had high negative predictive value but low positive predictive value in diagnosing high-risk lung lesions in a clinical setting.
Schwyzer M, et al., 2022 [57]	Laboratory study	A phantom with 15 pulmonary nodules	In vitro diagnostics	Dose reduction and iterative reconstruction settings have an impact on detectability of pulmonary nodules by AI software.
Hendrix W, et al., 2023 [59]	Benchmarking study	2189 patients with lung nodules	In vitro diagnostics	The AI system reliably detects benign and malignant pulmonary nodules in clinically indicated CT scans and can potentially assist radiologists in this setting.
Gorenstein L, et al., 2023 [60]	Prospective study	123 patients underwent ultra-LDCT	Research	A novel AI-based denoising method allows for a substantial decrease in radiation dose, without misinterpretation of actionable pulmonary nodules or life-threatening findings such as aortic aneurysms.
Morozov SP, et al., 2019 [61]	Benchmarking study	Patients underwent ultra-LDCT	Research	AI protocols allow for the performance of high-quality chest computed tomography for lung nodule detection with an effective dose of less than 1 mSv.
Andre F, et al., 2023 [62]	Benchmarking study	176 patients underwent heart and lung CT	Research	ECG-gated tin-filtered ultra-low dose chest CT protocol allows for reliable coronary artery disease and lung cancer screening with ultra-low radiation exposure.
Zhao K, et al., 2022 [63]	Prospective observational study	141 patients	In vitro diagnostics	The measured diameters of pulmonary lesions and lymph nodes in AI images of ultra-low dose CT are highly close to those of contrast-enhanced CT.
Sousa JV, et al., 2023 [67]	Benchmarking study	1079 cases of lung nodules in 1005 patients	In vitro diagnostics	The combination of different types in lung cancer screening may have the potential to produce more comprehensive analyses of the disease by the models.
Ziegelmayer S, et al., 2022 [72]	Cost-effectiveness study	Model input parameters based on current literature	Research	The use of an AI-based system in the initial low-dose CT scan of lung cancer screening is a feasible diagnostic strategy from a cost-effectiveness perspective.
Yang H, et al., 2022 [78]	Benchmarking study	93 patients with pulmonary nodules	In vitro diagnostics	Circulating genetically abnormal cell had a higher diagnostic value than traditional tumor markers in early-stage lung cancer and a supportive value for AI in the diagnosis of cancer based on lung nodules.
Yang J, et al., 2022 [81]	Benchmarking study	1046 patients who underwent LDCT and DXA the same period	Research	The routine chest CT with AI has great value in opportunistic screening for osteopenia or osteoporosis, which can quickly screen the population at high risk of osteoporosis without increasing radiation dose, thus reducing the incidence of osteoporotic fracture.
Naghavi M, et al., 2023 [82]	Benchmarking study	169 cases with 29,567 lung scan images + 6941 controls	In vitro diagnostics	No extra radiation plus the high prevalence of asymptomatic osteoporosis makes AI an ideal screening tool for osteopenia and osteoporosis in CT scans done for other reasons.
Aquino GJ, et al., 2022 [83]	Retrospective observational study	273 patients who underwent LDCT	Research	AI-Left Atrium Volume index is significantly associated with increased risk of new-onset atrial fibrillation, hospitalization due to heart failure, and major adverse cardiac and cerebrovascular events within 5 years.
Xu K, et al., 2023 [84]	Secondary analysis of a clinical trial	20,768 participants	In vitro diagnostics	The body composition measurements automatically derived from baseline low-dose CT examinations added predictive value for lung cancer death, CVD death, and all-cause death, but not for lung cancer incidence.
Sabia F, et al., 2023 [85]	Clinical trial	2239 volunteers	In vitro diagnostics	Fully automated coronary artery calcium scoring was effective in predicting all-cause mortality at 12 years in a lung cancer screening setting.
Ruggirello M, et al., 2023 [86]	Retrospective observational study	6495 heavy smokers	In vitro diagnostics	The lower coronary artery calcification burden observed in women at all ages might contribute to explaining their lower rates of all-cause mortality and better lung cancer survival.
Chamberlin J, et al., 2021 [87]	Retrospective observational study	117 patients who underwent LDCT	In vitro diagnostics	AI rapidly and accurately identifies significant risk factors for cardiopulmonary disease on standard screening low-dose chest CT.
Ahmed I, et al., 2023 [95]	Benchmarking study	LIDC-IDRI data set	In vitro diagnostics	Using AI, the false positive rate is reduced, and the accuracy is enhanced for nodule detection and classification.
Wang C, et al., 2022 [98]	Benchmarking study	1135 NSCLC patients with PD-L1 status	In vitro diagnostics	AI model combined with clinical characteristics improved prediction capabilities, which could assist physicians in making rapid decisions on clinical treatment options.
Tong H, et al., 2022 [100]	Retrospective multi-cohort study	1405 NSCLC patients	In vitro diagnostics	^18^F-FDG PET/CT radiomics–clinical combined model could be a clinically practical method to non-invasively detect the tumor immune status in NSCLCs.
Zhou G, et al., 2023 [103]	Benchmarking study	164 patients with lung adenocarcinoma	Research	There were significant differences between the symptoms of cavity sign, hair prick sign, and chest depression sign and EGFR gene, but no significant difference between the symptoms of lobulation sign and EGFR gene.
Chang C, et al., 2021 [104]	Benchmarking study	526 patients with lung adenocarcinoma	In vitro diagnostics	PET/CT radiomics-based AI model has potential to be used as a non-invasive diagnostic method to help diagnose ALK mutation status for lung adenocarcinoma patients in the clinic.
Tang FH, et al., 2023 [108]	Benchmarking study	422 NSCLC patients	Research	The proposed AI model has the potential to improve NSCLC prognosis and facilitate personalized treatment decisions.
Lafrenière M, et al., 2023 [127]	Benchmarking study	271 images from 129 patients	In vitro diagnostics	An AI model can distinguish features of trackable and untrackable lesions in digital radiographs reconstructed from the simulation CT scan, and can predict successful candidates for fiducial-free lung tumor tracking.
Gichoya JW, et al., 2022 [128]	Observational study	4 private and 5 public datasets	Research	AI can accurately predict self-reported race, even from corrupted, cropped, and noised medical images.
Chen L, et al., 2022 [160]	Benchmarking study	8540 pulmonary CT images of 7716 patients	In vitro diagnostics	This AI system for nodule detection, segmentation, and analysis can effectively reduce the workloads of radiologists while assisting in diagnosis.
Lan CC, et al., 2022 [164]	Observational study	CT images of 60 patients	In vitro diagnostics	The AI-assisted program improved the performance of detecting lung nodules, especially for error-prone nodules.
Liu X, et al., 2022 [165]	Benchmarking study	500 cases of chest CT in T1 stage lung cancer	Research	The automatic learning of early lung cancer chest CT images by AI can lead to high sensitivity and specificity of early lung cancer identification, assisting doctors in the diagnosis of lung cancer.
Li X, et al., 2022 [166]	Benchmarking study	200 chest CT data	Research	AI achieved the detection of all malignancies and improved the sensitivity of pulmonary nodules detection beyond radiologists, with a low false positive rate after excluding small nodules.
Shafiee MJ, et al., 2022 [167]	Benchmarking study	LIDC-IDRI dataset	In vitro diagnostics	This AI model showed improved sensitivity (93.42%), specificity (82.39%), and diagnostic accuracy (88.78%) relative to previous radiomics approaches.
Vasilev Y, et al., 2022 [168]	Comparative study	2 chest CT external datasets	In vitro diagnostics	Update of the probability threshold together with the population shift seems to be valuable to preserve AI models performance without re-training them.
Tan JR, et al., 2022 [169]	Prospective observational study	4900 chest CT studies	In vitro diagnostics	This AI implementation demonstrates the viability of using an automated and secure radiology double-read system to improve patient safety in radiology workflows.
Fan W, et al., 2024 [170]	Benchmarking study	229 pulmonary nodules from 130 patients	Research	AI had a sensitivity of 94.69% and radiologists had a sensitivity of 85.40% in identifying pulmonary nodules and an AUC of 0.814 and 0.798 respectively.
Judson MA, et al., 2022 [171]	Benchmarking study	222 chest CT scans	In vitro diagnostics	This AI model shows promise to distinguish sarcoidosis from alternative pulmonary conditions, especially lung cancer, using minimal radiologic data.
Pan W, et al., 2022 [172]	Retrospective observational study	360 nodules in 309 patients	Research	AI demonstrates favorable accuracy for CT diagnosis of lung cancer and requires a shorter time for film reading.
Elia S, et al., 2022 [173]	Retrospective observational study	71 patients with solitary pulmonary nodules	Research	AI analysis of radiomic features may be applied to the decision-making process in elderly frail patients with suspicious nodules, in order to minimize the false positive rate and reduce the incidence of unnecessary surgery.
Saied M, et al., 2022 [174]	Comparative study	1007 nodules were obtained from 551 patients	In vitro diagnostics	Deep learning methods with transfer learning showed several benefits over statistical learning in terms of nodule prediction performance and saving efforts and time in training on large datasets.
Naseer I, et al., 2022 [175]	Comparative study	888 CT scans from a publicly available dataset	In vitro diagnostics	The performances of the three AI architectures were measured for accuracy, specificity, sensitivity, positive predictive value, false omission rate, negative predictive value, and F1 score.
Li K, et al., 2022 [176]	Benchmarking study	486 consecutive resected lung lesions	In vitro diagnostics	The accuracy of the AI for risk prediction was very high for primary lung cancers, but poor for metastases and benign lesions.
Brocki L, et al., 2022 [177]	Benchmarking study	CT images of lung tumors from the LIDC-IDRI dataset	In vitro diagnostics	The proposed model combines biomarkers and radiomics features in an interpretable AI model which demonstrates excellent performance for lung nodule malignancy classification.
Sun H, et al., 2022 [178]	Observational study	32 patients with benign lung lesions and 58 with lung cancer	Research	Among the cases initially misdiagnosed as malignant yet subsequently identified as benign, a notable number of these instances were solid nodules, often resembling malignant lesions in imaging characteristics.
Uthoff J, et al., 2022 [179]	Benchmarking study	463 lung nodules	Research	Radiomic features extracted from the parenchyma surrounding lung nodules contain valid signals with spatial relevance for the task of lung cancer risk classification.
Ozcelik N, et al., 2022 [180]	Benchmarking study	408 CT images from 64 patients with pleural effusion	In vitro diagnostics	Advances in AI-aided diagnostic analysis of CT images and obtaining a pre-diagnosis of pleural fluid may reduce the need for interventional procedures by guiding physicians regarding which patients may have malignancies.
Orooji M, et al., 2022 [181]	Benchmarking study	CT scans of 195 patients	Research	The best AI classifier yielded an AUC of 77.8% in differentiating between benign and malignant nodules.
Baldwin DR, et al., 2022 [182]	Benchmarking study	1397 nodules in 1187 patients	In vitro diagnostics	The AI score has better discriminatory value and allows a larger proportion of benign nodules to be identified without missing cancers than the Brock model.
Sahu P, et al., 2022 [183]	Benchmarking study	LIDC and IDRI dataset	Research	The AI model achieved a mean of 93.18% classification accuracy for pulmonary nodules.
Zhang L, et al., 2022 [185]	Benchmarking study	104 cases of pulmonary nodules	In vitro diagnostics	CT image-based radiomics could differentiate benign from malignant pulmonary nodules, which might provide a new method for clinicians to determine pulmonary nodules.
Yanagawa M, et al., 2022 [186]	Benchmarking study	285 patients with lung adenocarcinoma	Research	The AI model can support a less-experienced radiologist to improve diagnostic accuracy for pulmonary invasive adenocarcinoma without deteriorating any diagnostic performances.
Zhang H, et al., 2022 [187]	Benchmarking study	1018 CT scans	Research	Extensive cross-evaluation results illustrate the effect of unsure-annotation data for AI-based methods in lung nodule classification.
Tsakok MT, et al., 2022 [188]	Retrospective observational study	158 benign and 32 malignant nodules	In vitro diagnostics	AI might reduce the need for follow-up scans and intervention in low-scoring benign nodules, whilst potentially accelerating the investigation and treatment of high-scoring cancer nodules.
Wu XY, et al., 2022 [189]	Retrospective observational study	61 patients, 21 of them with COVID-19	Research	COVID-19 patients with solitary pulmonary nodules might have been misdiagnosed using the AI system, suggesting that the AI system needs to be further optimized, especially in the event of a new disease outbreak.
Blanc D, et al., 2022 [190]	Benchmarking study	A dataset consisted of 1031 patients	In vitro diagnostics	A fully functional pipeline using an AI algorithm was developed, resulting in high capabilities for pulmonary nodule classification.
Hung SC, et al., 2022 [191]	Benchmarking study	1018 case records involving 1010 patients	In vitro diagnostics	AI model demonstrated better diagnostic performance, showing calcification: 0.9873 ± 0.006, margin: 0.9207 ± 0.009, subtlety: 0.9026 ± 0.014, texture: 0.9685 ± 0.006, sphericity: 0.8652 ± 0.021, and malignancy: 0.9685 ± 0.006.
Bhattacharjee A, et al., 2022 [192]	Benchmarking study	1190 images from 110 cases	In vitro diagnostics	The AI model obtained 99.39% accuracy, 99.33% precision, and 98% recall for lung cancer multi-class classification.
Lyu X, et al., 2024 [193]	Benchmarking study	1057 lung nodules in 420 cases	In vitro diagnostics	The AI assisted diagnosis system is a valuable tool for training junior radiology residents and medical imaging students to perform pulmonary nodules detection and diagnosis.
Aydın N, et al., 2022 [194]	Benchmarking study	301 patients with lung cancer	Research	Lung tumors were successfully detected and differentiated between adenocarcinoma and squamous cell carcinoma groups with an AI model.
Toda R, et al., 2022 [195]	Benchmarking study	Chest CT images of 66 patients with lung cancer	In vitro diagnostics	AI can perform constant feature learning and generate images with a variety of shapes using a small dataset.
Kirienko M, et al., 2022 [196]	Retrospective single-center observational study	151 surgically treated patients with NSCLC	Research	Radiogenomic data may provide clinically relevant information in NSCLC patients regarding the histotype, aggressiveness, and progression, while gene expression analysis showed potential new biomarkers and targets which are valuable for patient management and treatment.
Dunn B, et al., 2022 [197]	Benchmarking study	CT images from 355 lung cancer patients	In vitro diagnostics	The results demonstrate the potential of AI-based computer-aided diagnostic tools to automatically diagnose subtypes of lung cancer through coupling deep learning image segmentation with supervised classification.
Fang W, et al., 2022 [198]	Retrospective observational study	224 ground-glass nodules from 210 patients	In vitro diagnostics	AI parameters are valuable for identifying subtypes of early lung adenocarcinoma and have improved diagnostic efficacy when combined with CT signs.
Su Z, et al., 2022 [199]	Benchmarking study	223 patients with early-stage adenocarcinoma	Research	AI-assisted diagnosis system could predict the invasive subtypes of early-stage lung adenocarcinoma appearing as pulmonary nodules, and has a certain predictive value.
Lv Y, et al., 2022 [200]	Benchmarking study	901 patients with early-stage resected NSCLC	Research	This AI model achieved comparable performance to intraoperative frozen section analysis in determining tumor invasiveness, and may contribute to clinical decisions related to the extent of surgical resection.
Zhang Y, et al., 2022 [201]	Case report	A 57-year-old woman	In vitro diagnostics	This medical imaging material can help to recognize lung cancer metastasis and has good value for popularization of science.
Iuga AI, et al., 2022 [202]	Benchmarking study	89 contrast-enhanced chest CTs	Research	The proposed AI approach for automatic classification of thoracic lymph node levels in chest CT, as well as the proof-of-principle experiment for automatic N-staging, revealed promising results.
Geng M, et al., 2022 [203]	Benchmarking study	48 patients with lung cancer and 34 controls	Research	The image segmentation algorithm based on the AI model could extract CT lung cancer lesions efficiently and quasi-determinately, which could be used as an effective tool for radiologists to diagnose lung cancer.
Shimada Y, et al., 2022 [204]	Benchmarking study	720 patients with resected stage 0-IA NSCLC	In vitro diagnostics	Measuring the average solid-CT value of tumors for pathological lymph node metastasis may have broad applications, such as guiding individualized surgical approaches and post-operative treatment.
Lee KH, et al., 2022 [205]	Benchmarking study	1426 patients with stage I–IV lung cancer	Research	The CT-based composite score obtained from AI of histopathological features predicted survival in early-stage lung adenocarcinomas with high reproducibility.
Huo T, et al., 2022 [206]	Benchmarking study	126 patients with lung cancer	Research	The proposed AI model for automatic lung cancer bone metastasis detection can improve diagnostic efficiency and reduce the diagnosis time and workload of junior radiologists.
Wang S, et al., 2022 [207]	Benchmarking study	18,232 patients with lung cancer	In vitro diagnostics	This AI system provides a non-invasive method to detect EGFR genotype and identify patients with an EGFR mutation at high risk of TKI resistance.
Silva P, et al., 2022 [208]	Benchmarking study	Chest CT images	Research	This work provides a more accurate and fast characterization of lung cancer EGFR status, which is fundamental for the development of an adequate treatment plan for lung cancer patients.
Shiri I, et al., 2022 [209]	Benchmarking study	136 NSCLC patients	In vitro diagnostics	AI significantly improved the performance of all radiomic models toward more successful prediction of EGFR and KRAS mutation statuses in lung cancer patients.
Rios Velazquez E, et al., 2022 [210]	Benchmarking study	763 adenocarcinoma patients	Research	Somatic mutations drive distinct radiographic phenotypes that can be predicted via radiomics.
Shao J, et al., 2022 [211]	Benchmarking study	1096 NSCLC patients	In vitro diagnostics	This AI system simultaneously analyzed 10 molecular expressions, which might be utilized as an assistive tool in conjunction with or in lieu of ancillary testing to support precision treatment options.
Dora D, et al., 2022 [212]	Cohort study	129 advanced stage NSCLC patients	Research	CT-based texture analysis and gut microbiome signatures can predict overall survival, the response to therapy, the PD-L1 expression, and toxicity in NSCLC patients treated with ICI.
Tonneau M, et al., 2022 [213]	Benchmarking study	642 advanced NSCLC patients	In vitro diagnostics	A risk prediction model combining clinical and AI-based radiomics was generalizable following CT scan harmonization and AI generalization methods, with similar performances to routine oncology practice in predicting ICI response.
Trebeschi S, et al., 2022 [214]	Benchmarking study	1055 primary and meta- lesions from 203 + 262 NSCLC patients	Research	Radiographic characteristics of lesions on standard-of-care imaging may function as non-invasive biomarkers for response to immunotherapy, and may show utility for improved patient stratification in both neoadjuvant and palliative settings.
Jin W, et al., 2022 [215]	Benchmarking study	143 primary lung cancer patients	Research	The AI model predicts the response of lesions to PD-1 blockade, which implies the importance of introducing the non-linear dynamic effects and AI approaches to the original delta-radiomics in the future.
Deng K, et al., 2022 [216]	Retrospective study	570 patients with stage IV EGFR-mutant NSCLC	In vitro diagnostics	This study developed and externally validated a pre-operative CT image-based AI model to predict the survival benefits of EGFR-TKI and ICI therapies in stage IV NSCLC patients, which will facilitate optimized and individualized treatment strategies.
Zhang DY, et al., 2022 [217]	Case report	1 patient with multiple primary lung cancer	In vitro diagnostics	The surgical plan for multiple pulmonary nodules should be carefully considered and AI might be a helpful tool.
Hooshangnejad H, et al., 2022 [218]	Virtual clinical trial	Data from 15 patients with NSCLC	In vitro diagnostics	AI reduces the common online adaptive radiation therapy course by at least two weeks, resulting in a 50% shorter time to treatment to lower the chance of restaging and loss of local control.
Kim H, et al., 2022 [219]	Retrospective study	135 patients with lung cancer	Research	This study exhibited the external validity and transportability of the CT-based AI prediction model for radiotherapy candidates, predicting local recurrence-free survival, disease-free survival, and overall survival.
Zhang Z, et al., 2022 [220]	Benchmarking study	701 patients with lung cancer	In vitro diagnostics	This novel AI approach combining CT and radiation dose images can effectively and accurately predict the occurrence of radiation pneumonitis, and this model can be adjusted easily to fit new cohorts.
Trebeschi S, et al., 2022 [221]	Benchmarking study	152 stage-IV NSCLC patients	Research	AI-derived survival score was independent of clinical, radiological, PD-L1, and histopathological factors, and visual analysis of AI-generated prognostic heatmaps revealed the relative prognostic importance of morphological nodal changes in the mediastinum, supraclavicular, and hilar regions; lung and bone metastases; as well as pleural effusions, atelectasis, and consolidations.
Park C, et al., 2022 [222]	Benchmarking study	512 patients with NSCLC	Research	In this AI model, CT radiomics and predicted tumor-infiltrated lymphocytes were significantly associated with immune checkpoint inhibitors’ outcomes in NSCLC patients.
van Amsterdam WAC, et al., 2022 [223]	Laboratory study	1018 scans from 1010 patients	Research	Through combining AI and structural causal models, an unbiased individual prognosis prediction can be achieved.
Huang B, et al., 2022 [224]	Multi-center retrospective observational study	1168 lung nodules	Research	AI trained using pre-treatment FDG-PET/CT performed well in predicting lung malignancy progression and overall survival—comparable to a radiomics approach—and could inform treatment options and improve patient care.
Lee JH, et al., 2022 [225]	Retrospective cohort study and systematic review with meta-analysis	612 patients with adenocarcinoma and 13 studies with 12,080 patients	In vitro diagnostics	Early-stage lung adenocarcinomas manifesting as solid nodules in pre-operative chest CT, which indicates the absence of ground-glass opacity, were associated with poor post-operative survival.
Kawaguchi Y, et al., 2022 [226]	Benchmarking study	772 patients with stage 0-I adenocarcinoma	Research	The solid-part volume measured by AI was superior to conventional methods in predicting the prognosis of clinical stage 0–I adenocarcinoma.
Shimada Y, et al., 2022 [227]	Benchmarking study	642 patients with NSCLC	In vitro diagnostics	CT-based radiomics coupled with AI contributes to the non-invasive prediction of early recurrence in patients with c-stage 0–IA NSCLC.
Chen X, et al., 2022 [228]	Retrospective observational study	33,410 nodules among 2126 participants	Research	The AI-detected total nodule number is significantly associated with survival rates in patients with surgically resected stage III NSCLC.
Lee J, et al., 2022 [229]	Retrospective observational study	70 consecutive patients with SQCLC	Research	Patients with advanced SQCLC under palliative chemotherapy lose a significant amount of skeletal muscle, and the muscle area reduction is faster at the end of life.
Smith D, et al., 2022 [230]	Benchmarking study	100 patients with lung nodules	In vitro diagnostics	Autonomous software use can affect radiologist assessments of pulmonary nodules, such that suggested follow-up is altered.
Yoo SJ, et al., 2022 [231]	Benchmarking study	Νon-contrast chest CT images from 203 patients	Research	AΙ achieved excellent performance in automatically delineating the boundaries of lung parenchyma with extensive pathological conditions on non-contrast chest CT images.
Gu H, et al., 2022 [232]	Benchmarking study	105 locally advanced NSCLC	Research	The 2D–3D hybrid AI model achieved accurate automatic segmentation of lung lobes on conventional slice-thickness CT of locally advanced lung cancer patients, exhibiting good clinical practicability for radiotherapy.
Sadeghi AH, et al., 2022 [233]	Prospective observational pilot study	10 patients referred for segmentectomy	In vitro diagnostics	The successful development and clinical application of the first dedicated AI and virtual reality platform for the planning of pulmonary segmentectomy is demonstrated.
Chen X, et al., 2022 [234]	Benchmarking study	27 cases with lung cancer	Research	The AI segmentation algorithm achieves the recognition of the segmental pulmonary artery and the lobular pulmonary vein, with efficiency approximately that of junior thoracic surgery attendings, and is potentially beneficial for lung cancer patients.
Dudurych I, et al., 2022 [235]	Benchmarking study	Images from chest CT scans datasets	Research	Manual correction results in significantly improved segmentations and is potentially a useful and time-efficient method to improve the performance of AI tools for a specific hospital or research dataset.
Ferl GZ, et al., 2022 [236]	Benchmarking study	3100 chest CT images	Research	Faster and more efficient automated segmentation of lungs and lung tumors in mouse micro-CT scans.
van de Worp WRPH, et al., 2022 [237]	Benchmarking study	60 unique manually delineated lung tumors	Research	This AI algorithm provides a helpful tool for the non-invasive detection and analysis of tumor take, tumor growth, and therapeutic effects in mouse orthotopic lung cancer models.
Hosny A, et al., 2022 [238]	Benchmarking study	2208 patients with lung cancer	In vitro diagnostics	In silico geometric segmentation metrics might not correlate with clinical utility of the models, while experts’ segmentation style and preference might affect model performance.
Toda R, et al., 2022 [239]	Benchmarking study	147 tumors of 133 lung cancer patients	In vitro diagnostics	The proposed method can reproduce tumors with complex shapes, and the one-to-many image generation suggests effectiveness in data augmentation applications.
Bilal A, et al., 2022 [240]	Benchmarking study	LIDC-IDRI dataset	In vitro diagnostics	In this AI system, noise in the pictures was reduced using a weighted filter, and the improved Gray Wolf Optimization method was performed before segmentation with watershed modification and dilation operations.
Peters AA, et al., 2022 [241]	Benchmarking study	169 patients with 196 nodules	Research	CT dose reduction may affect the AI-based lung cancer prediction regarding the classification of pulmonary malignancies, potentially altering pulmonary nodule management.
Hardie RC, et al., 2022 [242]	Benchmarking study	59 pediatric patients and 89 deidentified scans	In vitro diagnostics	This study indicates a need for pediatric-specific lung nodule AI systems, trained on data specific to pediatric patients.

**Table 6 cancers-17-00882-t006:** Studies in the sector of PET.

Reference	Type of Study	Participants	AI Status	Main Conclusions
Schwyzer M, et al., 2018 [64]	Benchmarking study	50 lung cancer patients and 50 controls	Research	AI algorithms may aid in fully automated lung cancer detection, even at very low effective radiation doses of 0.11 mSv.
Wang C, et al., 2022 [98]	Benchmarking study	1135 NSCLC patients with PD-L1 status	In vitro diagnostics	AI model combined with clinical characteristics improved prediction capabilities, which could assist physicians in making rapid decisions on clinical treatment options.
Chang C, et al., 2021 [104]	Benchmarking study	526 patients with lung adenocarcinoma	In vitro diagnostics	PET/CT radiomics-based AI model has potential to be used as a non-invasive diagnostic method to help diagnose ALK mutation status for lung adenocarcinoma patients in the clinic.
Li X, et al., 2019 [107]	Benchmarking study	137 patients with NSCLC	Research	Several potential prognostic biomarkers of PET/CT imaging have been extracted for predicting survival and selecting patients with NSCLC who are more likely to benefit from chemotherapy.
Kirienko M, et al., 2022 [196]	Retrospective single-center observational study	151 surgically treated patients with NSCLC	Research	Radiogenomic data may provide clinically relevant information in NSCLC patients regarding the histotype, aggressiveness, and progression, while gene expression analysis showed potential new biomarkers and targets which are valuable for patient management and treatment.
Shiri I, et al., 2022 [209]	Benchmarking study	136 NSCLC patients	In vitro diagnostics	AI significantly improved the performance of all radiomic models toward more successful prediction of EGFR and KRAS mutation statuses in lung cancer patients.
Huang B, et al., 2022 [224]	Multi-center retrospective observational study	1168 lung nodules	Research	AI trained using pre-treatment FDG-PET/CT performed well in predicting lung malignancy progression and overall survival, comparable to a radiomics approach, and could inform treatment options and improve patient care.
Hosny A, et al., 2022 [238]	Benchmarking study	2208 patients with lung cancer	In vitro diagnostics	In silico geometric segmentation metrics might not correlate with clinical utility of the models, while experts’ segmentation style and preference might affect model performance.
Pfaehler E, et al., 2021 [252]	Benchmarking study	124 PET/CT images of patients with NSCLC	Research	Both AI algorithms lead to accurate segmentations for both primary tumors as well as metastasis and, therefore, are good candidates for PET tumor segmentation.
Sharif MS, et al., 2010 [253]	Benchmarking study	2 phantom datasets with simulated tumors	Research	Tumor detection, classification, and quantification in PET imaging at early stage of disease are important issues for clinical diagnosis, assessment of response to treatment, and radiotherapy planning.
Borrelli P, et al., 2022 [254]	Benchmarking study	115 patients undergone FDG PET/CT	In vitro diagnostics	Both manual and AI total lesion glycolysis (TLG) were significantly associated with overall survival; however, its low sensitivity makes this AI model less useful to support clinical reading, reporting, and staging.
Borrelli P, et al., 2021 [255]	Benchmarking study	112 patients referred for FDG PET/CT due to suspected lung cancer	Research	The AI-based method is suitable for the detection of lung lesions and automatic calculation of TLG in small- to medium-sized tumors.
Borrelli P, et al., 2022 [256]	Benchmarking study	320 patients referred for FDG PET/CT due to suspected lung cancer	Research	AI was trained to segment lung tumors and thoracic lymph nodes, and fully-automated AI-based total lesion glycolysis (TLG) measurements were significantly associated with overall survival in patients with lung cancer.
Liu Z, et al., 2023 [257]	Benchmarking study	225 patients with stage IIB/III NSCLC	Research	Evaluation using AI can lead to findings that are inconsistent with evaluation using the task-based figure of merit.
Rogasch JMM, et al., 2023 [258]	Benchmarking study	491 consecutive patients with NSCLC	Research	An AI model based on routinely available variables from [^18^F]FDG-PET/CT improved accuracy in mediastinal staging, compared to established visual assessment criteria.
Kerhet A, et al., 2010 [259]	Benchmarking study	2 patients with NSCLC	Research	The potential applicability of a machine learning methodology as an auxiliary tool for radiation treatment planning in NSCLC was demonstrated.
Palumbo B, et al., 2020 [260]	Benchmarking study	78 patients with NSCLC	Research	The AI model showed ability to predict the correct MIB-1 proliferation group with overall accuracy >82%.
Ventura D, et al., 2023 [261]	Benchmarking study	44 patients with advanced NSCLC	In vitro diagnostics	This radiomics-based model might be able to predict response in advanced NSCLC patients treated with CKI-based first-line therapy with or without chemotherapy, including disease progression or death.
Joshi S, et al., 2022 [262]	Benchmarking study	99 patients with NSCLC	In vitro diagnostics	This detector can identify the stage of a lung tumor with an average accuracy of 97%.
Alshmrani GM, et al., 2023 [263]	Benchmarking study	A dataset of 51 PET/CT studies	Research	The segmentation of lung tumors, treatment evaluation, and tumor stage classification have become significantly more accessible with the advent of PET/CT scans.
Wallis D, et al., 2022 [264]	Benchmarking study	205 patients with lung cancer	Research	This study went directly from whole-body [^18^F]FDG-PET/CT scans to pathological mediastinal lymph node localization, with a sensitivity of 0.88.
Tau N, et al., 2020 [265]	Benchmarking study	264 patients with NSCLC	Research	An AI to analyze segmented PET images of patients with NSCLC can yield moderately high accuracy for designation of N category, although this may be insufficient to preclude invasive lymph node sampling.
Weikert T, et al., 2023 [266]	Benchmarking study	364 FDG-PET/CTs of patients with lung cancer	In vitro diagnostics	TNM categorization based on the anatomical region approach was correct in 94.3% of lesions.
Rogasch JMM, et al., 2023 [267]	Laboratory study	13 questions regarding [^18^F]FDG PET/CT	In vitro diagnostics	ChatGPT might adequately substitute for advice given to patients by nuclear medicine staff in the investigated settings.
Kasinathan G, et al., 2022 [268]	Benchmarking study	94 patients with NSCLC	In vitro diagnostics	For complete TNM evaluation, PET/CT accuracy varied between 97–99.1%.

**Table 7 cancers-17-00882-t007:** Summary of the biomarkers that have been used in the implementation of AI approaches for lung cancer diagnosis.

Biomarker	Significance
Chromatin structural changes in buccal mucosa	At two clinical sites, patients with stage-I lung cancer were distinguished from cancer-free controls with an AUC of 0.92 ± 0.06 and 0.82 ± 0.11, respectively.
DNA methylation analysis of circulating cell-free tumor DNA	4389 CpGs (cytosine methylation loci) in coding genes and 1812 CpGs in non-protein coding DNA regions were differentially methylated in lung cancer; using these, the AI achieved an AUC = 1.00 (95% CI, 0.90–1.00) for lung cancer detection.
mRNA expression	The NSCLC AI prediction model that used mRNA expression achieved a 93.7% macro F1-score.
microRNA (miRNA)	The analysis of miRNA expression yielded a sensitivity of 96.4% and a specificity of 88.2% for the detection of early-stage NSCLC with AUC = 0.91 (CI 95%: 0.80–1.0).
Long non-coding RNA (lncRNA)	In two datasets, AI computed the best AUCs at 0.8701, 0.8953, and 0.9110 under cross-validation on lncRNAs and at 0.9490, 0.9157, and 0.9708 on the two datasets.
Single-cell ribonucleic acid (scRNA)	This biomarker was used for the classification of carcinomas and for clustering NSCLC sub-types.
Peripheral venous circulating tumor cells (CTCs) Pulmonary venous CTCs Circulating tumor DNA (ctDNA)	Predicted unfavorable outcomes and distant metastases.
Tumor-infiltrating lymphocytes (TILes)	Patients with high predicted TILes (≥median) had significantly prolonged progression-free survival, compared to those with low predicted TILes (median 4.0 months [95% CI 2.2–5.7] versus 2.1 months [95% CI 1.6–3.1], *p* = 0.002).
TP63, EPHA10, FBN2, and IL1RAP genes	They are associated with the lung cancer histotype, while the best-performing AI radiogenomic rule predicting the outcome resulted in an AUC of 0.87.
EGFR, ALK, and KRAS mutations	Predictive modeling of survival duration using both clinical and radiomic features yielded AUCs of 0.977, 0.905, and 0.947 for the EGFR, ALK, and KRAS mutation-positive patient groups, respectively.
PD-L1 expression	In the automatic analysis of the immunohistochemical expression of PD-L1 in lung cancer patients, AI achieved an accuracy and specificity of 96.4 and 96.8% respectively.
Mesenchymal-epithelial transition (MET) C-ros oncogene 1 (ROS1) Tumor protein p53 (TP53)	The prognostic effects of MET, ROS1, and TP53 alterations in resected NSCLC remains debatable.
Pro-gastrin-releasing peptide (ProGRP) Vascular endothelial growth factor (VEGF) Carcinoembryonic antigen (CEA) CYFRA21-1 (a cytokeratin-19 fragment) Neuroson-specific enolase (NSE)	A combination of these five different biomarkers was used for the diagnosis of lung cancer using three different AI methods, with the highest sensitivity of 94.12% and AUCs of 0.908, 0.910, and 0.849 for the three methods, respectively.
Alpha-1-antichymotrypsin (SERPINA3) Ig kappa light chain Serum amyloid A-1 protein (SAA1)	AI classification yielded an AUC = 0.935 for the differentiation of adenocarcinoma vs. COPD and an AUC = 0.916 for the differentiation of adenocarcinoma with COPD vs. COPD.
Anti-cytotoxic T-lymphocyte-associated protein 4 (CTLA-4) antibodies	CTLA-4 is associated with progression-free survival and overall survival benefit with either immunotherapy alone or in combination with chemotherapy when compared to chemotherapy alone.
Phosphoproteins	Phosphoproteins predicted the most and least sensitive quartiles of drug sensitivity with AUC values of 0.79 and 0.78, respectively.
PRMT6	PRMT6 is associated with poor lung cancer prognosis.
CYFRA	CYFRA is associated with non-response in NSCLC patients treated with immunotherapy with a specificity > 95%.
Urinary concentrations of 4-methoxyphenylacetic acid (4MPLA)	Relative urinary concentrations of 4MPLA were significantly different between NSCLC patients and healthy controls (*p* = 0.008), while the ROC analysis of 4MPLA yielded a sensitivity of 82.1%, specificity of 88.2%, and AUC = 0.85.
Carcinoembryonic antigen (CEA)	CEA was associated with lymphovascular invasion which, in turn, predicted lower relapse-free survival and overall survival.
CEA levels in pleural fluid CEA levels in serum CEA pleural fluid/serum ratio	For the differentiation between benign and malignant pleural effusion they achieved AUCs of 0.726, 0.699, and 0.692 in the training cohort; 0.763, 0.695, and 0.731 in the validation cohort; and 0.722, 0.729, and 0.693 in the testing cohort, respectively.
Routine blood indices, such as whole blood count (WBC), creatine phosphokinase (CPK), creatine kinase (CK) isoenzymes, alkaline phosphatase (ALP), lactate dehydrogenase (LDH), albumin, albumin/globulin ratio	Lung cancer patients could be identified from other patients with a sensitivity, specificity, and total accuracy of 96.3%, 94.97%, and 95.7% in the cross-validation results, respectively.
Gut microbiome	AI identified gut microbiome signatures for patients with a favorable immunotherapy response and high PD-L1 expression, with 84% and 79% accuracy, respectively.
Bacteroides dorei and Parabacteroides distasonis	They are associated with longer overall survival.
Clostridium perfringens, Enterococcus faecium, Cortinarius davemallochii, Helotiales, Chaetosphaeriales, and Tremellomycetes	They are associated with shorter overall survival.
Hymenoscyphus immutabilis and Clavulinopsis fusiformis	They are associated with higher PD-L1 expression.
Thelephoraceae and Lachnospiraceae bacterium	They are associated with toxicity to immunotherapy.
Increased relative abundance of guanine, choline, and creatine	They are associated with shorter overall survival.
Increased leucine and tryptophan	They are associated with shorter progression-free survival.
N6,N6,N6-Trimethyl-L-lysine, L-pyrogluatmic acid, and benzoic acid	They were increased in patients who died.
Cystine, methionine sulfoxide, and histamine	They were decreased in patients who died.
Itaconic acid, pyruvate, and malonic acid	They were increased in patients with progression.

**Table 8 cancers-17-00882-t008:** Studies in the sector of biomarkers.

Reference	Type of Study	Participants	AI Status	Main Conclusions
Daneshkhah A, et al., 2023 [21]	Laboratory study	96 patients with lung cancer and 83 with negative LDCT	In vitro diagnostics	Smoking-related molecular abnormalities of the cells in the buccal mucosa, known as “field carcinogenesis,” were used for the detection of lung cancer in early stages through an optical spectroscopic statistical nanosensing AI technique.
Mapanga W, et al., 2021 [23]	Consensus study	27 health stakeholders	Research	Research on inexpensive biomarkers for asymptomatic disease detection is essential for lung cancer management.
Gauthier MP, et al., 2022 [26]	Retrospective observational study	333 patients with lung cancer	In vitro diagnostics	Automated data abstraction from electronic health records is highly accurate and faster than manual abstraction.
Duan S, et al., 2020 [48]	Benchmarking study	372 lung cancer and 470 benign lung patients	In vitro diagnostics	A three-layer diagnosis system for lung cancer based on 14 epidemiological data and clinical symptoms, 5 tumor biomarkers, and 22 CT nodule-based radiomic features.
Bahado-Singh R, et al., 2022 [79]	Laboratory study	10 lung cancer and 20 controls	Research	Using AI and DNA methylation analysis of ctDNA, high lung cancer detection rates were achieved.
Kirienko M, et al., 2022 [196]	Retrospective single-center observational study	151 surgically treated patients with NSCLC	Research	Radiogenomic data may provide clinically relevant information in NSCLC patients regarding the histotype, aggressiveness, and progression, while gene expression analysis showed potential new biomarkers and targets which are valuable for patient management and treatment.
Dora D, et al., 2022 [212]	Cohort study	129 advanced stage NSCLC patients	Research	CT-based texture analysis and gut microbiome signatures can predict overall survival, response to therapy, PD-L1 expression, and toxicity in NSCLC patients treated with ICI.
Tonneau M, et al., 2022 [213]	Benchmarking study	642 advanced NSCLC patients	In vitro diagnostics	A risk prediction model combining clinical and AI–Radiomics was generalizable following CT scan harmonization and AI generalization methods, with similar performances to routine oncology practice in predicting ICI response.
Trebeschi S, et al., 2022 [214]	Benchmarking study	1055 primary and meta- lesions from 203 + 262 NSCLC patients	Research	Radiographic characteristics of lesions on standard-of-care imaging may function as non-invasive biomarkers for response to immunotherapy, and may show utility for improved patient stratification in both neoadjuvant and palliative settings.
Jin W, et al., 2022 [215]	Benchmarking study	143 primary lung cancer patients	Research	The AI model predicts the response of lesions to PD-1 blockade, which implies the importance of introducing the non-linear dynamic effects and AI approaches to the original delta-radiomics in the future.
Park C, et al., 2022 [222]	Benchmarking study	512 patients with NSCLC	Research	In this AI model, CT radiomics and predicted tumor-infiltrated lymphocytes were significantly associated with immune checkpoint inhibitors’ outcomes in NSCLC patients.
Benedum CM, et al., 2023 [270]	Observational study	186,313 patients with lung cancer	In vitro diagnostics	A biomarker-defined cohort was utilized in order to perform analyses related to biomarker-associated survival.
Lococo F, et al., 2023 [275]	Protocol of a multi-center observational clinical trial	600 lung cancer patients	Research	The main goals aim of LANTERN project are: (i) To develop predictive models for lung cancer diagnosis and histological characterization; (ii) to set up personalized predictive models for individual-specific treatments; (iii) to enable feedback data loops for preventive healthcare strategies and quality of life management.
Park MK, et al., 2022 [277]	Benchmarking study	1122 and 763 cancer and non-cancer patients	Research	Gene ontology enrichment and pathway analysis of features revealed that two major sub-types of NSCLC—lung adenocarcinoma and lung squamous cell carcinoma—have both specific and common biological processes.
Bedard ELR, et al., 2021 [278]	Randomized controlled trial	Consecutively NSCLC patients and matched controls	Research	The ROC analysis of miRNA expression yielded a sensitivity of 96.4% and a specificity of 88.2% for the detection of early-stage NSCLC, with AUC = 0.91.
Su Z, et al., 2023 [279]	Laboratory study	2 human lncRNA-disease associations datasets	In vitro diagnostics	CCDC26 and IFNG-AS1 may be new biomarkers of lung cancer, SNHG3 may be associated with PD-L1 for lung cancer.
Xiao X, et al., 2004 [281]	Benchmarking study	30 lung cancer patients and 51 controls	Research	Serum is a valuable resource for the detection of specific lung cancer biomarkers.
Wu J, et al., 2019 [282]	Benchmarking study	183 and 94 patients with and without lung cancer	Research	Lung cancer can be identified based on the combination of 19 types of routine blood indices, which implies that AI can find the connections between a disease and fundamental blood indices.
Zhang H, et al., 2023 [284]	Benchmarking study	5450 RNA samples	Research	Targeted transcriptome combined with AI are highly useful for the diagnosis and classification of various cancers.
Bracht T, et al., 2022 [285]	Benchmarking study	176 patients with adenocarcinoma	Research	Plasma proteomics enable differentiation of lung adenocarcinoma and COPD.
Wei TT, et al., 2023 [286]	Randomized controlled trial	2352 patients with a new pleural effusion	In vitro diagnostics	The development of a machine learning model utilizing routine laboratory biomarkers significantly enhances the diagnostic capability for distinguishing between malignant and benign pleural effusion.
Cheng G, et al., 2022 [289]	Benchmarking study	1288 patients with lung cancer	Research	AI-assisted diagnostic models in PD-L1 expression are a promising tool for improving the efficiency of clinical pathologists.
Chen BT, et al., 2021 [290]	Benchmarking study	110 patients with NSCLC	Research	Predictive modeling of survival duration using both clinical and radiomic features yielded areas under the receiver operative characteristic curve of 0.977, 0.905, and 0.947 for the EGFR, ALK, and KRAS mutation-positive groups, respectively.
Wang X, et al., 2021 [291]	Benchmarking study	110 patients with NSCLC	Research	AI method to critically evaluate PD-L1 expression can empower deep learning applications in digital pathology and facilitate computer-aided diagnosis.
Zuo Z, et al., 2023 [292]	Cohort study	175 patients with clinical T1 stage NSCLC	In vitro diagnostics	A high-risk AI score is a diagnostic biomarker for lymphovascular invasion in patients with clinical T1 stage NSCLC, and can serve as a prognostic biomarker for them.
Prelaj A, et al., 2023 [295]	Multi-center, retrospective and prospective, observational study protocol	15 references	Research	The intention for biomarker-based AI algorithms to be used in other aspects of lung cancer management, such as personalized predictive models, feedback data loops for preventive healthcare strategies, individualized treatments, toxicity prevention (or, at least, minimization), quality of life improvement, efficient resource allocation, and cost-effectiveness.
Park S, et al., 2022 [300]	Benchmarking study	3166 whole-slide images	Research	The inflamed immune phenotype is associated with better response to immunotherapy.
Baxi V, et al., 2022 [301]	Benchmarking study	1746 whole-slide images	Research	High PD-L1 expression via AI evaluation is associated with better response to immunotherapy.
van Delft FA, et al., 2023 [302]	Benchmarking study	412 lung cancer cases	Research	Serum tumor markers, such as CYFRA and CEA, predicted poor response with very high specificity.
Kang Y, et al., 2022 [303]	Laboratory study	48 tissue samples from 32 cancer patients	In vitro diagnostics	A computational framework that uses advanced deep neural network modeling and feature elimination to reduce data on ∼26,000 genes to 6 that accurately predict the response to immunotherapy was developed.
Prelaj A, et al., 2022 [304]	Prospective observational study	164 patients with NSCLC	Research	The integration of multi-factorial data provided by AI techniques is a useful tool to select NSCLC patients as candidates for immunotherapy.
Coker EA, et al., 2022 [305]	Benchmarking study	35 NSCLC cell lines and 16 samples of NSCLC cells	Research	Phosphoproteins could predict response to several targeted anticancer drugs and their hundreds of combinations.
Shoji F, et al., 2022 [306]	Prospective study protocol	400 patients with lung cancer	Research	A rather unexpected biomarker that could predict the response to immunotherapy is the intestinal microbiome.
Mahmoud AM, et al., 2023 [309]	Benchmarking study	33 lung cancer tissue sections	Research	An AI algorithm for scoring PRMT6 expression in lung cancer that matches the degree of accuracy of scoring by pathologists was developed.
Miller HA, et al., 2022 [310]	Benchmarking study	Tumor core biopsies from 44 patients with lung cancer	Research	Various metabolic biomarkers, such as guanine, choline, creatine, leucine, tryptophan, N6,N6,N6-Trimethyl-L-lysine, L-pyrogluatmic acid, benzoic acid, cystine, methionine sulfoxide, histamine, itaconic acid, pyruvate, and malonic acid, have been associated with different outcomes.

**Table 9 cancers-17-00882-t009:** Studies in the sector of biopsy.

Reference	Type of Study	Participants	AI Status	Main Conclusions
Ozcelik N, et al., 2020 [9]	Benchmarking study	345 lymph nodes from 345 EBUS images	Research	The proposed AI approach could be useful in making diagnoses based on images of the mediastinal lymph nodes.
Ye M, et al., 2022 [20]	Benchmarking study	728 participants	Research	A model integrating clinical characteristics, radiological characteristics of pulmonary nodules, AI analysis of LDCT data, and liquid biopsy achieved the best diagnostic performance.
Zhang Y, et al., 2023 [50]	Observational study	4395 participants	Research	AI, circulating molecular biomarkers, and autofluorescence bronchoscopy were applied during LDCT screening.
Yang H, et al., 2022 [78]	Benchmarking study	93 patients with pulmonary nodules	In vitro diagnostics	Circulating genetically abnormal cell had a higher diagnostic value than traditional tumor markers in early-stage lung cancer and a supportive value for AI in the diagnosis of cancer based on lung nodules.
Tan Q, et al., 2021 [99]	Benchmarking study	167 patients with NSCLC	Research	The density of CD45RO⁺ TIL and the combination of CD45RO⁺ TIL and PD-L1 in tumor areas significantly correlated with clinicopathological features and prognosis of NSCLC, and can be used as a novel marker.
Kirienko M, et al., 2021 [196]	Retrospective single-center observational study	151 surgically treated patients with NSCLC	Research	Radiogenomic data may provide clinically relevant information in NSCLC patients regarding the histotype, aggressiveness, and progression, while gene expression analysis showed potential new biomarkers and targets which are valuable for patient management and treatment.
Lococo F, et al., 2023 [275]	Protocol of a multi-center observational clinical trial	600 lung cancer patients	Research	The main goals aim of LANTERN project are: (i) To develop predictive models for lung cancer diagnosis and histological characterization; (ii) to set up personalized predictive models for individual-specific treatments; (iii) to enable feedback data loops for preventive healthcare strategies and quality of life management.
Cheng G, et al., 2022 [289]	Benchmarking study	1288 patients with lung cancer	Research	AI-assisted diagnostic models for PD-L1 expression are a promising tool for improving the efficiency of clinical pathologists.
Wang X, et al., 2021 [291]	Benchmarking study	110 patients with NSCLC	Research	AI method to critically evaluate PD-L1 expression can empower the application of deep learning in digital pathology and facilitate computer-aided diagnosis.
Baxi V, et al., 2022 [301]	Benchmarking study	1746 whole-slide images	Research	AI-powered quantification of PD-L1 expression on tumor cells identified more PD-L1-positive samples, when compared with manual scoring.
Mahmoud AM, et al., 2023 [309]	Benchmarking study	33 lung cancer tissue sections	Research	An AI algorithm for scoring PRMT6 expression in lung cancer that matches the degree of accuracy of scoring by pathologists was developed.
Zhang X, et al., 2021 [331]	Comparative study	36 lung slices of SCLC and 12 normal lung slices	Research	AI was used to analyze the pathology of hesperetin-derived small cell lung cancer under fiberoptic bronchoscopy and achieved a diagnostic accuracy rate of 97.9%.
Gruionu LG, et al., 2022 [332]	Laboratory study	An anatomically accurate lung airway model	Research	Fiber-Bragg grating catheter and AI is a viable shape-sensing hardware and software navigation solution for flexible medical instruments to reach the peripheral airways.
Yu KL, et al., 2023 [333]	Multi-center benchmarking study	1161 EBUS images	Research	Sonographic images from EBUS have been used along with AI for differentiating between benign and malignant tumors with an AUC of 0.88.
Churchill IF, et al., 2022 [334]	Benchmarking study	298 lymph nodes from 140 patients	In vitro diagnostics	Sonographic images from EBUS have been used along with AI for differentiating between benign and malignant tumors and predicting nodal metastasis.
Maruyama S, et al., 2023 [335]	Benchmarking study	4 cancer cell lines	In vitro diagnostics	In AI-based cell detection and classification, attention should be paid to cells whose morphologies change significantly depending on the processing technique.
Teramoto A, et al., 2021 [336]	Benchmarking study	63 patients with lung cancer	In vitro diagnostics	The stain translation technique has potential applications in the massive training of AI systems for cell classification.
Ishii S, et al., 2022 [337]	Benchmarking study	106 and 32 cancer(+) and cancer(−) cytological samples	Research	The implementation of AI in cytological specimens containing lung cancer cells yielded an accuracy and precision of 0.945 and 0.991, respectively, in diagnosing lung cancer.
Tsukamoto T, et al., 2022 [338]	Benchmarking study	55 cases of lung cancer	In vitro diagnostics	The implementation of AI in cytological specimens containing lung cancer cells achieved an accuracy in differentiating between lung cancer sub-types between 66.8% and 76.8%.
Li J, et al., 2023 [340]	Benchmarking study	A dataset of frozen lung cancer tissues	Research	AI makes pathological image classification more accurate, transparent, and convincing.
Al-Jabbar M, et al., 2023 [341]	Benchmarking study	Histological images of the LC25000 dataset	In vitro diagnostics	With the fusion and handcrafted features, the AI reached a sensitivity of 99.85%, a precision of 100%, an accuracy of 99.64%, a specificity of 100%, and an AUC of 99.86%.
Hage Chehade A, et al., 2022 [342]	Benchmarking study	Histological images of the LC25000 dataset	In vitro diagnostics	The AI model gave the best performance in cancer classification, with an accuracy of 99%.
Masud M, et al., 2021 [343]	Benchmarking study	25,000 color-images	Research	The acquired results show that the proposed framework can identify cancer tissues with a maximum accuracy of 96.33%.
Toğaçar M. 2021 [344]	Benchmarking study	25,000 histopathological images	In vitro diagnostics	The overall accuracy rate obtained in the classification process of the histopathological images was 99.69%.
Civit-Masot J, et al., 2022 [345]	Benchmarking study	15,000 histopathological images from lung tissue	Research	Thanks to the given AI report, the time spent by the pathologist and the diagnostic turnaround time can be reduced.
Kriegsmann M, et al., 2020 [346]	Benchmarking study	270 specimens	In vitro diagnostics	Image patch and patient-based AI classification results were 95% and 100% in the test set after the application of strict quality control.
Zhao D, et al., 2020 [347]	Benchmarking study	1284 whole-slide images	In vitro diagnostics	With the development of histopathological AI, transfer learning can effectively train diagnosis models and shorten the learning period, while improving the model performance.
Kanavati F, et al., 2020 [348]	Benchmarking study	3554 whole-slide images	In vitro diagnostics	The AUC values for differentiating between lung carcinoma and non-neoplastic on four independent test sets were 0.975, 0.974, 0.988, and 0.981, respectively.
Cao L, et al., 2023 [349]	Benchmarking study	2607 whole-slide images	In vitro diagnostics	This AI method uses end-to-end learning to extract generalized morphological features automatically and identify discriminative histomorphological patterns with AUC values of 0.95–0.97 in test sets.
Sakamoto T, et al., 2022 [350]	Benchmarking study	201 cases of lung cancer	Research	The sensitivity and specificity of tumor segmentation were 97 and 87%, respectively, and the accuracy of nuclei recognition was 99%.
Kayı Cangır A, et al., 2021 [352]	Laboratory study	Paraffin blocks from 3 adenocarcinomas	Research	AI can help in the differentiation between benign and malignant regions in paraffin-embedded tissue blocks of pulmonary adenocarcinoma using micro-CT.
Chen P, et al., 2023 [353]	Laboratory study	98 patients, 162 slides, and 669 regions of interest	Research	AI can detect the progressive increase in atypical epithelial cells and the progressive decrease in lymphocytic cells, along with the progressively increasing cellular intratumor heterogeneity (from normal, to atypical adenomatous hyperplasia, to adenocarcinoma in situ, to minimally invasive adenocarcinoma, and eventually to invasive adenocarcinoma), consistent with the results from tissue-consuming and molecular/immune profiling.
Zhao Y, et al., 2023 [354]	Benchmarking study	523 whole-slide images	In vitro diagnostics	AI can also discriminate between lepidic, papillary, acinar, solid, micropapillary, and cribriform arrangements of invasive non-mucinous lung adenocarcinoma, with an AUC greater than 0.8.
Sha L, et al., 2019 [355]	Benchmarking study	130 NSCLC patients	Research	AI can determine the PD-L1 status of the tumor with an AUC of 0.8.
Choi S, et al., 2022 [356]	Benchmarking study	802 NSCLC whole-slide images	Research	The assistance provided by the AI-powered tumor proportion score analyzer improved the pathologists’ consensus of reading and prediction of the therapeutic response.
Hondelink LM, et al., 2022 [357]	Benchmarking study	199 whole-slide images and 60 cases	Research	A new deep learning-based PD-L1 tumor proportion score algorithm that is able to assess PD-L1 expression in daily routine diagnostic cases as well as pathologists was developed.
Huang Z, et al., 2022 [358]	Benchmarking study	222 cases of lung cancer	In vitro diagnostics	An AI model demonstrated potential in assisting routine diagnosis of NSCLC by pathologists through scoring of PD-L1 expression.
Wang L, et al., 2019 [359]	Benchmarking study	2311 lung cancer patients	Research	AI natural language processors can be used for the interpretation of histopathological reports.
Li J, et al., 2022 [360]	Benchmarking study	139 individuals with pulmonary nodules	Research	Genetic sequence abnormalities of circulating blood cells, detected by AI, can be used in the discrimination between benign and malignant lung nodules with AUC larger than 0.8.
Waissengrin B, et al., 2023 [361]	Case report	1 NSCLC patient	In vitro diagnostics	AI molecular analysis tool assisted in rapid treatment decision making in the lung cancer context.
Tan X, et al., 2022 [362]	Benchmarking study	2553 NSCLC patients	In vitro diagnostics	AI based on robust variables could help clinicians to identify EGFR mutations and ALK rearrangement status, providing vital guidance regarding TKI selection for targeted therapy in NSCLC patients.
Sorin M, et al., 2023 [363]	Laboratory study	Samples from 416 patients with lung cancer	In vitro diagnostics	Imaging mass cytometry enabled the spatial analysis of immune lineages and activation states through AI, with distinct clinical correlates, using a single 1 mm^2^ tumor core.
Daniel N, et al., 2023 [365]	Laboratory study	Immunohistochemistry lung biopsies	Research	Noise in the correct frequency improves the quality of the synthetic histopathology images for digital pathology.
Saha M, et al., 2021 [366]	Laboratory study	712 tissue specimens of lung cancer	Research	TIL regions in tumor microenvironment were accurately classified by AI through 3D pathology imaging.
Lin YY, et al., 2022 [367]	Laboratory study	33 needle-biopsy-sized specimens	Research	PD-L1 expression, was accurately determined by AI through 3D pathology imaging.
Keller A, et al., 2020 [368]	Laboratory study	210 serum samples from cancer patients	Research	MicroRNAs, which can be obtained through liquid biopsy and analyzed through AI, might be indicative of lung tumors decades before their clinical appearance.

**Table 10 cancers-17-00882-t010:** Studies in the sector of staging.

Reference	Type of Study	Participants	AI Status	Main Conclusions
Ozcelik N, et al., 2020 [9]	Benchmarking study	345 lymph nodes from 345 EBUS images	Research	The proposed AI approach could be useful in making diagnoses based on images of the mediastinal lymph nodes.
Daneshkhah A, et al., 2023 [21]	Laboratory study	96 patients with lung cancer and 83 with negative LDCT	In vitro diagnostics	Smoking-related molecular abnormalities of the cells in the buccal mucosa, known as “field carcinogenesis,” were used for the detection of lung cancer in early stages through an optical spectroscopic statistical nanosensing AI technique.
Gauthier MP, et al., 2022 [26]	Retrospective observational study	333 patients with lung cancer	In vitro diagnostics	Automated data abstraction from electronic health records is highly accurate and faster than manual abstraction.
Zhao K, et al., 2022 [63]	Prospective observational study	141 patients	In vitro diagnostics	The measured diameters of pulmonary lesions and lymph nodes in AI images of ultra-low dose CT are highly close to those of contrast-enhanced CT.
Lyu Q, et al., 2023 [65]	Benchmarking study	62 LDCT and 76 brain MRI reports	Research	It is feasible to utilize large language models in clinical education, and further efforts are needed to address limitations and maximize their potential.
Ruggirello M, et al., 2023 [86]	Retrospective observational study	6495 heavy smokers	In vitro diagnostics	The lower coronary artery calcification burden observed in women at all ages might contribute to explaining their lower rates of all-cause mortality and better lung cancer survival.
Li K, et al., 2022 [176]	Benchmarking study	486 consecutive resected lung lesions	In vitro diagnostics	The accuracy of AI for risk prediction was very high for primary lung cancers, but poor for metastases and benign lesions.
Ozcelik N, et al., 2022 [180]	Benchmarking study	408 CT images from 64 patients with pleural effusion	In vitro diagnostics	Advances in AI-aided diagnostic analysis of CT images and obtaining a pre-diagnosis of pleural fluid may reduce the need for interventional procedures through guiding physicians regarding which patients may have malignancies.
Zhang Y, et al., 2022 [201]	Case report	A 57-year-old woman	In vitro diagnostics	This medical imaging material can help to recognize lung cancer metastasis and has good value for the popularization of science.
Iuga AI, et al., 2022 [202]	Benchmarking study	89 contrast-enhanced chest CTs	Research	The proposed AI approach for automatic classification of thoracic lymph node levels in chest CT, as well as the proof-of-principle experiment for automatic N-staging, revealed promising results.
Geng M, et al., 2022 [203]	Benchmarking study	48 patients with lung cancer and 34 controls	Research	The nomogram model based on CT image features and related clinical indicators was an effective method for non-invasive prediction of lymph node metastasis in lung cancer.
Shimada Y, et al., 2022 [204]	Benchmarking study	720 patients with resected stage 0-IA NSCLC	In vitro diagnostics	Measuring the average solid-CT value of tumors for pathological lymph node metastasis may have broad applications, such as guiding individualized surgical approaches and postoperative treatment.
Lee KH, et al., 2022 [205]	Benchmarking study	1426 patients with stage I–IV lung cancer	Research	The CT-based composite score obtained from AI of histopathological features predicted survival in early-stage lung adenocarcinomas with high reproducibility.
Huo T, et al., 2022 [206]	Benchmarking study	126 patients with lung cancer	Research	The proposed AI model for automatic lung cancer bone metastasis detection can improve diagnostic efficiency and reduce the diagnosis time and workload of junior radiologists.
Trebeschi S, et al., 2022 [221]	Benchmarking study	152 stage-IV NSCLC patients	Research	AI-derived survival score was independent of clinical, radiological, PD-L1, and histopathological factors, and visual analysis of AI-generated prognostic heatmaps revealed the relative prognostic importance of morphological nodal changes in the mediastinum, supraclavicular, and hilar regions, lung and bone metastases, as well as pleural effusions, atelectasis, and consolidations.
Pfaehler E, et al., 2021 [252]	Benchmarking study	124 PET/CT images of patients with NSCLC	Research	Both AI algorithms lead to accurate segmentations for both primary tumors as well as metastasis and are, therefore, good candidates for PET tumor segmentation.
Borrelli P, et al., 2022 [254]	Benchmarking study	115 patients undergone FDG PET/CT	In vitro diagnostics	Both manual and AI total lesion glycolysis (TLG) were significantly associated with overall survival; however, low sensitivity makes this AI model less useful in supporting clinical reading, reporting, and staging.
Borrelli P, et al., 2022 [256]	Benchmarking study	320 patients referred for FDG PET/CT due to suspected lung cancer	Research	AI was trained to segment lung tumors and thoracic lymph nodes and fully automated AI-based total lesion glycolysis (TLG) measurements were significantly associated with overall survival in patients with lung cancer.
Rogasch JMM, et al., 2023 [258]	Benchmarking study	491 consecutive patients with NSCLC	Research	An AI model based on routinely available variables from [^18^F]FDG-PET/CT improved accuracy in mediastinal staging, compared to established visual assessment criteria.
Joshi S, et al., 2022 [262]	Benchmarking study	99 patients with NSCLC	In vitro diagnostics	This detector can identify the stage of a lung tumor with an average accuracy of 97%.
Alshmrani GM, et al., 2023 [263]	Benchmarking study	A dataset of 51 PET/CT studies	Research	The segmentation of lung tumors, treatment evaluation, and tumor stage classification have become significantly more accessible with the advent of PET/CT scans.
Wallis D, et al., 2022 [264]	Benchmarking study	205 patients with lung cancer	Research	This study went directly from whole-body [^18^F]FDG-PET/CT scans to pathological mediastinal lymph node localization, with a sensitivity of 0.88.
Tau N, et al., 2020 [265]	Benchmarking study	264 patients with NSCLC	Research	An AI to analyze segmented PET images of patients with NSCLC can yield moderately high accuracy for designation of N category, although this may be insufficient to preclude invasive lymph node sampling.
Weikert T, et al., 2023 [266]	Benchmarking study	364 FDG-PET/CTs of patients with lung cancer	In vitro diagnostics	TNM categorization based on the anatomical region approach was correct in 94.3% of lesions.
Rogasch JMM, et al., 2023 [267]	Laboratory study	13 questions regarding [^18^F]FDG PET/CT	In vitro diagnostics	ChatGPT might adequately substitute for advice given to patients by nuclear medicine staff in the investigated settings.
Kasinathan G, et al., 2022 [268]	Benchmarking study	94 patients with NSCLC	In vitro diagnostics	For complete TNM evaluation, PET/CT accuracy varied between 97–99.1%.
Wei TT, et al., 2023 [286]	Randomized controlled trial	2352 patients with a new pleural effusion	In vitro diagnostics	The development of a machine learning model utilizing routine laboratory biomarkers significantly enhances the diagnostic capability for distinguishing between malignant and benign pleural effusion.
Chen BT, et al., 2021 [290]	Benchmarking study	110 patients with NSCLC	Research	Radiomic scores based on MR imaging of brain metastases from NSCLC can be used as non-invasive biomarkers for survival duration.
Zuo Z, et al., 2023 [292]	Cohort study	175 patients with clinical T1 stage NSCLC	In vitro diagnostics	A high-risk AI score is a diagnostic biomarker for lymphovascular invasion in patients with clinical T1 stage NSCLC, and can serve as a prognostic biomarker for them.
Churchill IF, et al., 2022 [334]	Benchmarking study	298 lymph nodes from 140 patients	In vitro diagnostics	Sonographic images from EBUS have been used along with AI for differentiating between benign and malignant tumors and predicting nodal metastasis.
Wang L, et al., 2019 [359]	Benchmarking study	2311 lung cancer patients	Research	AI can manage a vast quantity of information regarding patients’ clinical features (including staging), reproduce them through natural language processing approaches, and use them in various clinical and research projects.
Liu Y, et al., 2021 [371]	Benchmarking study	3352 patients with malignancy	Research	This AI model demonstrated an AUC of 0.870 for lung cancer in identifying bone metastasis in bone scintigraphy.
Zhao Z, et al., 2020 [372]	Benchmarking study	12,222 cases of bone scintigraphy	Research	This AI model demonstrated an AUC of 0.957 for lung cancer in identifying bone metastasis in bone scintigraphy.
Meißner AK, et al., 2023 [374]	Benchmarking study	53 NSCLC patients with brain metastasis	Research	In patients with brain metastasis, the AUC for PD-L1 expression was calculated between 0.77 and 0.84, depending on the parameters included in its prediction.
Tanaka Y, et al., 2020 [375]	Laboratory study	3 NSCLC cell lines with 19,849 genes	Research	AI models predicted the process of metastasis using a Bayesian network applied to the Epithelial–Mesenchymal Transition data set.
Li Z, et al., 2023 [376]	Benchmarking study	4466 NSCLC patients with brain metastasis	In vitro diagnostics	AI model predicted the development of brain metastasis, with an AUC at 0.825, using various information from the patients’ electronic health records.
Huang Z, et al., 2020 [377]	Benchmarking study	5973 cases of NSCLC	In vitro diagnostics	Many AI prognostic models use information related to bone metastases, or TNM stage in general.
Yang Y, et al., 2022 [378]	Benchmarking study	998 patients with lung cancer	Research	One of the most important predictors of the AI prognostic model among clinical status was TNM stage.
Kinoshita F, et al., 2023 [379]	Benchmarking study	1049 patients with resected NSCLC	In vitro diagnostics	AI prognostic model showed that the AUC values related to disease-free survival, overall survival, and cancer-specific survival at 5 years were 0.890, 0.926, and 0.960, respectively.

**Table 11 cancers-17-00882-t011:** Studies in the sector of treatment (general information).

Reference	Type of Study	Participants	AI Status	Main Conclusions
Lemos R, et al., 2022 [106]	Prospective observational cohort study protocol	300 patients with breast or lung cancer	Research	If this AI application is effective, it will provide healthcare systems with a novel and innovative method to screen depressive symptoms in oncological settings.
Tang FH, et al., 2023 [108]	Benchmarking study	422 NSCLC patients	Research	The proposed AI model has the potential to improve NSCLC prognosis and facilitate personalized treatment decisions.
van de Worp WRPH, et al., 2022 [237]	Laboratory study	60 unique manually delineated lung tumors	Research	This AI algorithm provides a helpful tool for the non-invasive detection and analysis of tumor take, tumor growth, and therapeutic effects in mouse orthotopic lung cancer models.
Sharif MS, et al., 2010 [253]	Benchmarking study	2 phantom datasets with simulated tumors	Research	Tumor detection, classification, and quantification in PET imaging at early stage of disease are important issues for clinical diagnosis, assessment of response to treatment, and radiotherapy planning.
Benedum CM, et al., 2023 [270]	Observational study	186,313 patients with lung cancer	In vitro diagnostics	A treatment-defined cohort was utilized in order to perform analyses related to the comparative effectiveness of treatment.
Lococo F, et al., 2023 [275]	Protocol of a multi-center observational clinical trial	600 lung cancer patients	Research	The main goals aim of LANTERN project are: (i) To develop predictive models for lung cancer diagnosis and histological characterization; (ii) to set up personalized predictive models for individual-specific treatments; (iii) to enable feedback data loops for preventive healthcare strategies and quality of life management.
Waissengrin B, et al., 2023 [361]	Case report	1 NSCLC patient	In vitro diagnostics	AI molecular analysis tool assisted in rapid treatment decision making in lung cancer.
Lorenc A, et al., 2023 [394]	Retrospective observational study	2296 patients with lung cancer	In vitro diagnostics	AI demonstrated a high predictive ability for recurrence, varying between 87.5% and 89.9%.
Chang L, et al., 2022 [395]	Laboratory study	NSCLC drug datasets	Research	AI-assisted medical system proposes a drug selection framework for the personalized selection of NSCLC patients which, under the premise of ensuring the efficacy, predicts the drug effectiveness–cost trade-off.
Liu Z, et al., 2020 [397]	Benchmarking study	1206 clinical records of patients diagnosed with NSCLC	Research	With the aid of entity-level representation, data augmentation, and model fusion, deep learning-based multi-label classification approaches can better imitate traditional Chinese medicine syndrome differentiation in complex cases such as advanced lung cancer.
Araki K, et al., 2023 [398]	Benchmarking study	Electronic health data from 7 hospitals in Japan	Research	AI can assist in the extraction of precise medical data from vast databases, especially for patients in different lines of treatment.
Cesario A, et al., 2021 [399]	Laboratory study	Databases with patients with breast or lung cancer	Research	AI algorithm helps in the matchmaking between patient’s data and the inclusion criteria of clinical trials to personalize patient enrollment.
Field M, et al., 2021 [400]	Retrospective analysis of the data of a clinical trial	482 patients with lung cancer who received radiotherapy	Research	External validation of a 2-year overall survival model for NSCLC gave an AUC of 0.65 and C-index of 0.62 across the network; AI may assist in recruiting suitable patients for clinical trials of novel radiotherapy methods.
Prelaj A, et al., 2024 [401]	Protocol of a multi-center observational study	Patients with advanced lung cancer from different Italian centers	Research	The overall aim of the project is to build a consortium integrating different datasets and a virtual biobank from participating Italian lung cancer centers.
Janopaul-Naylor JR, et al., 2024 [402]	Laboratory study	117 questions about lung cancer presented to 2 AI chatbots	Research	AI chatbots provide multiple opportunities for innovating health care, particularly around cancer prediction, in order to limit misleading counseling, confusion, and emotional distress to patients and families.
Chebanov DK, et al., 2023 [404]	Laboratory study	Gene expression profiles in lung cancer patients	Research	AI culminated in the derivation of several small-molecule formulas projected to bind selectively to specific proteins.
Grisoni F, et al., 2019 [405]	Laboratory study	14 peptides from 1000 de novo designs	Research	AI can assist in the de novo design of anticancer peptides.
Mistro M, et al., 2020 [412]	Benchmarking study	6 lung/mediastinum radiotherapy patients	Research	This knowledge model-based training program brought unexperienced planners to a level close to experienced planners in fewer than 2 days.
Zhou N, et al., 2019 [413]	Retrospective observational study	362 patients with cancer	In vitro diagnostics	Incidence and pharmaceuticals may be the major causes of discordance between AI and clinicians.
Kim MS, et al., 2020 [414]	Retrospective observational study	405 patients with lung cancer	In vitro diagnostics	Treatment recommendations made by AI and physicians were highly concordant for lung cancer cases, especially in metastatic stage; however, AI was only an assisting tool in stage I–III NSCLC and limited disease SCLC.
Yao S, et al., 2020 [415]	Retrospective comparative study	165 patients with advanced NSCLC	In vitro diagnostics	Most of the treatment recommendations of AI were consistent with the real-world treatment, while patient preferences, prices, drug approval and medical insurance ultimately affect the inconsistency.
Liu C, et al., 2018 [416]	Retrospective observational study	182 patients with lung cancer	In vitro diagnostics	Most of the treatment recommendations of AI were consistent with those of the expert group, although a relatively high proportion of cases were still not supported; therefore, AI cannot currently replace oncologists.
You HS, et al., 2020 [417]	Benchmarking study	310 patients with metastatic NSCLC	In vitro diagnostics	AI recommendations had high applicability for metastatic NSCLC patients.
Hesso I, et al., 2023 [420]	Survey	95 healthcare professionals	Research	Healthcare professionals reported favorable opinions of AI-based cancer imaging technologies and noted a number of care pathway concerns where AI can be useful.
Ankolekar A, et al., 2022 [421]	Retrospective cohort study	257 NSCLC patients and 9 lung cancer specialists	In vitro diagnostics	AI can support shared decision-making through presenting the harms and benefits of different treatment options, rather than giving single treatment advice.

**Table 12 cancers-17-00882-t012:** Studies in the sector of surgical treatment.

Reference	Type of Study	Participants	AI Status	Main Conclusions
Zhou H, et al., 2021 [19]	Prospective, single-arm, multi-center, phase II trial	40 patients with lung cancer	Research	This study indicates whether a new multi-omic AI system constitutes a valid strategy for neoadjuvant therapy effect prediction and decision-making assistance in the context of neoadjuvant treatment for patients with stage IB–IIIA NSCLC.
Lafrenière M, et al., 2023 [127]	Benchmarking study	271 images from 129 patients	In vitro diagnostics	An AI model can distinguish the features of trackable and untrackable lesions in digital radiographs reconstructed from the simulation CT scan, and can predict successful candidates for fiducial-free lung tumor tracking.
Elia S, et al., 2023 [173]	Retrospective observational study	71 patients with solitary pulmonary nodules	Research	AI analysis of radiomic features may be applied to the decision-making process in elderly frail patients with suspicious nodules, in order to minimize the false positive rate and reduce the incidence of unnecessary surgery.
Lv Y, et al., 2022 [200]	Benchmarking study	901 patients with early-stage resected NSCLC	Research	This AI model achieved comparable performance to intra-operative frozen section analysis in determining tumor invasiveness, and may contribute to clinical decisions related to the extent of surgical resection.
Zhang DY, et al., 2022 [217]	Case report	1 patient with multiple primary lung cancer	In vitro diagnostics	The surgical plan for multiple pulmonary nodules should be carefully considered, and AI might be a helpful tool in this regard.
Chen X, et al., 2022 [228]	Retrospective observational study	33,410 nodules among 2126 participants	Research	The AI-detected total nodule number is significantly associated with survival rates in patients with surgically resected stage III NSCLC.
Sadeghi AH, et al., 2022 [233]	Prospective observational pilot study	10 patients referred for segmentectomy	In vitro diagnostics	The successful development and clinical application of the first dedicated AI and virtual reality platform for the planning of pulmonary segmentectomy is demonstrated.
Chen X, et al., 2022 [234]	Benchmarking study	27 cases with lung cancer	Research	The AI segmentation algorithm achieves the recognition of the segmental pulmonary artery and the lobular pulmonary vein, with efficiency approximately equal to that of junior thoracic surgery attendings, being potentially beneficial for lung cancer patients.
Wang L, et al., 2019 [359]	Benchmarking study	2311 lung cancer patients	Research	AI can assist with data extraction related to therapy, including surgical data, from large databases with an accuracy and precision of 100%.
Sorin M, et al., 2023 [363]	Laboratory study	Samples from 416 patients with lung cancer	In vitro diagnostics	Using deep learning, patients who will progress after surgery could be predicted using a single 1-mm^2^ tumor core, which could be informative for clinical management following surgical resection.
Huang Z, et al., 2020 [377]	Benchmarking study	5973 cases of NSCLC	In vitro diagnostics	The kind of surgical treatment has been related, by AI, to the subsequent appearance of bone metastasis and survival.
Xu H, et al., 2023 [423]	Benchmarking study	100 Chest CTs	In vitro diagnostics	An atlas for lobectomy and segmentectomy at the sub-segmental or more distal level, where the recognition accuracy of pulmonary arteries in a non-time-sensitive experimental scenario was still unfavorable.
Li X, et al., 2019 [425]	Benchmarking study	53 patients with multiple pulmonary nodules	In vitro diagnostics	AI may represent a relevant diagnostic aid that provides more accurate and objective results for the diagnosis of multiple pulmonary nodules, reducing the time required for interpretation of results, offering plans for follow-up and treatment.
Liu HC, et al., 2023 [427]	Benchmarking study	12 patients underwent thoracic surgery	In vitro diagnostics	The system is designed to mark the lesion’s location on the image smartly and perform tumor grading in real-time, potentially facilitating clinical decision making, with an overall accuracy of 84.9%.
Kudo Y, et al., 2022 [428]	Cohort study	157 patients who had clinical N0 NSCLC	In vitro diagnostics	SUV_max_ values in PET scans and abnormal CEA levels are unfavorable prognostic factors, and might suggest that lobectomy should be preferred to segmentectomy.
Wang X, et al., 2023 [429]	Protocol for a retrospective observational study	All NSCLC patients during an 8-year period from 30 large Chinese centers	Research	Study protocol for new research about the effectiveness of surgical treatment and neo-adjuvant chemotherapy, with the help of AI, in clinical stages I–IIIA.

**Table 13 cancers-17-00882-t013:** Studies in the sector of radiotherapy.

Reference	Type of Study	Participants	AI Status	Main Conclusions
Zhang D, et al., 2023 [109]	Benchmarking study	21,093 patients’ clinical data	In vitro diagnostics	The interpretable AI survival predictive tool for SCLC using clinical data, including radiotherapy, had a reliable predictive value regarding overall survival.
Hooshangnejad H, et al., 2022 [218]	Virtual clinical trial	Data from 15 patients with NSCLC	In vitro diagnostics	AI reduces the common online adaptive radiation therapy course by at least two weeks, resulting in a 50% shorter time to treatment and a lower chance of restaging and loss of local control.
Kim H, et al., 2022 [219]	Retrospective study	135 patients with lung cancer	Research	This study exhibited the external validity and transportability of a CT-based AI prediction model for radiotherapy candidates, predicting local recurrence-free survival, disease-free survival, and overall survival.
Zhang Z, et al., 2022 [220]	Benchmarking study	701 patients with lung cancer	In vitro diagnostics	This novel AI approach combining CT and radiation dose images can effectively and accurately predict the occurrence of radiation pneumonitis, and can be easily adjusted to fit new cohorts.
Gu H, et al., 2022 [232]	Benchmarking study	105 locally advanced NSCLC	Research	The 2D–3D hybrid AI model achieved accurate automatic segmentation of lung lobes on conventional slice-thickness CT of locally advanced lung cancer patients, and exhibits good clinical practicability for radiotherapy.
Hosny A, et al., 2022 [238]	Benchmarking study	2208 patients with lung cancer	In vitro diagnostics	In silico geometric segmentation metrics might not correlate with clinical utility of the models, while experts’ segmentation style and preference might affect model performance.
Sharif MS, et al., 2010 [253]	Benchmarking study	2 phantom datasets with simulated tumors	Research	Tumor detection, classification, and quantification in PET imaging at the early stage of disease are important issues for clinical diagnosis, assessment of response to treatment, and radiotherapy planning.
Kerhet A, et al., 2010 [259]	Benchmarking study	2 patients with NSCLC	Research	The potential applicability of a machine learning methodology as an auxiliary tool for radiation treatment planning in NSCLC was demonstrated.
Wang L, et al., 2019 [359]	Benchmarking study	2311 lung cancer patients	Research	AI can assist with data extraction related to therapy, including radiotherapy, from large databases.
Huang Z, et al., 2020 [377]	Benchmarking study	5973 cases of NSCLC	In vitro diagnostics	Radiotherapy seems to be a significant prognostic factor in AI models for bone metastasis and survival prediction.
Field M, et al., 2021 [400]	Retrospective analysis of the data of a clinical trial	482 patients with lung cancer who received radiotherapy	Research	External validation of a 2-year overall survival model for NSCLC gave an AUC of 0.65 and C-index of 0.62 across the network; AI may assist in recruiting suitable patients for clinical trials of novel radiotherapy methods.
Sun W, et al., 2022 [431]	Retrospective observational study	67 NSCLC patients	In vitro diagnostics	A systematic method to integrate expert human knowledge with AI recommendations for optimizing clinical decision making in patients receiving radiotherapy is described.
Niraula D, et al., 2023 [433]	Benchmarking study	10,000 synthetic patients and 67 NSCLC patients	In vitro diagnostics	Preliminary qualitative results deeply accelerated adaptive radiation therapy through AI, but seem to be modest.
Wolfs CJA, et al., 2020 [434]	Retrospective observational study	59 radiotherapy treatment plans in 47 patients	In vitro diagnostics	Deep learning is a promising powerful tool for identifying types and magnitude of treatment errors with electronic portal imaging device dosimetry, providing additional information which is not currently available.
Wolfs CJA, et al., 2022 [435]	Retrospective observational study	122 radiotherapy treatment plans in 109 patients	In vitro diagnostics	The choice of dose comparison method has the largest impact on error identification for pre-treatment quality assurance using deep learning, compared to image pre-processing.
Kiser K, et al., 2023 [436]	Cohort study	10 patients received radiotherapy	In vitro diagnostics	This AI model seems to effectively deliver stereotactic body radiation therapy, bypassing the mobile target barrier with no local recurrences or acute grade ≥ 3 toxicities after three months.
Katsuta Y, et al., 2022 [438]	Retrospective observational study	153 patients received radiotherapy	In vitro diagnostics	The prediction of radiation pneumonitis by AI through cumulative and differential dose–volume histograms carries an AUC of 0.73.
de Dios N R, et al., 2023 [439]	Benchmarking study	63 patients receiving radiotherapy	Research	For hippocampal avoidance during prophylactic cranial irradiation, the implementation of an AI planning resulted in a 55% reduction in average optimization time, compared to manual plan.
Ladbury C, et al., 2023 [440]	Retrospective analysis of the data of a clinical trial	482 patients with lung cancer who received radiotherapy	Research	The thresholds for pulmonary and esophageal toxicity were identified at a mean dose of >18 Gy and lung volume receiving ≥20 Gy (V20) > 37% and a mean dose > 34 Gy and V20 > 37%, respectively, with no significantly identifiable thresholds for cardiac toxicity.

**Table 14 cancers-17-00882-t014:** Studies in the sector of chemotherapy.

Reference	Type of Study	Participants	AI Status	Main Conclusions
Gu Q, et al., 2022 [15]	Benchmarking study	80 elderly patients with lung cancer	In vitro diagnostics	Methotrexate combined with platinum chemotherapy can improve the chemotherapy effect in elderly patients with lung cancer, with a low degree of adverse reactions and good overall tolerance.
Zhou H, et al., 2021 [19]	Prospective, single-arm, multi-center, phase II trial	40 patients with lung cancer	Research	This study indicates whether a new multi-omics AI system constitutes a valid strategy for neoadjuvant therapy effect prediction and decision-making assistance in the context of neoadjuvant treatment for patients with stage IB–IIIA NSCLC.
Li X, et al., 2019 [107]	Benchmarking study	137 patients with NSCLC	Research	Several potential prognostic biomarkers of PET/CT imaging have been extracted for prediction of survival and selecting patients with NSCLC who are more likely to benefit from chemotherapy.
Zhang D, et al., 2023 [109]	Benchmarking study	21,093 patients’ clinical data	In vitro diagnostics	The interpretable AI survival predictive tool for SCLC using clinical data (including chemotherapy) had a reliable predictive value on their overall survival.
Lee J, et al., 2021 [229]	Retrospective observational study	70 consecutive patients with SQCLC	Research	Patients with advanced SQCLC under palliative chemotherapy lose a significant amount of skeletal muscle, where the reduction of muscle area occurs faster at the end of life.
Ventura D, et al., 2023 [261]	Benchmarking study	44 patients with advanced NSCLC	In vitro diagnostics	This radiomics-based model might be able to predict the response in advanced NSCLC patients treated with CKI-based first-line therapy with or without chemotherapy, including disease progression or death.
Huang Z, et al., 2020 [377]	Benchmarking study	5973 cases of NSCLC	In vitro diagnostics	Chemotherapy has been shown to be a significant prognostic factor for bone metastasis and survival.
Ren S, et al., 2022 [441]	Laboratory study	954 and 477 separate cell lines	Research	AI-based graph regularized matrix factorization is a model that can be used for the de novo prediction of cell drug sensitivities, including chemotherapy.
Nipp RD, et al., 2016 [442]	Randomized controlled trial	191 lung cancer patients	In vitro diagnostics	The use of emotional support and acceptance coping strategies correlated with better quality of life and mood, whereas the use of denial and self-blame negatively correlated with these outcomes in patients with incurable lung cancer under chemotherapy.
Mei C, et al., 2022 [443]	Randomized controlled trial	118 lung cancer patients receiving chemotherapy	Research	The successful management of adverse events (e.g., vomiting) through AI has been shown to significantly improve the psychological state, quality of life, serum tumor marker levels, and nursing satisfaction of patients receiving chemotherapy.
Rozynek M, et al., 2024 [444]	Cohort study	178 patients with NSCLC	In vitro diagnostics	Automatically extracted muscle and adipose tissue radiomic features could be used as predictors of survival in patients with non-small cell lung cancer.

**Table 15 cancers-17-00882-t015:** Studies in the sector of targeted therapy.

Reference	Type of Study	Participants	AI Status	Main Conclusions
Yang R, et al., 2022 [18]	Benchmarking study	7413 patients with lung adenocarcinoma	In vitro diagnostics	AI-based prediction model may become a practical tool to guide EGFR mutation prediction in lung adenocarcinoma.
Wang C, et al., 2022 [101]	Benchmarking study	9136 patients with NSCLC	Research	Both clinical indicators and radiomics features showed a complementary role in prediction and provided accurate estimates to predict EGFR and PD-L1 status.
Zhou G, et al., 2023 [103]	Benchmarking study	164 patients with lung adenocarcinoma	Research	There was a statistical difference between tumors with vascular convergence sign and EGFR gene mutation, while pleural effusion, pericardial effusion, pleural thickening, and other signs in tumor imaging were not significantly associated with EGFR gene mutation.
Chang C, et al., 2021 [104]	Benchmarking study	526 patients with lung adenocarcinoma	In vitro diagnostics	PET/CT radiomics-based AI model has potential to be used as a non-invasive diagnostic method to help diagnose ALK mutation status for lung adenocarcinoma patients in the clinic.
Wang S, et al., 2022 [207]	Benchmarking study	18,232 patients with lung cancer	In vitro diagnostics	This AI system provides a non-invasive method to detect EGFR genotype and identify patients with an EGFR mutation at high risk of TKI resistance.
Silva P, et al., 2022 [208]	Benchmarking study	Chest CT images	Research	This work provides a more accurate and rapid characterization of lung cancer EGFR status, which is fundamental for the development of an adequate treatment plan for lung cancer patients.
Shao J, et al., 2022 [211]	Benchmarking study	1096 NSCLC patients	In vitro diagnostics	This AI system simultaneously analyzed 10 molecular expressions, which might be utilized as an assistive tool in conjunction with or in lieu of ancillary testing to support precision treatment options.
Deng K, et al., 2022 [216]	Retrospective study	570 patients with stage IV EGFR-mutant NSCLC	In vitro diagnostics	This study developed and externally validated a pre-operative CT image-based AI model to predict the survival benefits of EGFR-TKI and ICI therapies in stage IV NSCLC patients, which will facilitate optimized and individualized treatment strategies.
Palumbo B, et al., 2020 [260]	Benchmarking study	78 patients with NSCLC	Research	The AI model showed an ability to predict the correct MIB-1 proliferation group with overall accuracy >82%.
Coker EA, et al., 2022 [305]	Benchmarking study	35 NSCLC cell lines and 16 samples of NSCLC cells	Research	Phosphoproteins could predict response to several targeted anticancer drugs and their hundreds of combinations.
Ishii S, et al., 2022 [337]	Benchmarking study	106 and 32 cancer(+) and cancer(−) cytological samples	Research	AI has been utilized in cytological images to predict the status of certain biomarkers which are critical for targeted therapy, with an accuracy of ~0.95 for the EGFR and KRAS groups, ~0.75 for the ALK group, and ~0.80 for the “None” group.
Tan X, et al., 2022 [362]	Benchmarking study	2553 NSCLC patients	In vitro diagnostics	AI based on robust variables could help clinicians to identify EGFR mutations and ALK rearrangement status and provide vital guidance in TKI selection for targeted therapy in NSCLC patients.
Chen Y, et al., 2021 [445]	Retrospective comparative observational study	115 patient tissue samples	In vitro diagnostics	Compared with manual curation by a bioinformatician, AI analysis provided comprehensive insights and additional genetic alterations to inform clinical therapeutic strategies for patients with lung cancer.
Srisongkram T, et al., 2022 [446]	Laboratory study	67 FDA-approved KRAS^G12C^ inhibitors	Research	AI model evaluated KRAS^G12C^ inhibitors, with an accuracy score of validation = 0.85 and Q^2^_Ext_ = 0.76, predicting that, of 67 Food and Drug Administration (FDA)-covalent drugs, afatinib, dacomitinib, acalabrutinib, neratinib, zanubrutinib, dutasteride, and finasteride would be active inhibitors.
Bui ATN, et al., 2022 [447]	Laboratory study	A library of approved drugs	Research	AI identified octenidine as an inhibitor of B-cell lymphoma-extra-large (Bcl-xL), an anti-apoptotic protein which is an important modulator of the mitochondrial apoptosis pathway.
Fan Q, et al., 2022 [448]	Laboratory study	Lung cancer cells in vitro	Research	AI analysis found that optimizing component formula (OCF)—a molecule extracted from Salvia miltiorrhiza Bunge and Panax ginseng C. A. Meyer—might elicit an anti-lung cancer effect through blocking denticleless E3 ubiquitin protein ligase homolog (DTL)-mediated programmed cell death 4 (PDCD4) ubiquitination and suppression of the mitogen-activated protein kinases (MAPK)/Jun N-terminal kinase (JNK) pathway.
Choi G, et al., 2021 [449]	Laboratory study	Data related to tyrosine kinase inhibitors (TKIs) from accessible libraries	Research	AI methods were used to develop prospective EGFR TKIs that would overcome the limitations of existing third-generation drugs, including the simplified molecular-input line-entry system datasets of the generated ligands, from which drug-like ligands were obtained, via parameter-filtering, cyclic skeleton analysis, and virtual screening analysis.
Park J, et al., 2022 [450]	Laboratory study	965 samples of non-small-cell lung carcinoma	In vitro diagnostics	Among the three immune phenotypes of lung cancer (inflamed, immune-excluded, and immune-desert), it was found that KRAS mutation, BRAF mutation, and MET splicing variant were mostly observed in the inflamed sub-type, while EGFR and PIK3CA mutations were the most prominent mutations found in the immune-excluded sub-type.
Itahashi K, et al., 2018 [451]	Comparative cohort study	198 lung cancer patients	In vitro diagnostics	The most frequently detected single nucleotide variants were found in TP53 (n = 107), BRCA2 (n = 24), and NOTCH2 (n = 23), while MYC (n = 10) was the most frequently detected gene amplification, followed by ERBB2 (n = 9) and CCND1 (n = 6).
Boyarskikh U, et al., 2018 [452]	Laboratory study	Several lung cancer cell lines	In vitro diagnostics	The lung cancer cell lines insensitive to the cytotoxic action of the p53 activator Nutlin-3, due to the mTOR and PI3K pathway, were actually highly sensitive to the dual PI3K/mTOR inhibitor NVP-BEZ235, while they did not respond to either the PI3K-specific LY294002 or the Bcl-XL-specific 2,3-DCPE.
Liu T, et al., 2023 [453]	Laboratory study	Human lung cancer A549 cell line	In vitro diagnostics	Solasonine treatment rescued the Bcl-2/Bax ratio and cleaved caspase-3 expression, suggesting that solasonine induces apoptosis, while simultaneously exhibiting anti-inflammatory effects through inhibiting P38 MAPK.
Vashistha V, et al., 2021 [454]	Retrospective observational study	1749 patients with NSCLC	In vitro diagnostics	Up to 32.1% of lung cancer patients with highly actionable gene variants, such as ALK, BRAF, EGFR, ERBB2, MET, RET, and ROS1, might not receive targeted agents as they do not carry a diagnosis of metastatic disease, or their treating provider did not discuss the AI results or felt that the patient could not tolerate the therapy.

**Table 16 cancers-17-00882-t016:** Studies in the sector of immunotherapy.

Reference	Type of Study	Participants	AI Status	Main Conclusions
Zhou H, et al., 2021 [19]	Prospective, single-arm, multi-center, phase II trial	40 patients with lung cancer	Research	The efficacy of ICIs is influenced by many factors, including patient’s driver genes and smoking status.
Tong H, et al., 2022 [100]	Retrospective multi-cohort study	1405 NSCLC patients	In vitro diagnostics	^18^F-FDG PET/CT radiomics–clinical combined model could be a clinically practical method to non-invasively detect the tumor immune status in NSCLCs.
Wang C, et al., 2022 [101]	Benchmarking study	9136 patients with NSCLC	Research	Both clinical indicators and radiomics features showed a complementary role in prediction and provided accurate estimates to predict EGFR and PD-L1 status.
Shao J, et al., 2022 [211]	Benchmarking study	1096 NSCLC patients	In vitro diagnostics	This AI system simultaneously analyzed 10 molecular expressions, which might be utilized as an assistive tool in conjunction with or in lieu of ancillary testing to support precision treatment options.
Dora D, et al., 2022 [212]	Cohort study	129 advanced stage NSCLC patients	Research	CT-based texture analysis and gut microbiome signatures can predict overall survival, the response to therapy, the PD-L1 expression, and toxicity in NSCLC patients treated with ICI.
Tonneau M, et al., 2022 [213]	Benchmarking study	642 advanced NSCLC patients	In vitro diagnostics	A risk prediction model combining clinical and AI-Radiomics was generalizable following CT scan harmonization and AI generalization methods, with similar performances to routine oncology practice in predicting ICI response.
Trebeschi S, et al., 2022 [214]	Benchmarking study	1055 primary and meta- lesions from 203 + 262 NSCLC patients	Research	Radiographic characteristics of lesions on standard-of-care imaging may function as non-invasive biomarkers for response to immunotherapy, and may show utility for improved patient stratification in both neoadjuvant and palliative settings.
Jin W, et al., 2022 [215]	Benchmarking study	143 primary lung cancer patients	Research	The AI model predicts the response of lesions to PD-1 blockade, which implies the importance of introducing the non-linear dynamic effects and AI approaches to the original delta-radiomics in the future.
Deng K, et al., 2022 [216]	Retrospective study	570 patients with stage IV EGFR-mutant NSCLC	In vitro diagnostics	This study developed and externally validated a pre-operative CT image-based AI model to predict the survival benefits of EGFR-TKI and ICI therapies in stage IV NSCLC patients, which will facilitate optimized and individualized treatment strategies.
Trebeschi S, et al., 2022 [221]	Benchmarking study	152 stage-IV NSCLC patients	Research	AI-derived survival score was independent of clinical, radiological, PD-L1, and histopathological factors, and visual analysis of AI-generated prognostic heatmaps revealed the relative prognostic importance of morphological nodal changes in the mediastinum, supraclavicular, and hilar regions; lung and bone metastases; as well as pleural effusions, atelectasis, and consolidations.
Park C, et al., 2022 [222]	Benchmarking study	512 patients with NSCLC	Research	In this AI model, CT radiomics and predicted tumor-infiltrated lymphocytes were significantly associated with immune checkpoint inhibitors’ outcomes in NSCLC patients.
Ventura D, et al., 2023 [261]	Benchmarking study	44 patients with advanced NSCLC	In vitro diagnostics	This radiomics-based model might be able to predict response in advanced NSCLC patients treated with CKI-based first-line therapy with or without chemotherapy, including disease progression or death.
Su Z, et al., 2023 [279]	Laboratory study	2 human lncRNA-disease associations datasets	In vitro diagnostics	CCDC26 and IFNG-AS1 may be new biomarkers of lung cancer, SNHG3 may associate with PD-L1 for lung cancer.
Wang X, et al., 2021 [291]	Benchmarking study	110 patients with NSCLC	Research	An AI method to critically evaluate PD-L1 expression can empower the application of deep learning in digital pathology and facilitate computer-aided diagnosis.
Prelaj A, et al., 2023 [295]	Multi-center, retrospective and prospective, observational study protocol	15 references	Research	The final objective of the project is the construction of a novel, integrated, AI-assisted data storage and elaboration platform to guide immunotherapy administration in advanced NSCLC, ensuring easy access and cost-effective use by healthcare providers and patients.
Park S, et al., 2022 [300]	Benchmarking study	3166 whole-slide images	Research	The inflamed immune phenotype is associated with better response to immunotherapy.
Baxi V, et al., 2022 [301]	Benchmarking study	1746 whole-slide images	Research	High PD-L1 expression via AI evaluation is associated with better response to immunotherapy.
Kang Y, et al., 2022 [303]	Laboratory study	48 tissue samples from 32 cancer patients	In vitro diagnostics	A computational framework that uses advanced deep neural network modeling and feature elimination to reduce data on ∼26,000 genes to 6 genes that accurately predict response to immunotherapy was developed.
Prelaj A, et al., 2022 [304]	Prospective observational study	164 patients with NSCLC	Research	The integration of multi-factorial data provided by AI techniques is a useful tool to select NSCLC patients as candidates for immunotherapy.
Shoji F, et al., 2022 [306]	Prospective study protocol	400 patients with lung cancer	Research	A rather unexpected biomarker that could predict the response to immunotherapy is the intestinal microbiome.
Sha L, et al., 2019 [355]	Benchmarking study	130 NSCLC patients	Research	AI can determine the PD-L1 status of the tumor with an AUC of 0.8.
Choi S, et al., 2022 [356]	Benchmarking study	802 NSCLC whole-slide images	Research	The AI-powered tumor proportion score analysis-based assistance improved the pathologists’ consensus of reading and prediction of the therapeutic response.
Lin YY, et al., 2022 [367]	Laboratory study	33 needle-biopsy-sized specimens	Research	PD-L1 expression was accurately determined by AI through 3D pathology imaging.
Meißner AK, et al., 2023 [374]	Benchmarking study	53 NSCLC patients with brain metastasis	Research	In patients with brain metastasis, the AUC for PD-L1 expression was calculated as between 0.77 and 0.84, depending on the parameters included in its prediction.
Park J, et al., 2022 [450]	Laboratory study	965 samples of non-small-cell lung carcinoma	In vitro diagnostics	Among the three immune phenotypes of lung cancer—inflamed, immune-excluded, and immune-desert—it was found that, in the inflamed sub-type (which shows higher cytolytic score), the enriched pathways are generally associated with the immune response and immune-related cell types (which are highly expressed) whereas, in the immune excluded sub-type, enriched glycolysis, fatty acid, and cholesterol metabolism pathways are observed.
Ciccolini J, et al., 2020 [456]	Cohort study	450 advanced lung cancer patients	Research	This project aims to generate dense longitudinal data in lung cancer patients undergoing anti-PD1/PD-L1 therapy, in order to produce mathematical models with mechanistic learning algorithms that will help to decipher the mechanisms underlying the response or resistance to immunotherapy.
Prelaj A, et al., 2023 [458]	Retrospective observational study	480 lung cancer patients receiving immunotherapy (73 combined with chemotherapy)	Research	The most significant factors for the outcome of immunotherapy are neutrophil to lymphocyte ratio (NLR), eastern cooperative oncology group-performance status (ECOG-PS), PD-L1 expression, line of treatment of immunotherapy, and the combination of chemo-immunotherapy, with an accuracy ranging between 0.73 and 0.83; the main focus concerning lung cancer treatment and prognosis is directed to immunotherapy.
Li S, et al., 2022 [459]	Retrospective observational study	289 lung cancer patients receiving immunotherapy	Research	Several AI models predicting various outcomes, such as disease control rate (DCR), objective response rate (ORR), PFS, and OS, achieved AUCs varying between 0.704 and 0.9526; neoadjuvant and adjuvant immunotherapy may bring greater efficacy benefits to patients.
Guo H, et al., 2021 [460]	Cohort study	1859 tumor microarrays from 121 NSCLC patients and 214 whole-slide immunohistochemistry from an external cohort of 30 NSCLC patients	In vitro diagnostics	The features of galectin-9, OX40, OX40L, KIR2D, and KIR3D had significant contributions to overall survival (OS) and relapse-free survival (RFS) in the internal cohort; the integrated score presented the highest AUC for OS and RFS, which achieved 0.9 and 0.85 in the internal testing cohort; in the external cohort, the AUC varied between 0.80 and 0.87 for OS, and between 0.83 and 0.94 for RFS; the neutrophil-to-lymphocyte ratio (NLR) combined with the PD-1/PD-L1 signature can serve as a predictor for RFS in the external cohort.
Al-Droubi SS, et al., 2023 [461]	Retrospective observational study	2450 patients treated with immunotherapy	Research	AI models are able to predict the emergence of cardiovascular disease in cancer patients who receive immunotherapy with an accuracy and AUC of over 90%.

**Table 17 cancers-17-00882-t017:** Studies in the sector of prognosis.

Reference	Type of Study	Participants	AI Status	Main Conclusions
Zhou H, et al., 2021 [19]	Prospective, single-arm, multi-center, phase II trial	40 patients with lung cancer	Research	This study indicates whether a new multi-omics AI system constitutes a valid strategy for neoadjuvant therapy effect prediction and decision-making assistance in the context of neoadjuvant treatment for patients with stage IB–IIIA NSCLC.
Revel MP, et al., 2022 [22]	Prospective cohort study protocol	2400 asymptomatic women	Research	Most lung cancer screening studies have predominantly included male participants with women being under-represented, even though the benefit of screening is greater for them.
Mapanga W, et al., 2021 [23]	Consensus study	27 health stakeholders	Research	Finding solutions to overcome both individual and health system-level obstacles to lung cancer screening and management are vital to facilitate early identification and treatment, in addition to improving survival.
Chao HS, et al., 2023 [36]	Retrospective observational study	200 patients with lung cancer	In vitro diagnostics	AI algorithm-embedded CT screening significantly ameliorates tedious LDCT practices for doctors.
Zhang Y, et al., 2023 [50]	Observational study	4395 participants	Research	LDCT screening could detect a high proportion of early-stage lung cancer patients in a Chinese high-risk population.
Balbi M, et al., 2023 [52]	Retrospective observational study	4098 volunteers	In vitro diagnostics	Automated coronary artery calcium and quantitative emphysema added prognostic information to age, sex, and pack-years for predicting mortality, but not lung cancer incidence, in a lung cancer screening setting.
Xu K, et al., 2023 [84]	Secondary analysis of a clinical trial	20,768 participants	In vitro diagnostics	The body composition measurements automatically derived from baseline low-dose CT examinations added predictive value for lung cancer death, CVD death, and all-cause death, but not for lung cancer incidence.
Sabia F, et al., 2023 [85]	Clinical trial	2239 volunteers	In vitro diagnostics	Fully automated coronary artery calcium scoring was effective in predicting all-cause mortality at 12 years in a lung cancer screening setting.
Ruggirello M, et al., 2023 [86]	Retrospective observational study	6495 heavy smokers	In vitro diagnostics	The lower coronary artery calcification burden observed in women at all ages might contribute to explaining their lower rates of all-cause mortality and better lung cancer survival.
Chamberlin J, et al., 2021 [87]	Retrospective observational study	117 patients who underwent LDCT	In vitro diagnostics	AI rapidly and accurately identifies significant risk factors for cardiopulmonary disease on standard screening low-dose chest CT.
Tan Q, et al., 2021 [99]	Benchmarking study	167 patients with NSCLC	Research	The density of CD45RO⁺ TIL and the combination of CD45RO⁺ TIL and PD-L1 in tumor areas significantly correlated with clinicopathological features and prognosis of NSCLC, and can be used as a novel marker.
Tong H, et al., 2022 [100]	Retrospective multi-cohort study	1405 NSCLC patients	In vitro diagnostics	^18^F-FDG PET/CT radiomics–clinical combined model could be a clinically practical method to non-invasively detect the tumor immune status in NSCLCs.
Li X, et al., 2019 [107]	Benchmarking study	137 patients with NSCLC	Research	Several potential prognostic biomarkers of PET/CT imaging were extracted for predicting survival and selecting patients with NSCLC who are more likely to benefit from chemotherapy.
Tang FH, et al., 2023 [108]	Benchmarking study	422 NSCLC patients	Research	The proposed AI model has the potential to improve NSCLC prognosis and facilitate personalized treatment decisions.
Zhang D, et al., 2023 [109]	Benchmarking study	21,093 patients’ clinical data	In vitro diagnostics	The interpretable AI survival predictive tool for SCLC, using clinical data, had a reliable predictive value on their overall survival.
Brocki L, et al., 2022 [177]	Benchmarking study	CT images of lung tumors from the LIDC-IDRI dataset	In vitro diagnostics	The proposed model combines biomarkers and radiomics features in an interpretable AI model which demonstrates excellent performance for lung nodule malignancy classification.
Su Z, et al., 2022 [199]	Benchmarking study	223 patients with early-stage adenocarcinoma	Research	AI-assisted diagnosis system could predict the invasive sub-types of early-stage lung adenocarcinoma appearing as pulmonary nodules, and has certain predictive value.
Lee KH, et al., 2022 [205]	Benchmarking study	1426 patients with stage I–IV lung cancer	Research	The CT-based composite score obtained from AI based on histopathological features predicted survival in early-stage lung adenocarcinomas with high reproducibility.
Wang S, et al., 2022 [207]	Benchmarking study	18,232 patients with lung cancer	In vitro diagnostics	This AI system provides a non-invasive method to detect EGFR genotype and identify patients with an EGFR mutation at high risk of TKI resistance.
Dora D, et al., 2022 [212]	Cohort study	129 advanced stage NSCLC patients	Research	CT-based texture analysis and gut microbiome signatures can predict overall survival, the response to therapy, the PD-L1 expression, and toxicity in NSCLC patients treated with ICI.
Tonneau M, et al., 2022 [213]	Benchmarking study	642 advanced NSCLC patients	In vitro diagnostics	A risk prediction model combining clinical and AI-Radiomics was generalizable following CT scan harmonization and AI generalization methods, with similar performances to routine oncology practice in predicting ICI response.
Trebeschi S, et al., 2022 [214]	Benchmarking study	1055 primary and meta- lesions from 203 + 262 NSCLC patients	Research	Radiographic characteristics of lesions on standard-of-care imaging may function as non-invasive biomarkers for response to immunotherapy, and may show utility for improved patient stratification in both neoadjuvant and palliative settings.
Jin W, et al., 2022 [215]	Benchmarking study	143 primary lung cancer patients	Research	The AI model predicts the response of lesions to PD-1 blockade, which implies the importance of introducing the non-linear dynamic effects and AI approaches to the original delta-radiomics in the future.
Deng K, et al., 2022 [216]	Retrospective study	570 patients with stage IV EGFR-mutant NSCLC	In vitro diagnostics	This study developed and externally validated a pre-operative CT image-based AI model to predict the survival benefits of EGFR-TKI and ICI therapies in stage IV NSCLC patients, facilitating optimized and individualized treatment strategies.
Kim H, et al., 2022 [219]	Retrospective study	135 patients with lung cancer	Research	This study exhibited the external validity and transportability of the CT-based AI prediction model for radiotherapy candidates through predicting local recurrence-free survival, disease-free survival, and overall survival.
Trebeschi S, et al., 2022 [221]	Benchmarking study	152 stage-IV NSCLC patients	Research	AI-derived survival score was independent of clinical, radiological, PD-L1, and histopathological factors, and visual analysis of AI-generated prognostic heatmaps revealed relative prognostic importance of morphological nodal changes in the mediastinum, supraclavicular, and hilar regions; lung and bone metastases; as well as pleural effusions, atelectasis, and consolidations.
Park C, et al., 2022 [222]	Benchmarking study	512 patients with NSCLC	Research	In this AI model, CT radiomics and predicted tumor-infiltrated lymphocytes were significantly associated with immune checkpoint inhibitors’ outcomes in NSCLC patients.
van Amsterdam WAC, et al., 2022 [223]	Laboratory study	1018 scans from 1010 patients	Research	Through combining AI and structural causal models, an unbiased individual prognosis prediction can be achieved.
Huang B, et al., 2022 [224]	Multi-center retrospective observational study	1168 lung nodules	Research	AI trained using pre-treatment FDG-PET/CT performed well in predicting lung malignancy progression and overall survival, comparable to a radiomics approach, and could inform treatment options and improve patient care.
Lee JH, et al., 2022 [225]	Retrospective cohort study and systematic review with meta-analysis	612 patients with adenocarcinoma and 13 studies with 12,080 patients	In vitro diagnostics	Early-stage lung adenocarcinomas manifesting as solid nodules at pre-operative chest CT, which indicates the absence of ground-glass opacity, were associated with poor post-operative survival.
Kawaguchi Y, et al., 2022 [226]	Benchmarking study	772 patients with stage 0-I adenocarcinoma	Research	The solid-part volume measured by AI was superior to conventional methods in predicting the prognosis of clinical stage 0–I adenocarcinoma.
Shimada Y, et al., 2022 [227]	Benchmarking study	642 patients with NSCLC	In vitro diagnostics	CT-based radiomics coupled with AI contributes to the non-invasive prediction of early recurrence in patients with c-stage 0–IA NSCLC.
Chen X, et al., 2022 [228]	Retrospective observational study	33,410 nodules among 2126 participants	Research	The AI-detected total nodule number is significantly associated with survival rates in patients with surgically resected stage III NSCLC.
Lee J, et al., 2022 [229]	Retrospective observational study	70 consecutive patients with SQCLC	Research	Patients with advanced SQCLC under palliative chemotherapy lose a significant amount of skeletal muscle, with the reduction in muscle area becoming faster at the end of life.
Borrelli P, et al., 2022 [254]	Benchmarking study	115 patients who had undergone FDG PET/CT	In vitro diagnostics	Both manual and AI total lesion glycolysis (TLG) were significantly associated with overall survival; however, its low sensitivity makes this AI model less useful in supporting clinical reading, reporting, and staging.
Borrelli P, et al., 2022 [256]	Benchmarking study	320 patients referred for FDG PET/CT due to suspected lung cancer	Research	AI was trained to segment lung tumors and thoracic lymph nodes, and fully automated AI-based total lesion glycolysis (TLG) measurements were significantly associated with overall survival in patients with lung cancer.
Ventura D, et al., 2023 [261]	Benchmarking study	44 patients with advanced NSCLC	In vitro diagnostics	This radiomics-based model might be able to predict response in advanced NSCLC patients treated with CKI-based first-line therapy with or without chemotherapy, including disease progression or death.
Benedum CM, et al., 2023 [270]	Observational study	186,313 patients with lung cancer	In vitro diagnostics	A biomarker- and treatment-defined cohort, in order to perform analyses related to biomarker-associated survival and comparative effectiveness of treatments were utilized.
Park MK, et al., 2022 [277]	Benchmarking study	1122 and 763 cancer and non-cancer patients	Research	Using multi-omic data integration, a promising cancer prediction algorithm was developed.
Bedard ELR, et al., 2021 [278]	Randomized controlled trial	Consecutively NSCLC patients and matched controls	Research	A highly specific, sensitive, and non-invasive detection method for NSCLC which can potentially improve the early detection and, hence, the treatment and survival outcomes of patients was developed.
Chen BT, et al., 2021 [290]	Benchmarking study	110 patients with NSCLC	Research	Predictive modeling of survival duration using both clinical and radiomic features yielded areas under the receiver operative characteristic curve of 0.977, 0.905, and 0.947 for the EGFR, ALK, and KRAS mutation-positive groups, respectively.
Zuo Z, et al., 2023 [292]	Cohort study	175 patients with clinical T1 stage NSCLC	In vitro diagnostics	A high-risk AI score is a diagnostic biomarker for lymphovascular invasion in patients with clinical T1 stage NSCLC, and can serve as a prognostic biomarker for them.
Prelaj A, et al., 2023 [295]	Multi-center, retrospective and prospective, observational study protocol	15 references	Research	Using AI-based tools, the aim of this study is to promote individualized treatment in advanced NSCLC, with the goals of improving survival and quality of life, minimizing or preventing undue toxicity, and promoting efficient resource allocation.
Park S, et al., 2022 [300]	Benchmarking study	3166 whole-slide images	Research	The inflamed immune phenotype is associated with better response to immunotherapy.
Baxi V, et al., 2022 [301]	Benchmarking study	1746 whole-slide images	Research	High PD-L1 expression via AI evaluation is associated with better response to immunotherapy.
Prelaj A, et al., 2022 [304]	Prospective observational study	164 patients with NSCLC	Research	The integration of multi-factorial data provided by AI techniques is a useful tool to select NSCLC patients as candidates for immunotherapy.
Mahmoud AM, et al., 2023 [309]	Benchmarking study	33 lung cancer tissue sections	Research	PRMT6 is associated with poor lung cancer prognosis.
Miller HA, et al., 2022 [310]	Benchmarking study	Tumor core biopsies from 44 patients with lung cancer	Research	Various metabolic biomarkers, such as guanine, choline, creatine, leucine, tryptophan, N6,N6,N6-Trimethyl-L-lysine, L-pyrogluatmic acid, benzoic acid, cystine, methionine sulfoxide, histamine, itaconic acid, pyruvate, and malonic acid have been associated with different outcomes.
Li J, et al., 2022 [360]	Benchmarking study	139 individuals with pulmonary nodules	Research	AI-based approach may be used for “Super Early” cancer diagnosis and amend the current immunotherapy for lung cancer.
Sorin M, et al., 2023 [363]	Laboratory study	Samples from 416 patients with lung cancer	In vitro diagnostics	More than 1.6 million cells, enabling spatial analysis of immune lineages and activation states with distinct clinical correlates (including survival), were resolved.
Keller A, et al., 2020 [368]	Laboratory study	210 serum samples from cancer patients	Research	MicroRNAs, which can be obtained through liquid biopsy and analyzed through AI, might be indicative of lung tumors decades before their clinical appearance.
Tanaka Y, et al., 2020 [375]	Laboratory study	3 NSCLC cell lines with 19,849 genes	Research	AI models predicted the process of metastasis using a Bayesian network applied to the Epithelial–Mesenchymal Transition data set.
Huang Z, et al., 2020 [377]	Benchmarking study	5973 cases of NSCLC	In vitro diagnostics	The kind of surgical treatment, radiotherapy, and chemotherapy have been related, through AI, with the subsequent appearance of bone metastasis and survival.
Yang Y, et al., 2022 [378]	Benchmarking study	998 patients with lung cancer	Research	One of the most important predictors of the AI prognostic model among clinical status was TNM stage.
Kinoshita F, et al., 2023 [379]	Benchmarking study	1049 patients with resected NSCLC	In vitro diagnostics	AI prognostic model showed that the AUC values for disease-free survival, overall survival, and cancer-specific survival at 5 years were 0.890, 0.926, and 0.960, respectively.
Field M, et al., 2021 [400]	Retrospective analysis of the data of a clinical trial	482 patients with lung cancer who received radiotherapy	Research	External validation of a 2-year overall survival model for NSCLC gave an AUC of 0.65 and C-index of 0.62 across the network; AI may assist in recruiting suitable patients for clinical trials of novel radiotherapy methods.
Janopaul-Naylor JR, et al., 2024 [402]	Laboratory study	117 questions about lung cancer presented to 2 AI chatbots	Research	AI chatbots provide multiple opportunities for innovating health care, particularly relating to cancer prediction, in order to limit misleading counseling, confusion, and emotional distress in patients and families.
Kudo Y, et al., 2022 [428]	Cohort study	157 patients who had clinical N0 NSCLC	In vitro diagnostics	SUV_max_ values in PET scans and abnormal CEA levels are unfavorable prognostic factors, and might suggest that lobectomy should be preferred to segmentectomy.
Wang X, et al., 2023 [429]	Protocol for a retrospective observational study	All NSCLC patients during an 8-year period from 30 large Chinese centers	Research	Study protocol for new research focused on the effectiveness of surgical treatment and neo-adjuvant chemotherapy, with the help of AI, in clinical stages I–IIIA.
Rozynek M, et al., 2024 [444]	Cohort study	178 patients with NSCLC	In vitro diagnostics	Automatically extracted muscle and adipose tissue radiomic features could be used as predictors of survival in patients with non-small cell lung cancer.
Prelaj A, et al., 2023 [458]	Retrospective observational study	480 lung cancer patients receiving immunotherapy (73 combined with chemotherapy)	Research	The most significant factors for the outcome of immunotherapy are neutrophil to lymphocyte ratio (NLR), eastern cooperative oncology group-performance status (ECOG-PS), PD-L1 expression, line of treatment of immunotherapy, and the combination of chemo-immunotherapy, with an accuracy ranging between 0.73 and 0.83; the main focus concerning lung cancer treatment and prognosis is directed to immunotherapy.
Li S, et al., 2022 [459]	Retrospective observational study	289 lung cancer patients receiving immunotherapy	Research	Several AI models predicting various outcomes, such as disease control rate (DCR), objective response rate (ORR), PFS, and OS, achieved AUCs varying between 0.704 and 0.9526; neoadjuvant and adjuvant immunotherapy may bring greater efficacy benefits to patients.
Guo H, et al., 2021 [460]	Cohort study	1859 tumor microarrays from 121 NSCLC patients and 214 whole-slide immunohistochemistry from an external cohort of 30 NSCLC patients	In vitro diagnostics	The features of galectin-9, OX40, OX40L, KIR2D, and KIR3D had significant contributions to overall survival (OS) and relapse-free survival (RFS) in the internal cohort; the integrated score presented the highest AUC for OS and RFS, which achieved 0.9 and 0.85 in the internal testing cohort; in the external cohort, the AUC varied between 0.80 and 0.87 for OS, and between 0.83 and 0.94 for RFS; the neutrophil-to-lymphocyte ratio (NLR) combined with the PD-1/PD-L1 signature can serve as a predictor for RFS in the external cohort.
Torrente M, et al., 2023 [469]	Retrospective observational study	5275 patients with different types of cancer	In vitro diagnostics	A prognostic model identified low- and high-risk profile patients. The reconstruction of the population’s risk profile for the cancer-specific predictive model proved useful in clinical practice using artificial intelligence.
Hsu JC, et al., 2023 [470]	Retrospective observational study	3714 lung cancer patients from a dataset	Research	Nine machine-learning algorithms with various modes were used. The best performance of the artificial neural network (ANN) model was achieved when integrating all variables, with AUC, accuracy, precision, recall, and F1-score of 0.89, 0.82, 0.91, 0.75, and 0.65, respectively.
Loureiro H, et al., 2023 [471]	Benchmarking study	136,719 + 1187 patients from two datasets (training and validation)	In vitro diagnostics	An increase in the number of prognostic covariates did not lead to an increase in model performance.
Zhu F, et al., 2023 [472]	Benchmarking study	601,480 + 4512 patients from two datasets (training and validation)	In vitro diagnostics	The model outperformed the traditional Cox model, was robust with missing data, and provided the AI certainty of prediction. It can be used for patient self-evaluation and risk stratification in clinical trials. Researchers can fine-tune the pre-trained model and integrate their own database to explore other prognostic factors.
Kothari G, et al., 2023 [473]	Retrospective observational study	92 + 20 patients from two datasets	In vitro diagnostics	Decreased robustness of a radiomic feature is significant, as it may impact the features’ prognostic capability.
DuCote TJ, et al., 2023 [474]	Laboratory study	19 human and 33 murine lung tumor samples	In vitro diagnostics	In human samples, a strong negative correlation between neutrophils and lymphocytes, as well as between mesenchymal cells and lymphocytes and higher percentages of mesenchymal cells, correlate with poor prognosis.

**Table 18 cancers-17-00882-t018:** Summary of the main applications of AI for lung cancer diagnosis and management and their reported performance.

AI Application	Accuracy	Sensitivity	Specificity	Positive Predictive Value (PPV)	Negative Predictive Value (NPV)	Area Under the Curve (AUC)
Screening						
Lung cancer detection via epidemiological features						0.916
Lung cancer detection via LDCT screening		94.6%	93.6%			
Early lung cancer detection via microRNA (miRNA)		96.4%	88.2%			0.910
Early lung cancer detection via long non-coding RNA (lncRNA)						0.870–0.971
Lung cancer detection using routine blood indices	95.7%	96.3%	95.0%			
Outcomes due to major cardiopulmonary events via LDCT						0.911–0.942
Data extraction from electronic health records	66.0–100%	94.6%	93.6%			
Screening cancer patients for clinical trials	91.6%					
Clinical features						
Lung cancer detection based on LDCT, age, and smoking history		89.5%	81.3%			0.895
Differentiate benign from malignant tumors using clinical features	96.2%					
Outcomes in lung cancer patients using clinical data	82.0%					0.833–0.890
X-rays						
Lung cancer detection via chest X-ray		60.0–90.0%	60.0–97.0%	80.0–95.0%	89.0–91.0%	0.740–0.884
Chest CT						
Automated nodule detection on chest CT		59.2–100%	70.8–99.1%	55.6%	88.2%	0.880–0.942
Malignancy discrimination of lung nodules in chest CT		40.8–99.0%	39.3–97.3%	78.9–97.6%	58.2–97.3%	0.640–0.970
Automated nodule detection and malignancy definition in chest CT	87.0–99.4%	87.0%	87.0%			0.930
Discrimination between SCLC, AC, and SqCLC in chest CT	57.7%	90.0%	44.0%			0.870–0.970
Differentiation between AC sub-types on chest CT						0.779–0.946
Determine N stage via chest CT image analysis	86.4–98.6%	81.0–98.4%	75.3–100%			0.761–0.864
Identify bone metastases using CT images	87.9%	89.4%				
EGFR status prediction using chest CT						0.748–0.940
KRAS status prediction using chest CT						0.630–0.930
ALK status prediction using chest CT						0.740–0.870
Classify EGFR mutation sub-type via chest CT						0.805–0.868
Identify EGFR, ALK, ERBB2, BRAF, MET, ROS1, RET, and KRAS mutations						0.862
Predict CD8 expression using chest CT						0.861
Predict PD-L1 expression using chest CT						0.856–0.930
Predict response to radiotherapy and survival outcomes via chest CT						0.660–0.720
Predict radiation pneumonitis after radiotherapy via chest CT						0.550–0.830
Predict response to chemotherapy regimens via chest CT	80.1–97.4%					
Predict response to tyrosine kinase inhibitors via chest CT				77.4–80.4%		0.748–0.813
Predict response to immune checkpoint inhibitors via chest CT	84.0%			90.0%		0.520–0.910
Predict response to immunotherapy and survival via chest CT						0.600–0.950
Predict outcomes via radiomic features on chest CT						0.670–0.949
PET scan						
AI-based automated nodule detection on PET	97.0%	90.0–95.9%	94.2–98.1%	88.0%	100%	0.970–0.989
Discrimination between benign and malignant lung nodules in PET		55.6–83.0%	70.0–93.0%			0.743–0.900
Histological prediction and gene expression of lung cancer in PET						0.870
Identification of T lesions in PET/CT staging		86.2%				
Identification of N lesions in PET/CT staging	80.0%	74.0–88.0%	84.0%			0.880–0.940
Identification of M lesions in PET/CT staging	63.0%	45.0%	79.0%			
Complete TNM stage categorization via PET images	55.6–99.1%					
Prediction of ALK status via PET/CT images						0.870–0.880
Prediction of KRAS status via PET/CT images						0.930
Prediction of EGFR status via PET/CT images						0.940
Prediction of MIB-1 (Ki 67) status via PET	78.0–86.0%					
Identify ALK/ROS1/RET fusion-positive tumors via PET/CT		73.0%	70.0%			
Predict CD8 expression using PET/CT						0.932
Predict response to immune checkpoint inhibitors via PET						0.690–0.750
Predict outcome via gene expression and PET/CT						0.870
Predict outcomes via PET/CT metabolic indices						0.459–0.634
Biomarkers						
Early lung cancer detection via microRNA (miRNA)		96.4%	88.2%			0.910
Early lung cancer detection via long non-coding RNA (lncRNA)						0.870–0.971
Lung cancer detection via chromatin structural changes in buccal mucosa						0.820–0.920
Lung cancer detection via DNA methylation analysis of ctDNA						1.000
Lung cancer detection via mRNA expression	93.7%					
Lung cancer detection using routine blood indices	95.7%	96.3%	95.0%			
Lung cancer detection via ProGRP, VRGF, CEA, CYFRA21-1, and NSE		94.1%				0.849–0.910
Lung cancer detection using urinary 4MPLA		82.1%	88.2%			0.850
Differentiation of pleural effusion malignancy via CEA						0.692–0.763
Histologic type via TP63, EPHA10, FBN2, and IL1RAP gene expression						0.870
AC differentiation via SERPINA3, Ig Kappa light chain, and SAA1						0.916–0.935
Immunohistochemical expression of PD-L1	96.4%		96.8%			0.670–0.810
High PD-L1 expression using gut microbiome	79.0%					
Tyrosine kinase inhibitor sensitivity using phosphoproteins						0.780–0.790
Favorable immunotherapy response using gut microbiome	84.0%					
Non-response to immunotherapy using CYFRA			95.0%			
Predicting survival based on MRI image of brain metastasis and EGFR						0.977
Predicting survival based on MRI image of brain metastasis and ALK						0.905
Predicting survival based on MRI image of brain metastasis and KRAS						0.947
Biopsy						
Lung cancer diagnosis in histopathological specimens		93.0%	68.0%			0.900
Differentiate lung cancer sub-types in histopathological specimens	96.3–99.7%	97.0–99.9%	87.0–100%			0.940–0.999
Differentiate between different types of lung adenocarcinoma						0.800
Lung cancer diagnosis in cytological specimens	94.5%					0.991
Differentiate between lung cancer sub-types in cytological specimens	66.8–76.8%					
Pathology of hesperetin-derived SCLC via fiber optic bronchoscopy	97.9%					
Differentiate between benign and malignant tumors in EBUS images	82.0%					0.880
Differentiate lung tumor malignancy via white blood cells						0.822–0.883
Prediction of EGFR status in histopathological images						0.740–0.897
Prediction of ALK status in histopathological images						0.921–0.995
Prediction of EGFR status in cytological images	95.0%					0.950
Prediction of KRAS status in cytological images	75.0%					0.950
Prediction of ALK status in cytological images	80.0%					0.750
Guide TKI selection based on histopathological images	83.0–93.0%					
Immunohistochemical expression of PD-L1	96.4%		96.8%			0.670–0.810
Staging						
Identification of T lesions on PET/CT staging		86.2%				
Determine lymph node metastasis using any examination via AI						0.740
Determine N stage via chest CT image analysis	86.4–98.6%	81.0–98.4%	75.3–100%			0.859–0.864
Identification of N lesions in PET/CT staging	80.0%	74.0–88.0%	84.0%			0.880–0.940
Determine N stage via EBUS image analysis	72.9–82.0%	89.0%	72.0–90.8%	83.0%	75.9–81.0%	0.782
Mediastinal lymph node invasion in 0–IA stage tumors		69.0%	65.0%		94.0%	0.761
Discriminate between benign and malignant pleural effusion		87.5%	92.9%	93.3%	86.7%	0.886–0.918
Predicting brain metastasis via electronic health records						0.825
Identify distant metastases in whole-body CT	50.9%					
Identify bone metastases using CT images	87.9%	89.4%				
Identification of M lesions in PET/CT staging	63.0%	45.0%	79.0%			
Progression of brain metastases after radiotherapy via MRI		77.0%	74.0%			
Identify bone metastases via bone scintigraphy	76.7–81.2%	81.3%	81.1%			0.838–0.957
Complete TNM stage categorization via PET images	55.6–99.1%					
PD-L1 status prediction based on MRI images of brain metastases						0.770–0.840
Predicting survival based on MRI image of brain metastasis and EGFR						0.977
Predicting survival based on MRI image of brain metastasis and ALK						0.905
Predicting survival based on MRI image of brain metastasis and KRAS						0.947
Predicting disease-free survival based on lung cancer stage						0.890
Predicting cancer specific survival based on lung cancer stage						0.960
Predicting overall survival based on lung cancer stage						0.926
Treatment general						
Concordance between AI and multidisciplinary teams’ decisions	81.5–85.0%					
Effectiveness of AI in personalized treatment strategies	99.0%	100%	96.0%			0.993
AI predicting ability of disease recurrence	87.5–89.9%					
Surgical treatment						
Recognize pulmonary vessels in thoracic surgery	82.8%					
Recognize segmental arteries in thoracic surgery	79.7%					
Recognize lobular veins in thoracic surgery	96.3%					
Robotic radiosurgery via AI-system guidance	100%					
Distinguish between different types of AC and normal tissue	84.9%	89.0–94.0%	80.6–82.7%			0.862
Major pathologic remission to pre-operative PD-1 blockade						0.910
Predict outcomes after a curable surgery						0.800–0.960
Recall information for therapeutic purposes	100%					
Radiotherapy						
Recruit patients for radiotherapy clinical trials						0.650
Dice similarity coefficient via auto-segmentation of lungs	75.0%					
Dice similarity coefficient via auto-segmentation of lungs, heart, and liver	90.0%					
Dice similarity coefficient via auto-segmentation of esophagus	71.0–87.0%					
Avoidance of pulmonary, cardiac, and esophageal toxicity						0.706–0.739
Predict radiation pneumonitis after radiotherapy via chest CT						0.550–0.830
Prediction response to radiotherapy and local progression						0.660–0.800
Prediction response to radiotherapy and progression of brain metastasis		77.0%	74.0%			
Predict response to radiotherapy and survival outcomes via chest CT						0.660–0.720
Predict outcomes in lung cancer after radiotherapy						0.750–0.800
Chemotherapy						
Tolerance of chemotherapy plus methotrexate in elderly patients	97.4%					
Vomiting management in chemotherapy for early lung cancer		95.0%	89.0%			0.870
Predict response to chemotherapy regimens via chest CT	80.1–97.4%					
Response to immune checkpoint inhibitors plus chemotherapy						0.690–0.750
Survival in chemotherapy patients based on body composition	98.0%					0.730–0.740
Targeted therapy						
EGFR status prediction using chest CT						0.748–0.940
KRAS status prediction using chest CT						0.630–0.930
ALK status prediction using chest CT						0.740–0.870
Classify EGFR mutation sub-type via chest CT						0.805–0.868
Identify EGFR, ALK, ERBB2, BRAF, MET, ROS1, RET, and KRAS mutations						0.862
Prediction of ALK status via PET/CT images						0.870–0.880
Prediction of KRAS status via PET/CT images						0.930
Prediction of EGFR status via PET/CT images						0.940
Identify ALK/ROS1/RET fusion-positive tumors via PET/CT		73.0%	70.0%			
Prediction of MIB-1 (Ki 67) status via PET	78.0–86.0%					
Prediction of EGFR status in histopathological images						0.740–0.897
Prediction of ALK status in histopathological images						0.921–0.995
Prediction of EGFR status in cytological images	95.0%					0.950
Prediction of KRAS status in cytological images	75.0%					0.950
Prediction of ALK status in cytological images	80.0%					0.750
Distinguish between EGFR mutations via genomic analysis	83.0–93.0%					
Guide TKI selection based on histopathological images	83.0–93.0%					
Prediction of response to KRAS^G12C^ inhibitors	85.0%					
Tyrosine kinase inhibitors sensitivity using phosphoproteins						0.780–0.790
Prediction of response to EGFR-TKI treatment and survival benefit				75.4–80.4%		0.748–0.813
Predicting survival based on MRI image of brain metastasis and EGFR						0.977
Predicting survival based on MRI image of brain metastasis and ALK						0.905
Predicting survival based on MRI image of brain metastasis and KRAS						0.947
Immunotherapy						
Predict CD8 expression using chest CT						0.861
Predict CD8 expression using PET/CT						0.932
Prediction of PD-L1 expression via clinical and radiomic features	83.0–88.0%					0.630–0.905
Predict PD-L1 expression by using chest CT						0.856–0.930
Immunohistochemical expression of PD-L1	96.4%		96.8%			0.670–0.810
High PD-L1 expression using gut microbiome	79.0%					
PD-L1 status prediction based on MRI images of brain metastases						0.770–0.840
Quantification of different levels of PD-L1 expression						0.934–0.950
Prediction of PD-L1 sub-types via clinical and radiomic features						0.837–0.868
Major pathologic remission to pre-operative PD-1 blockade						0.910
Predict response to immune checkpoint inhibitors	75.6–84.0%			90.0%		0.520–0.910
Predict response to immune checkpoint inhibitors via PET						0.690–0.750
Favorable immunotherapy response using gut microbiome	84.0%			90.0%		
Non-response to immunotherapy using CYFRA			95.0%			
Predict cardiovascular events in patients receiving immunotherapy	94.4–99.2%					0.968–0.996
Predict outcomes in patients receiving immunotherapy	73.0–83.0%					0.670–0.953
Prognosis						
Predict outcomes due to major cardiopulmonary events via LDCT						0.911–0.942
Predict outcomes in lung cancer patients using clinical data	82.0%					0.833–0.890
Predict outcomes via radiomic features in chest CT						0.670–0.949
Predict response to immunotherapy and survival via chest CT						0.600–0.950
Predict response to radiotherapy and survival outcomes via chest CT						0.660–0.720
Predict response to chemotherapy regimens via chest CT	80.1–97.4%					
Predict response to immune checkpoint inhibitors via PET						0.690–0.750
Predict outcomes via PET/CT metabolic indices						0.459–0.634
Predict outcome via gene expression and PET/CT						0.870
Tyrosine kinase inhibitors sensitivity using phosphoproteins						0.780–0.790
Non-response to immunotherapy using CYFRA			95.0%			
Favorable immunotherapy response using gut microbiome	84.0%			90.0%		
Predicting survival based on MRI image of brain metastasis and EGFR						0.977
Predicting survival based on MRI image of brain metastasis and ALK						0.905
Predicting survival based on MRI image of brain metastasis and KRAS						0.947
Predicting disease-free survival based on lung cancer stage						0.890
Predicting cancer specific survival based on lung cancer stage						0.960
Predicting overall survival based on lung cancer stage						0.926
Major pathologic remission to pre-operative PD-1 blockade						0.910
Predict outcomes after a curable surgery						0.800–0.960
Prediction response to radiotherapy and local progression						0.660–0.800
Prediction response to radiotherapy and progression of brain metastasis		77.0%	74.0%			
Predict outcomes in lung cancer after radiotherapy						0.750–0.800
Survival in chemotherapy patients based on body composition	98.0%					0.730–0.740
Response to chemotherapy plus immune checkpoint inhibitors						0.690–0.750
Prediction response to KRAS^G12C^ inhibitors	85.0%					
Prediction of response to EGFR-TKI treatment and survival benefit				75.4–80.4%		0.748–0.813
Predict response to immune checkpoint inhibitors	75.6–84.0%			90.0%		0.520–0.910
Predict cardiovascular events in patients receiving immunotherapy	94.4–99.2%					0.968–0.996
Predict the outcome in patients receiving immunotherapy	73.0–83.0%					0.670–0.953

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
