# Peer review of "A Thorough Review of the Clinical Applications of Artificial Intelligence in Lung Cancer"

_cancers, 2025, doi:10.3390/cancers17050882_

Round 1
Reviewer 1 Report
Comments and Suggestions for Authors
This paper provides a comprehensive review of the application of AI in the diagnosis and treatment of lung cancer. It covers a wide range of areas, including prevention, diagnosis, and treatment decision-making. The authors should be commended for their efforts in summarizing the vast amount of literature reporting on AI utilization in lung cancer. Overall, I consider this a highly important review that effectively highlights the current state-of-the-art in this field.
While the detailed descriptions in each section are commendable, the central themes of the various topics are somewhat difficult to discern. For example, in the section on AI use in CT scans, both diagnostic applications and treatment response assessments are discussed, leading to some overlap with the later section on treatment decision-making.
Additionally, the extensive amount of information may make it challenging for readers to identify the key features in each section. The inclusion of summarized tables or figures could improve readability and help emphasize the main points more clearly.
Author Response
Comments 1: While the detailed descriptions in each section are commendable, the central themes of the various topics are somewhat difficult to discern. For example, in the section on AI use in CT scans, both diagnostic applications and treatment response assessments are discussed, leading to some overlap with the later section on treatment decision-making.
Response 1: Thank you for your excellent comment. It is true that in many studies several aspects of AI use in lung cancer were discussed. In the example that you mention, several authors presented AI applications in CT scans and then provided treatment response rates, based on the respective CT scan application, in order to support the usefulness of the application that was presented. This was also the case in other sections such as the PET scan or the biomarkers. It is sure that this presentation leads to some overlap with the later section on treatment decision-making. However, in many cases the boundaries between the diagnostic and therapeutic utilities of many applications were indiscernible, therefore, since our review is rather extended, we chose to present our findings in a way that would allow other researchers or clinicians to use each section of our manuscript autonomously, accessing all the necessary information needed in each case for their research or clinical use. Unfortunately, this led to some overlap between the different sections of our manuscript, however, we believe that in this way our manuscript might be more useful for other researchers or clinicians. This is something that we mention in the limitations of our revised manuscript.
Comments 2: Additionally, the extensive amount of information may make it challenging for readers to identify the key features in each section. The inclusion of summarized tables or figures could improve readability and help emphasize the main points more clearly.
Response 2: Thank you for pointing that out. We have added several more tables and/or figures in the revised version of our manuscript.
Reviewer 2 Report
Comments and Suggestions for Authors
WELL DONE!A WELL ARTICULATED ARTICLE THAT COVERS ALL THE ASPECTS OF THE ISSUE.FEW THINGS TO TAKE UNDER CONSIDERATION:MAYBE IT IS PROPER TO NAME THE DIFFERENT PLATFORMS USED AND MAKE A CONFRONT.FOR EXAMPLE THE X-PET PLATFORM WAS MORE /LESS COMPATIBLE WITH THE AI PROGRAMM THAN THE Y-PLATFORM.IT IS IMPORTANT FOR WHOM IS INTERESTED TO PUT IT IN THE ROUTINE.
ANOTHER POINT IS TO ADD CRYSTAL CLEAR WHICH OF THEM ARE IVD AND WHICH STILL IN RESEARCH.
Author Response
Comments 1: MAYBE IT IS PROPER TO NAME THE DIFFERENT PLATFORMS USED AND MAKE A CONFRONT.FOR EXAMPLE THE X-PET PLATFORM WAS MORE /LESS COMPATIBLE WITH THE AI PROGRAMM THAN THE Y-PLATFORM.IT IS IMPORTANT FOR WHOM IS INTERESTED TO PUT IT IN THE ROUTINE.
Response 1: Thank you for pointing that out. We chose not to name the different platforms because we did not want our paper to be considered as more favorable to one platform over another. We felt that, by citing the different results and the respective references, it would be convenient enough to whom it might concern to dig a little further and conclude based on his/her own needs, rather than be biased by our promotion of certain platforms. We think that this approach is the fairest one, since doctors from different countries might value different things in an AI platform based on the conditions of their country (i.e. the financial status), and the same also applies for physicians of different specialties, since surgeons, radiologists, pathologists, cytologists, pulmonologists and oncologists might also need different things from an AI platform. This is something that we mention in the limitations of our revised manuscript.
Comments 2: ANOTHER POINT IS TO ADD CRYSTAL CLEAR WHICH OF THEM ARE IVD AND WHICH STILL IN RESEARCH.
Response 2: Thank you for your excellent comment. We have added this information in the new tables of our revised manuscript.
Reviewer 3 Report
Comments and Suggestions for Authors
1. The study title is rather misleading and ambiguous. Please specify whether the paper is a systematic review or narrative review. Avoid using terms like “atlas” and “primitive” as these are non-specific and could in fact suggest that the findings are incomplete or rudimentary.
2. The introduction section needs to be strengthened, particularly when it comes to the rationale for the present review. There is no clear articulation of why this review is needed at this time, particularly given the existence of numerous similar reviews (DOI: 10.3390/cancers15215236).
3. It is unorthodox to rely on a single database for a systematic review. This seriously undermines the completeness and comprehensiveness of the systematic review. Databases like IEEE Xplore and ACM Digital Library, which are rich sources of computer science and machine learning literature, are especially relevant and should have been searched as well.
4. In many aspects, the review does not follow the PRISMA guidelines. Please provide the full electronic search strategy used to identify studies, including all search terms and limits for at least one database in the main manuscript.
5. Was the review protocol publicly registered in PROSPERO or through the Open Science Framework (osf.io)? This is a requirement under PRISMA. Please indicate if so.
6. The CONSORT diagram is meant for RCTs and not systematic reviews. Please replace the CONSORT diagram with a properly formatted PRISMA diagram. The number of studies included and excluded should be broken down clearly at each stage. Details about why studies were excluded after screening and eligibility should be provided. The current diagram does not make it easy to trace the flow from initial search results through to the final included studies.
7. Many crucial and relevant performance metrics were either not reported or inadequately covered in this review, limiting the evaluation of AI models for lung cancer. Please include detailed reporting on computational complexity, training time, robustness, and generalizability etc. in the results section.
8. Please avoid making strong claims about the accuracy of AI models (e.g., “near or over 90%,” “in some cases 100%”). You need to critically assess these claims or discuss the potential biases in the underlying data.
9. There are also many practical issues surrounding the clinical integration of these AI models, including their ease of use, interpretability, and the infrastructural requirements for hospitals or clinics to adopt these technologies (DOI: 10.2196/preprints.62941). This should be at least briefly discussed.
10. "Although there haven’t been many applications of AI in the prevention of the appearance of lung cancer per se ..." - Odd phrasing and convoluted sentence, please rephrase this.
Comments on the Quality of English LanguageModerate edits required.
Author Response
Comments 1: The study title is rather misleading and ambiguous. Please specify whether the paper is a systematic review or narrative review. Avoid using terms like “atlas” and “primitive” as these are non-specific and could in fact suggest that the findings are incomplete or rudimentary.
Response 1: Thank you for your suggestion. We have changed the title of our revised manuscript accordingly.
Comments 2: The introduction section needs to be strengthened, particularly when it comes to the rationale for the present review. There is no clear articulation of why this review is needed at this time, particularly given the existence of numerous similar reviews (DOI: 10.3390/cancers15215236).
Response 2: Thank you for pointing that out and give us the opportunity to improve this section of our manuscript. We have enriched the “Introduction” section of our revised manuscript according to your suggestions.
Comments 3: It is unorthodox to rely on a single database for a systematic review. This seriously undermines the completeness and comprehensiveness of the systematic review. Databases like IEEE Xplore and ACM Digital Library, which are rich sources of computer science and machine learning literature, are especially relevant and should have been searched as well.
Response 3: Thank you for your comment which gives us the opportunity to justify that. It is true that in previous decades it might have been unorthodox to rely on a single database for a comprehensive review. However, nowadays, the majority of the medical journals and their respective articles are indexed in multiple databases, including PubMed, which is a main one. Furthermore, PubMed comprises more than 37 million citations for biomedical literature from MEDLINE, life science journals, and online books and is widely considered as the most comprehensive and qualitative medical database available. Considering all that, we feel that the systematic search of medical databases other than PubMed is a waste of valuable resources, since it would require far too many resources for a questionable gain in the quality of our manuscript, a fact that according to medical research methodology would be pointless or even unethical. For example, according to evidence-based medicine, before a randomized controlled trial is conducted, a sample size calculation is performed, and the inclusion of a higher number of participants is considered unethical for different reasons, and wasting valuable resources is one of them. As far as the other non-medical databases that you mention, we agree that they are rich sources of computer science and machine learning literature, however, we do not consider them particularly relevant with our paper, since our goal was to present the applications of AI in lung cancer from the physicians’ point of view, while those databases comprise mainly scientific literature that is more suited for computer engineers. In any case, as we explain in the answer of your 5th point, we have mistakenly considered our review a systematic one, since it would be better described as a comprehensive one. This is something that we mention in the limitations of our revised manuscript.
Comments 4: In many aspects, the review does not follow the PRISMA guidelines. Please provide the full electronic search strategy used to identify studies, including all search terms and limits for at least one database in the main manuscript.
Response 4: Thank you for your suggestion. In the “Materials and Methods” section of our revised manuscript we cite: “For the purpose of the present comprehensive review, we performed a PubMed search in “All Fields” for “artificial intelligence” OR “artificial neural network” OR “convolutional neural network” OR “deep learning” OR “machine learning” OR “omics” AND “lung cancer” to the 31st of December 2023 with no start date. This search identified 6095 unique results, 11 of which were duplicates and were removed before screening. During screening no studies were removed by automation tool, 147 studies were removed because their reports could not be retrieved, 9 studies were removed because they were editorials, 28 studies were removed because they were comments on articles, 7 studies were removed because they were meeting abstracts and 5420 studies were removed because they were irrelevant to the research theme. As a result, 5622 items were removed in total and 473 were included in this review. Interestingly, more than 75% of all the papers included in this review were published the last five years and more than 90% the last decade. Articles were categorized based on the thematic sectors of this review. Some of the studies included, were used in more than one thematic sector. For ‘epidemiology’, 78 studies were used (14 for the ‘prevention’ sector and 69 for the ‘screening’ sector). For the ‘clinical features’ sector, 23 studies were used. For ‘diagnosis’, 350 studies were used (25 for the ‘X-ray’ sector, 181 for the ‘chest CT’ sector, 44 for the ‘PET’ sector, 62 for the biomarkers sector, 80 for the ‘biopsy’ sector, and 62 for the ‘staging’ sector). For ‘treatment’, 183 studies were used (75 for the ‘general information’ sector, 21 for the ‘surgical treatment’ sector, 25 for the ‘radiotherapy’ sector, 13 for the ‘chemotherapy’ sector, 26 for the ‘targeted therapy’ sector and 42 for the ‘immunotherapy’ sector). Finally, 123 studies were used for the ‘prognosis’ sector.”.
Comments 5: Was the review protocol publicly registered in PROSPERO or through the Open Science Framework (osf.io)? This is a requirement under PRISMA. Please indicate if so.
Response 5: Thank you for pointing that out. It is true that we have omitted that and consequently, our review cannot be justified as a systematic one, therefore, in our revised manuscript we have replaced the word “systematic” with the word “comprehensive”.
Comments 6: The CONSORT diagram is meant for RCTs and not systematic reviews. Please replace the CONSORT diagram with a properly formatted PRISMA diagram. The number of studies included and excluded should be broken down clearly at each stage. Details about why studies were excluded after screening and eligibility should be provided. The current diagram does not make it easy to trace the flow from initial search results through to the final included studies.
Response 6: Thank you for your excellent comment which gives us the opportunity to improve the quality of our manuscript. In our revised manuscript, we have provided a properly formatted PRISMA diagram, following your detailed suggestions regarding its content.
Comments 7: Many crucial and relevant performance metrics were either not reported or inadequately covered in this review, limiting the evaluation of AI models for lung cancer. Please include detailed reporting on computational complexity, training time, robustness, and generalizability etc. in the results section.
Response 7: Thank you for pointing that out. However, as we have already explained in our answer of your 3rd point, our goal was to present the applications of AI in lung cancer from the physicians’ point of view, therefore we focused on results concerning accuracy, sensitivity, specificity, positive predictive value (ppv) negative predictive value (npv), area under the curve (AUC), etc., which constitute useful indices for physicians to evaluate each AI model from a clinical point of view. The other indices that you mention, although interesting, are beyond the scope of the present review and additionally, since the papers that we included in our review are medical ones, they usually do not include indices like computational complexity, training time, robustness and generalizability. Perhaps, this could be the theme of a future review, something that we mention in the limitations of our revised manuscript.
Comments 8: Please avoid making strong claims about the accuracy of AI models (e.g., “near or over 90%,” “in some cases 100%”). You need to critically assess these claims or discuss the potential biases in the underlying data.
Response 8: Thank you for pointing that out and giving us the opportunity to improve our manuscript. In the present review we have just cited the results that are reported in each paper that we have included. However, we agree that there is possibly a significant bias in those results, since those who report them are the creators of the respective AI models. Therefore, we have discussed this potential bias in the limitations of our revised manuscript.
Comments 9: There are also many practical issues surrounding the clinical integration of these AI models, including their ease of use, interpretability, and the infrastructural requirements for hospitals or clinics to adopt these technologies (DOI: 10.2196/preprints.62941). This should be at least briefly discussed.
Response 9: Thank you for pointing that out. However, as we have already explained in our answer of your 3rd and 7th point, our goal was to present the applications of AI in lung cancer from the physicians’ point of view, therefore we focused on results concerning accuracy, sensitivity, specificity, positive predictive value (ppv) negative predictive value (npv), area under the curve (AUC), etc., which constitute useful indices for physicians to evaluate each AI model from a clinical point of view. The other indices that you mention, although interesting, are beyond the scope of the present review and additionally, since the papers that we included in our review are medical ones, they usually do not include indices like ease of use, interpretability, and infrastructural requirements for hospitals or clinicians to adopt these technologies. Perhaps, this could be the theme of a future review, something that we mention in the limitations of our revised manuscript.
Comments 10: "Although there haven’t been many applications of AI in the prevention of the appearance of lung cancer per se ..." - Odd phrasing and convoluted sentence, please rephrase this.
Response 10: Thank you for pointing that out. We have changed this phrase. Additionally, one of the co-authors who is a native English speaker has checked our revised manuscript for spelling, grammatical and syntactic errors and changed every phrase or sentence that she considered it as odd or convoluted.
Reviewer 4 Report
Comments and Suggestions for Authors
A very large review, the authors tried to cover all aspects of using AI in diagnostics, treatment, and prognosis of lung cancer. However, the review does not contain a single table and/or figure, except for the study design, which impairs the perception of information.
1. Is AI an area of ​​scientific research or are clinical trials involving AI already being conducted? If so, it would be interesting to see a list of such clinical trials, including status, number of patients, etc.
2. Add tables, for example in p. 3.3.4. - a list of biomarkers, description of studies, AI model, result, etc. There may be several such summary tables in a manuscript.
3. To what extent does the use of AI affect the indicators of accuracy, sensitivity, and reproducibility? Is it possible to provide a summary table of how much, on average, for each subsection, the characteristics of the method change when using AI?
4. Is progress in the field of lung cancer research associated with the introduction of AI technology or with the growing diversity of methods for prognosis, diagnosis, and treatment?
Author Response
Comments 1: Is AI an area of ​​scientific research or are clinical trials involving AI already being conducted? If so, it would be interesting to see a list of such clinical trials, including status, number of patients, etc.
Response 1: Thank you for pointing that out. AI is no more just an area of scientific research, but clinical trials involving AI have already been or are being conducted at the moment. We have added several more tables and/or figures in the revised version of our manuscript, in which we cite the type of research of each paper that we have included in our review.
Comments 2: Add tables, for example in p. 3.3.4. - a list of biomarkers, description of studies, AI model, result, etc. There may be several such summary tables in a manuscript.
Response 2: Thank you for your excellent comment that gives us the opportunity to improve the respective sections of our manuscript. We have added several more tables and/or figures in the revised version of our manuscript, including those ones that you suggest.
Comments 3: To what extent does the use of AI affect the indicators of accuracy, sensitivity, and reproducibility? Is it possible to provide a summary table of how much, on average, for each subsection, the characteristics of the method change when using AI?
Response 3: Thank you for your excellent comment that gives us the opportunity to improve the “conclusion” section of our manuscript. We have added several more tables and/or figures in the revised version of our manuscript, including this one.
Comments 4: Is progress in the field of lung cancer research associated with the introduction of AI technology or with the growing diversity of methods for prognosis, diagnosis, and treatment?
Response 4: Thank you for your comment. The progress in the field of lung cancer research is mainly associated with the growing diversity of methods for prognosis, diagnosis and treatment. However, the introduction of AI technology is part of this process and is expected to boost it further. This is something that we mention in the conclusion section of our revised manuscript.
Round 2
Reviewer 3 Report
Comments and Suggestions for Authors
1. I am not sure what is meant by a 'comprehensive' review. I would suggest the authors rewrite this carefully and structure it as a 'scoping review' instead. The "comprehensive" label in the title is misleading, as the review mostly lists studies without critically analyzing their findings or methodologies.
2. The search strategy was limited to solely PubMed, which is too limited for any kind of review and this risks excluding relevant studies indexed in other databases like Embase, IEEE Xplore, or Scopus etc. This also undermines the claim of being "comprehensive."
3. There is an excessive number of tables, each detailing study characteristics in a verbose manner. This overwhelms the reader and disrupts the flow of the paper. It would be more effective to have effective summarizing figures or diagrams.
4. Tables can be more concise, e.g., proper columns reporting aggregated data, such as total number of studies by type, range of performance metrics (e.g., sensitivity: 80-95%), findings on use cases etc.
5. There is minimal discussion on ethical concerns related to AI in lung cancer, such as data privacy, biases in training datasets, or patient consent. This is typically a pertinent issue especially given the paucity of guidelines.
6. The practical implementation of AI in low-resource settings is superficially addressed. Challenges like infrastructure needs, cost-effectiveness, and cultural barriers should be cited (citation: doi.org/10.3390/cancers17020197).
Comments on the Quality of English LanguageMany sentences make little to no sense, e.g., "Lung cancer tends to become a global epidemic since it is categorized particularly high in the leading causes of death, not only in developed countries, but also worldwide, and of course occupies the leading position in cancer-related mortality." I would highly suggest proofreading by a native English speaker or a professional copyediting service, please.
Author Response
Comments 1: I am not sure what is meant by a 'comprehensive' review. I would suggest the authors rewrite this carefully and structure it as a 'scoping review' instead. The "comprehensive" label in the title is misleading, as the review mostly lists studies without critically analyzing their findings or methodologies.
Response 1: Thank you for your suggestion. We have changed the word “comprehensive” with the word “thorough”, where appropriate, throughout the manuscript.
Comments 2: The search strategy was limited to solely PubMed, which is too limited for any kind of review and this risks excluding relevant studies indexed in other databases like Embase, IEEE Xplore, or Scopus etc. This also undermines the claim of being "comprehensive."
Response 2: Thank you for your comment. We have changed the word “comprehensive” with the word “thorough”, where appropriate, throughout the manuscript.
Comments 3: There is an excessive number of tables, each detailing study characteristics in a verbose manner. This overwhelms the reader and disrupts the flow of the paper. It would be more effective to have effective summarizing figures or diagrams.
Response 3: Thank you for your comment. It is true that the tables of our revised manuscript are very comprehensive, a fact that might be overwhelming for some readers. However, we have created these tables following the request of the other 3 reviewers and the academic editor during the 1st round of peer review, therefore we cannot change them now, since this would contradict with their request. During this 2nd round of peer review, we have shortened these tables considerably, by removing from them all the studies that were reviews or systematic reviews. In these tables, we summarize the characteristics and the findings of every study that we included in our paper. The text that proceeds each table is effectively the summary of the information that are presented in the table. Furthermore, in the last table of our revised manuscript, we present all the performance metrics concisely.
Comments 4: Tables can be more concise, e.g., proper columns reporting aggregated data, such as total number of studies by type, range of performance metrics (e.g., sensitivity: 80-95%), findings on use cases etc.
Response 4: Thank you for your comment. It is true that the tables of our revised manuscript are very comprehensive, a fact that might be overwhelming for some readers. However, we have created these tables following the request of the other 3 reviewers and the academic editor during the 1st round of peer review, therefore we cannot change them now, since this would contradict with their request. During this 2nd round of peer review, we have shortened these tables considerably, by removing from them all the studies that were reviews or systematic reviews. In these tables, we summarize the characteristics and the findings of every study that we included in our paper. The text that proceeds each table is effectively the summary of the information that are presented in the table. Furthermore, in the last table of our revised manuscript, we present all the performance metrics concisely.
Comments 5: There is minimal discussion on ethical concerns related to AI in lung cancer, such as data privacy, biases in training datasets, or patient consent. This is typically a pertinent issue especially given the paucity of guidelines.
Response 5: Thank you for raising this issue in this 2nd round of peer review, even though it was not mentioned at all in the 1st round. As we have already explained in the previous round, our review focused on the clinical applications of AI on lung cancer. Consequently, the discussion on ethical concerns was minimal and only relevant to the clinical applications. Nevertheless, in this 2nd round, we have also added that in the limitations of our study.
Comments 6: The practical implementation of AI in low-resource settings is superficially addressed. Challenges like infrastructure needs, cost-effectiveness, and cultural barriers should be cited (citation: doi.org/10.3390/cancers17020197).
Response 6: Thank you for raising this issue in this 2nd round of peer review, even though it was not mentioned at all in the 1st round. As we have already explained in the previous round, our review focused on the clinical applications of AI on lung cancer. Consequently, the discussion about the practical implementation of AI in low-resource settings was minimal and only relevant to the clinical applications. Nevertheless, in this 2nd round, we have also added that in the limitations of our study. Furthermore, we have already added in our manuscript the references that we considered relevant after reviewing the literature. Adding references that were suggested by the reviewers is not ethical, especially when reviewers choose not to sign their reports, since this could be a try for self-citation.
Comments 7: Many sentences make little to no sense, e.g., "Lung cancer tends to become a global epidemic since it is categorized particularly high in the leading causes of death, not only in developed countries, but also worldwide, and of course occupies the leading position in cancer-related mortality." I would highly suggest proofreading by a native English speaker or a professional copyediting service, please.
Response 7: Thank you for pointing that out. A native English speaker has reviewed our manuscript for grammatical and syntactic errors for the 3rd time (original submission, 1st round of peer review and now).
Reviewer 4 Report
Comments and Suggestions for Authors
The authors have substantially revised the manuscript and taken into account the reviewer's comments. Table 18 is exactly what the manuscript was missing in its original version. However, I have a comment: usually, references to other reviews are not included in summary tables, but only experimental studies are used. All references to reviews should be removed from the tables. Reviews can be referenced in the text of the manuscript, if necessary.
Author Response
Comments 1: The authors have substantially revised the manuscript and taken into account the reviewer's comments. Table 18 is exactly what the manuscript was missing in its original version. However, I have a comment: usually, references to other reviews are not included in summary tables, but only experimental studies are used. All references to reviews should be removed from the tables. Reviews can be referenced in the text of the manuscript, if necessary.
Response 1: Thank you for your comment, which gives us the opportunity to improve our manuscript. We have removed all references to reviews from our revised manuscript.